# AdaDetectGPT: Adaptive Detection of LLM-Generated Text with Statistical Guarantees

**Hongyi Zhou**[*]
Department of Mathematics
Tsinghua University
Beijing, China

**Jin Zhu**[*]
School of Mathematics
University of Birmingham
Birmingham, UK

**Pingfan Su**
Department of Statistics
LSE
London, UK

**Kai Ye**
Department of Statistics
LSE
London, UK

**Ying Yang**
Department of Statistics and Data Science
Tsinghua University
Beijing, China

**Shakeel A O B Gavioli-Akilagun**[†]
Department of Decision Analytics and Operations
City University Hong Kong
Hongkong, China

**Chengchun Shi**[†]
Department of Statistics
LSE
London, UK

## Abstract

We study the problem of determining whether a piece of text has been authored by a human or by a large language model (LLM). Existing state of the art logits-based detectors make use of statistics derived from the log-probability of the observed text evaluated using the distribution function of a given source LLM. However, relying solely on log probabilities can be sub-optimal. In response, we introduce AdaDetectGPT – a novel classifier that adaptively learns a witness function from training data to enhance the performance of logits-based detectors. We provide statistical guarantees on its true positive rate, false positive rate, true negative rate and false negative rate. Extensive numerical studies show AdaDetectGPT nearly uniformly improves the state-of-the-art method in various combination of datasets and LLMs, and the improvement can reach up to 37%. A python implementation of our method is available at `https://github.com/Mamba413/AdaDetectGPT`.

## 1 Introduction

Large language models (LLMs) such as ChatGPT (OpenAI, 2022), PaLM (Chowdhery et al., 2023), Llama (Grattafiori et al., 2024) and DeepSeek (Bi et al., 2024) have revolutionized the field of generative artificial intelligence by enabling large-scale content generation across various fields including journalism, education, and creative writing (Demszky et al., 2023; Milano et al., 2023; Doshi & Hauser, 2024). However, their ability to produce highly human-like text poses serious risks, such as the spread of misinformation, academic dishonesty, and the erosion of trust in written communication (Ahmed et al., 2021; Lee et al., 2023; Christian, 2023). Consequently, accurately distinguishing between human- and LLM-generated text has emerged as a critical area of research.

There is a growing literature on the detection of machine-generated text; refer to Section 1.1 for a review. One popular line of research focuses on statistics-based detectors, typically rely on log-

---

[*]Hongyi Zhou and Jin Zhu contributed equally to this paper and are listed in alphabetical order.
[†]Corresponding authors: `sgavioli@cityu.edu.hk`, `c.shi7@lse.ac.uk`

39th Conference on Neural Information Processing Systems (NeurIPS 2025).

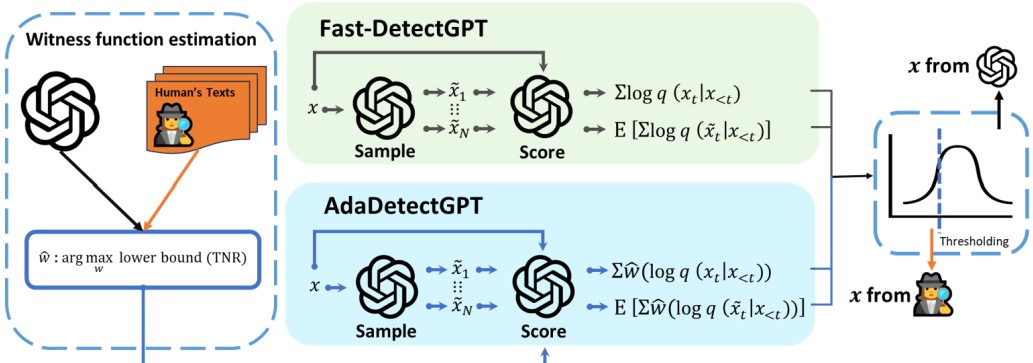

Figure 1: Workflow of AdaDetectGPT. Built upon Fast-DetectGPT (Bao et al., 2024), our method adaptively learn a witness function $\widehat{w}$ from training data by maximizing a lower bound on the TNR, while using normal approximation for FNR control.

probability outputs (i.e., logits) from a source LLM to construct the statistics for classification (see e.g., Gehrmann et al., 2019; Mitchell et al., 2023). These works are motivated by the empirical observation that LLM-generated text tends to exhibit higher log-probabilities or larger differences between the logits of original and perturbed tokens. However, as we demonstrated in Section 3, relying solely on the logits can be sub-optimal for detecting LLM-generated text.

**Our contribution**. In this paper, we propose AdaDetectGPT (see Figure 1 for a visualization), an adaptive LLM detector that leverages external training data to enhance the effectiveness of existing logits-based detectors. Our approach derives a lower bound on the true negative rate (TNR) of logits-based detectors and adaptively learns a witness function by optimizing this bound, resulting in a more powerful detection statistic. The optimization is straightfoward and requires only solving a system of linear equations. Based on this statistic, we further introduce an approach to select the classification threshold that controls AdaDetectGPT's false negative rate (FNR).

Empirically, we conduct extensive evaluations across multiple datasets and a variety of target language models to demonstrate that AdaDetectGPT consistently outperforms existing logits-based detectors. In white-box settings – where the target LLM to be detected is the same as the source LLM used to compute the logits – AdaDetectGPT achieves improvements over the best alternative in area under the curve (AUC) ranging from 12.5% to 37%. In black-box settings, where the source and target LLMs differ, it similarly offer gains of up to 20%.

Theoretically, we provide statistical performance guarantees for AdaDetectGPT, deriving finite-sample error bounds for its TNR, FNR, true positive rate (TPR) and false positive rate (FPR). Existing literature on logits-based detectors generally lacks systematic statistical analysis. Our work aims to fill in this gap and contribute toward a deeper understanding of these methods in this emerging field, by offering a comprehensive analysis based on the aforementioned standard classification metrics.

## 1.1 Related works

Existing methods for detecting machine-generated text generally fall into three categories: machine learning (ML)-based, statistics-based, and watermarking-based; see Yang et al. (2024c); Wu et al. (2025) for recent comprehensive surveys. Our method is most closely related to the first two categories and, unlike the third, does not rely on knowledge of the specific hash function or random number generator used during token generation, which are often model-specific and not publicly available. In what follows, we review the first two categories and defer the discussion of watermarking-based approaches to Appendix A.

**ML-based detection**. ML-based methods train classification models on external human- and machine-authored text for detection. Many methods can be further categorized into two types. The first type extracts certain features from text and apply classical ML models to train classifiers based on these features. Various features have been proposed in the literature, ranging from classical term frequency-inverse document frequency (TF-IDF), unigram, and bigram features (Solaiman et al., 2019), to more complex features engineered specifically for this task, such as the cross-entropy loss computed between the source text and a surrogate LLM (Guo et al., 2024a) and the rewriting-based measure

that quantifies the difference between original texts and their LLM-rewritten versions (Mao et al., 2024).

The second type of methods fine-tune LLMs directly for classification. This approach is intuitive, as LLMs are inherently designed for processing text data; we only need to modify the model's output to predict a binary label rather than token probabilities. Various LLMs have been employed for fine-tuning, including RoBERTa (Solaiman et al., 2019; Guo et al., 2023), BERT (Ippolito et al., 2020) and DistilBERT (Mitrović et al., 2023).

In addition to these two types of methods, Abburi et al. (2023) propose a hybrid approach that uses the outputs of fine-tuned LLMs as input features for classical ML-based classification. Further efforts have focused on handling adversarial attacks (Crothers et al., 2022; Krishna et al., 2023; Koike et al., 2024; Sadasivan et al., 2025), short texts (Tian et al., 2024), out-of-distribution texts (Guo et al., 2024b), unobserved prompts (Zhou et al., 2026a), biases against non-native English writers (Liang et al., 2023), accommodating statistical inference (Zhou et al., 2026b), as well as the downstream applications of these methods in domains such as education, social media and medicine (Herbold et al., 2023; Kumarage et al., 2023; Liao et al., 2023).

**Statistics-based detection**. Statistics-based methods leverage differences in token-level metrics such as log-probabilities to distinguish between human- and machine-authored text. Unlike ML-based approaches, many of these methods do not rely on external training data; instead, they directly use predefined statistical measures as classifiers. In particular, a seminal work by Gehrmann et al. (2019) propose several such measures, including the average log-probability and the distribution of absolute ranks of probabilities of tokens across a text. These measures exhibit substantial differences between human- and machine-authored text, and they have been widely employed and extended in the literature (see e.g., Mitchell et al., 2023; Su et al., 2023; Bao et al., 2024; Hans et al., 2024).

Other statistical measures employed are calculated based on the N-gram distributions (Solaiman et al., 2019; Yang et al., 2024b), the intrinsic dimensionality of text (Tulchinskii et al., 2023), the reward model used in LLMs (Lee et al., 2024) and the maximum mean discrepancy (Zhang et al., 2024; Song et al., 2025). Recent works have extended these methods to more challenging scenarios, such as to handle adversarial attacks (Hu et al., 2023), machine-revised text (Chen et al., 2025a) and black-box settings (Yu et al., 2024; Zeng et al., 2024). Theoretically, Chakraborty et al. (2024) establish a sample complexity bound for detecting machine-generated text.

To conclude this section, we remark that our proposal lies at the intersection between statistics- and ML-based methods. Similar to many statistics-based approaches, our classifier is constructed based on the log-probabilities. However, we adaptively learn a witness function via ML to improve its effectiveness. In this way, our method leverages the strengths of both approaches, leading to superior detection performance.

## 2   Preliminaries

We first define the white-box and black-box settings as well as our objective. We next review two baseline methods, DetectGPT (Mitchell et al., 2023) and Fast-DetectGPT (Bao et al., 2024), as they are closely related to our proposal. Finally, we introduce the martingale central limit theorem (see e.g., Hall & Heyde, 2014), which serves as the theoretical basis for our threshold selection.

**Task and settings**. We study the problem of determining whether a given passage $X$, represented as a sequence of tokens $(X_1, X_2, \ldots, X_L)$, was authored by a human or generated by a target LLM. Specifically, let $p$ and $q$ denote the distributions over human-written and LLM-generated tokens, respectively. Each distribution can be represented as a product of conditional probability functions, $\prod_t p_t$ for humans and $\prod_t q_t$ for the target LLM, where each $p_t(x_t|x_{<t})$ (and similarly $q_t$) denotes the conditional probability mass function of the next token $x_t$ given the preceding tokens, where $x_{<t} := (x_1, x_2, \ldots, x_{t-1})$ when $t > 1$ and $x_{<t} := \emptyset$ otherwise. Our goal is to develop a classifier to discriminate between $X \sim p$ (human) and $X \sim q$ (LLM).

We assume access to a source LLM's probability distribution function $q' = \prod_t q'_t$. When $q' = q$, it corresponds to the *white-box setting* where the source model we have is the same as the target model we wish to detect. This is the primary setting considered in this paper. For closed-source LLMs such as GPT-3.5 and GPT-4, their probability functions are not publicly available. In such cases, we utilize

an open-source model with distribution $q'$ as an approximation of $q$, resulting in the *black-box setting*, which our method is also extended to handle.

We also assume access to a corpus of $n$ human-authored passages $\mathcal{H} = \{\boldsymbol{X}^{(i)}\}_{i=1}^n \sim p$. This assumption is reasonable, as large corpora of human-written text are readily available online (e.g., Wikipedia). Without loss of generality, we assume that all passages have the same number of tokens $L$, achieved by zero-padding shorter sequences to match the maximum token length. Throughout this paper, we use boldface letters (e.g., $\boldsymbol{X}$) to denote passages, and non-boldface letters (e.g., $X$) to denote individual tokens.

**Baseline methods**. Both DetectGPT and Fast-DetectGPT are statistics-based and rely on the log-probability of a passage, $\log q'(\boldsymbol{X})$, as the basis for classification. Specifically, DetectGPT considers the following statistic:

$$\frac{\log q'(\boldsymbol{X}) - \mathbb{E}_{\widetilde{\boldsymbol{X}} \sim p'(\bullet|\boldsymbol{X})}[\log q'(\widetilde{\boldsymbol{X}})]}{\sqrt{\mathrm{Var}_{\widetilde{\boldsymbol{X}} \sim p'(\bullet|\boldsymbol{X})}(\log q'(\widetilde{\boldsymbol{X}}))}}, \tag{1}$$

where both the expectation in the numerator and the variance in the denominator are evaluated under a perturbation function $p'$, which produces $\widetilde{\boldsymbol{X}}$ that is a slightly modified version of $\boldsymbol{X}$ with similar meaning. The rationale behind this statistic is that, empirically, machine-generated text tends to yield higher values than human-written text when evaluated using (1) (Mitchell et al., 2023, Figure 2). As a result, a passage is classified as machine-generated if this statistic is larger than a certain threshold.

A potential limitation of DetectGPT is that sampling from the perturbation distribution $p'$ requires multiple calls to the source LLM to generate rewritten versions of the input passage, making the calculation of (1) computationally expensive. Fast-DetectGPT addresses this issue by proposing a modified version of (1), given by

$$\frac{\sum_t \log q'_t(X_t|X_{<t}) - \sum_t \mathbb{E}_{\widetilde{X}_t \sim s_t(\bullet|X_{<t})} \log q'_t(\widetilde{X}_t|X_{<t})}{\sqrt{\sum_t \mathrm{Var}_{\widetilde{X}_t \sim s_t(\bullet|X_{<t})}(\log q'_t(\widetilde{X}_t|X_{<t}))}}. \tag{2}$$

Specifically, notice that $\log q'(\boldsymbol{X})$ can be decomposed as a token-wise sum $\sum_t \log q_s(X_t|X_{<t})$. Thus, the first term in the numerator of (1) is the same as that in (2). However, Fast-DetectGPT replaces the centering term in (1) with $\sum_t \mathbb{E}_{\widetilde{X}_t \sim s_t(\bullet|X_{<t})} \log q'_t(\widetilde{X}_t|X_{<t})$. Here, $s = \prod_t s_t$ denotes a sampling distribution function which may equal $q$ or be derived from another LLM. By replacing the perturbation function with $s$, the centering term can be efficiently computed directly from the LLM's conditional probabilities. Additionally, due to the conditioning on $X_{<t}$ in the centering term, the variance term is equal to the sum of the conditional variances of $\log q'_t(\widetilde{X}_t|X_{<t})$. Finally, it classifies a passage as machine-generated if this modified statistic is larger than a certain threshold.

**Martingale central limit theorem**. The martingale central limit theorem (MCLT) is a fundamental result in probability theory that enables rigorous statistical inference for time-dependent data. It is well-suited for analyzing text data where tokens are generated sequentially given their predecessors. Consider a time series $\{Z_t\}_t$ where each $Z_t$ represents a real-valued random variable. Suppose there exists a sequence of monotonically increasing sets of random variables $\mathcal{F}_1 \subseteq \mathcal{F}_2 \subseteq \cdots$ so that $Z_t \in \mathcal{F}_t$ for any $t$. Under certain regularity conditions, MCLT states that the normalized partial sum

$$\frac{\sum_{t=1}^L [Z_t - \mathbb{E}(Z_t|\mathcal{F}_{t-1})]}{\sqrt{\sum_{t=1}^L \mathrm{Var}(Z_t|\mathcal{F}_{t-1})}} \tag{3}$$

converges in distribution to a standard normal random variable as $L$ approaches infinity (Brown, 1971). Notice that the statistic in (3) shares similar structures with that employed in Fast-DetectGPT (see (2)). In the next section, we will leverage this connection for FNR control through normal approximation.

## 3 AdaDetectGPT

We present AdaDetectGPT in this section. We begin by discussing the white-box setting in Parts (a)–(c): Part (a) introduces the proposed statistical measure; Part (b) discusses our choice of the classification threshold for FNR control; Part (c) derives a lower bound on the TNR to learn the witness

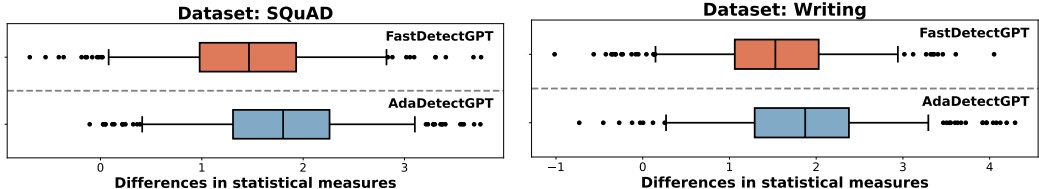

Figure 2: Boxplots visualizing the differences in the statistical measures between human- and LLM-authored passages, comparing AdaDetectGPT (with a learned witness function) and Fast-DetectGPT (without it). A larger positive difference from zero indicates better detection power. As observed, the difference computed by AdaDetectGPT is consistently larger than that of Fast-DetectGPT across the first quartile, median, and third quartile. The left panel shows statistics evaluated on the SQuAD dataset, while the right panel displays results for the WritingPrompts dataset.

function. Next, in Part (d), we extend our proposal to the black-box setting. Finally, in Part (e), we establish the statistical properties of AdaDetectGPT.

**(a) Statistical measure**. Notice that $q' = q$ under the white-box setting. Given a passage $\boldsymbol{X}$, the proposed classifier is built upon the following statistic:

$$T_w(\boldsymbol{X}) := \frac{\sum_t [w(\log q_t(X_t|X_{<t})) - \mathbb{E}_{\widetilde{X}_t \sim q_t(\bullet|X_{<t})} w(\log q_t(\widetilde{X}_t|X_{<t}))]}{\sqrt{\sum_t \mathrm{Var}_{\widetilde{X}_t \sim q_t(\bullet|X_{<t})}(w(\log q_t(\widetilde{X}_t|X_{<t})))}}, \tag{4}$$

where $w : \mathbb{R} \to \mathbb{R}$ denotes a one-dimensional witness function defined over the space of log-probabilities.

By definition, $T_w(\boldsymbol{X})$ is very similar to Fast-DetectGPT's statistic in (2), with two modifications: (i) First, rather than using the raw log conditional probability $q_t$, we apply a witness function $w$ to these $\log q_t$ to enhance the detection power of the resulting classifier. Our numerical experiments demonstrate that this transformation better distinguishes between human- and machine-authored text (see Figure 2). Below, we further provide a simple analytical example to illustrate this power enhancement. (ii) Second, we set the the sampling function $s$ in (2) to the source LLM's $q$. This allows the numerator to match the form of the partial sum in (3), which enables the application of MCLT for FNR control.

*An analytical example*. Consider a hypothetical "Kingdom of Bit" where all communication uses just two tokens: 1 (yes) and 0 (no). Let $\{p_t\}_t$ denote the true token distribution of this language. Suppose a malicious wizard creates synthetic citizens who appear similar to ordinary people, but their language follows a simpler distribution $q_t(x_t|x_{<t}) = q(x_t)$ for some fixed function $q$, being independent of the prior context $x_{<t}$. As we will show later, the detection power of our statistic crucially depends on the following quantity:

$$\frac{1}{L} \sum_{t=1}^{L} \left[ \mathbb{E}_{\widetilde{X}_t \sim q} w(\log q(\widetilde{X}_t)) - \mathbb{E}_{X_t \sim p} w(\log q(X_t)) \right]. \tag{5}$$

The greater the deviation of the expression (5) from zero, the higher the power to detect synthetic humans. When setting $w$ to the identity function, this expression reduces to

$$\log\left(\frac{q(1)}{q(0)}\right) \times \left[ q(1) - \frac{1}{L} \sum_{t=1}^{L} p_t(1) \right],$$

where $p_t(1) = \mathbb{E}_{X_{<t} \sim p} p(1|X_{<t})$ is determined by the true language distribution $p$. In this case, (5) converges to zero as $q(1) \to 1/2$, regardless of the difference between $q(1)$ and the average of $p_t(1)$. As such, there are simple settings in which existing logits-based detectors with an identity witness function will struggle to distinguish between human and machine authored text, independent of the actual distance between $p$ and $q$. However, for any $q(1) \neq 1/2$, there exists a function $w$ that makes (5) equal to $q(1) - L^{-1} \sum_{t=1}^{L} [p_t(1)]$, independent of the log ratio (see Appendix B for a formal proof). Thus, whenever $q(1)$ differs from the average $p_t(1)$, an appropriate transformation $w$ can reliably detect synthetic humans.

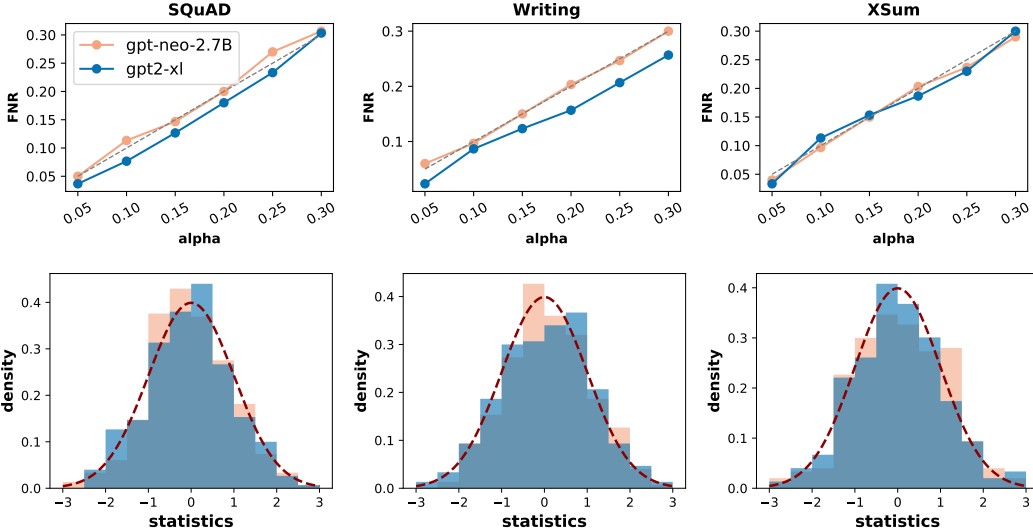

Figure 3: Top panel: FNR of the classifier plotted against the significance level $\alpha$. Bottom panel: Distribution of statistics evaluated on LLM-generated text. The dashed red line is the density function of standard normal random variable. Results are shown across three datasets (from left to right) and two language models (indicated by different colors).

We will discuss how to properly learn the witness function below. Since this is inherently a classification problem, it is natural to learn a witness function that maximizes the AUC of the resulting detector. Equivalently, this amounts to finding a witness function that maximizes the TNR for each fixed FNR. In (b), we discuss how to select the classification threshold to control the FNR at a specified level. Given this threshold, we then derive a lower bound on the TNR in (c) to be maximized for learning the witness function.

**(b) Classification threshold**. Given a witness function $w$, we aim to determine a threshold $c$ so that the FNR of the classifier $T_w(\boldsymbol{X}) > c$ is below a specified level $\alpha > 0$. A key observation is that, when the passage $\boldsymbol{X}$ is generated from the source LLM, $T_w(\boldsymbol{X})$ in (4) can be represented by the partial sum in (3) with $Z_t = w(\log q_t(X_t|X_{<t}))$ and $\mathcal{F}_t = X_{<t}$. It follows from the MCLT that

$$\text{FNR}_w := \mathbb{P}_{\boldsymbol{X} \sim q}(T_w(\boldsymbol{X}) \leq c) \to \Phi(c), \quad \text{as } L \to \infty, \tag{6}$$

where $\Phi(\bullet)$ denotes the cumulative distribution function of a standard normal random variable. Thus, to ensure the desired FNR control, we set the threshold $c$ to the $\alpha$th quantile of $\Phi$, denoted by $z_\alpha$.

We validate this threshold selection both theoretically and empirically. Specifically, Theorem 2 in (e) establishes a finite-sample error bound for our classifier's FNR. Figure 3 illustrates the effectiveness of FNR control and normal approximation across three benchmark datasets and two language models.

**(c) Learning the witness function**. With the classifier's FNR fixed asymptotically at level $\alpha$, one can identify the optimal witness function $w$ by maximizing its TNR, defined by

$$\text{TNR}_w := \mathbb{P}_{\boldsymbol{X} \sim p}(T_w(\boldsymbol{X}) \leq z_\alpha), \tag{7}$$

where the probability is evaluated under the human-generated text distribution $p$. Toward that end, we employ MCLT again to derive a closed-form expression of (7).

By definition, we can represent $T_w(\boldsymbol{X})$ by the difference $T_w^{(1)}(\boldsymbol{X}) - T_w^{(2)}(\boldsymbol{X})$ where

$$T_w^{(1)}(\boldsymbol{X}) = \frac{\sum_{t=1}^T [\log w(X_t|X_{<t}) - \mathbb{E}_{\widetilde{X}_t \sim p_t} \log w(\widetilde{X}_t|X_{<t})]}{\sqrt{\sum_t \text{Var}_{\widetilde{X}_t \sim q_t}(\log w(\widetilde{X}_t|X_{<t}))}},$$

$$T_w^{(2)}(\boldsymbol{X}) = \frac{\sum_t [\mathbb{E}_{\widetilde{X}_t \sim q_t} w(\log q_t(\widetilde{X}_t|X_{<t})) - \mathbb{E}_{\widetilde{X}_t \sim p_t} w(\log q_t(\widetilde{X}_t|X_{<t}))]}{\sqrt{\sum_t \text{Var}_{\widetilde{X}_t \sim q_t}(w(\log q_t(\widetilde{X}_t|X_{<t})))}}.$$

Here, to ease notations, $p_t$ and $q_t$ in the expectation and variance are implicitly taken with respect to their conditional distributions $p_t(\bullet|X_{<t})$ and $q_t(\bullet|X_{<t})$ given $X_{<t}$.

Notice that $T_w^{(1)}$ is very similar to $T_w$ — the only difference lies in the centering term: the conditional expectation is taken with respect to $p_t$ instead of $q_t$. Under an equal variance condition where the variance terms in the denominator evaluated at $q_t$ and $p_t$ converge to the same quantity asymptotically, we can invoke the MCLT to prove the asymptotic normality of $T_w^{(1)}(\boldsymbol{X})$ under the human distribution $p$. As a result, $\text{TNR}_w$ can be approximated by

$$\mathbb{P}_{\boldsymbol{X} \sim p}(T_w^{(1)}(\boldsymbol{X}) \leq z_\alpha + T_w^{(2)}(\boldsymbol{X})) \approx \mathbb{E}_{\boldsymbol{X} \sim p}\Phi(z_\alpha + T_w^{(2)}(\boldsymbol{X})). \tag{8}$$

Given the external human-written text dataset $\mathcal{H}$, one can estimate the right-hand-side of (8) by $n^{-1}\sum_{i=1}^n \Phi(z_\alpha + T_w^{(2)}(\boldsymbol{X}^{(i)}))$ and optimize this estimator to compute the witness function $\widehat{w}$. However, the resulting $\widehat{w}$ depends on the choice of $\alpha$, which limits its flexibility. To eliminate this dependence, we derive a lower bound on the TNR in the following theorem.

**Theorem 1** (TNR lower bound). *Under the equal variance condition specified in Appendix D, $TNR_w$ is asymptotically lower bounded by $\min\{\alpha + \Phi'(z_\alpha)T_w^{(2*)}, 1 - \alpha\}$ where $\Phi'$ is the derivative of $\Phi$ and $T_w^{(2*)}$ denotes a population version of $T_w^{(2)}(\boldsymbol{X})$, given by*

$$T_w^{(2*)} = \frac{\sum_t [\mathbb{E}_{X_{<t} \sim p, \widetilde{X}_t \sim q_t} w(\log q_t(\widetilde{X}_t|X_{<t})) - \mathbb{E}_{X_{<t} \sim p, \widetilde{X}_t \sim p_t} w(\log q_t(\widetilde{X}_t|X_{<t}))]}{\sqrt{\sum_t \mathbb{E}_{X_{<t} \sim p} Var_{\widetilde{X}_t \sim q_t}(w(\log q_t(\widetilde{X}_t|X_{<t})))}}. \tag{9}$$

Compared to $T_w^{(2)}(\boldsymbol{X})$, both the numerator and denominator of $T_w^{(2*)}$ are defined by taking expectations with respect to $X_{<t} \sim p$ for each $t$. In the analytical example, the numerator simplifies to (5) after appropriate scaling. According to Theorem 1, optimizing the lower bound is equivalent to maximizing $T_w^{(2*)}$, whose solution is independent of $\alpha$. We also remark that the maximal value $\max_w T_w^{(2*)}$ is similar to certain integral probability metrics (Müller, 1997) such as the maximum mean discrepancy measure widely studied in machine learning (see e.g., Gretton et al., 2012).

Motivated by Theorem 1, we replace the expectations $\mathbb{E}_{X_{<t} \sim p}$ in both the numerator and denominator of $T_w^{(2*)}$ with their empirical average over the dataset $\mathcal{H}$, denote the resulting estimator by $\widehat{T}_w^{(2)}$ and compute $\widehat{w} = \arg\max_{w \in \mathcal{W}} \widehat{T}_w^{(2)}$ over a function class $\mathcal{W}$. Since $w$ is a one-dimensional function over the space of real-valued logits, the optimization is relatively simple.

Specifically, we adopt a linear function class $\mathcal{W} = \{w(z) = \phi(z)^\top \beta : \|\beta\|_2 = 1\}$ for some bounded $d$-dimensional feature mapping $\phi$. In this case, $\widehat{T}_w^{(2)}$ simplifies to $\beta^\top \psi / \sqrt{\beta^\top \Sigma \beta}$ for some $d$-dimensional vector $\psi$ and $d \times d$ semi-definite positive matrix $\Sigma$ (see Appendix C for the detailed derivation). With some calculations, our estimated regression coefficients $\widehat{\beta}$ can be efficiently obtained by solving the linear system $\Sigma\beta = \psi$, leading to $\widehat{w} = \widehat{\beta}^\top \phi$. We set $\phi$ to the B-spline basis function (De Boor, 1978) in our implementation and relegate additional details to Appendix C.

**(d) Extension to black-box settings**. When the target LLM's logits are unavailable, we employ an open-source LLM with a distribution similar to that of the target model to construct the statistical measure in (4). The witness function is then learned in the same manner as described in (c). We empirically evaluate this approach in Section 4.

**(e) Statistical guarantees**. In the following, we establish finite-sample bounds for the FNR, TNR, FPR and TPR of the proposed classifier in the white-box setting; see Figure 4 for a summary of our theories. Recall that $L$ denotes the number of tokens in a passage, $n$ denotes the number of human-authored passages in the training data and $\alpha$ is the target FNR level we wish to control. We also define $w^* = \arg\max_{w \in \mathcal{W}} T_w^{(2*)}$ as the population limit of our estimated witness function $\widehat{w}$. Finally, let $V_L$ denote the ratio

$$\frac{L^{-1}\sum_{t=1}^L \text{Var}_{\widetilde{X}_t \sim q_t}(\widehat{w}(\log q_t(\widetilde{X}_t|X_{<t})))}{L^{-1}\sum_{t=1}^L \text{Var}_{\boldsymbol{X} \sim q}(\widehat{w}(\log q_t(X_t|X_{<t})))}.$$

**Inference**

| | | |
|---|---|---|
| **Ground truth** | TNR
**Theorem 3** | FPR
**Corollary 5** |
| | FNR
**Theorem 2** | TPR
**Corollary 4** |

Figure 4: A summary of our theories.

**Theorem 2** (FNR). *Assume the denominator of $V_L$ is bounded away from zero. Then the expected FNR of our classifier $FNR_{\widehat{w}}$ is upper bounded by $\mathbb{E}(FNR_{\widehat{w}}) \leq \alpha + O\big(L^{-1/2}\log L\big) + O\big(\mathbb{E}|V_L - 1|^{1/3}\big)$, where the expectation on the left-hand-side is taken with respect to $\widehat{w}$.*

**Theorem 3** (TNR). *Under a minimal eigenvalue assumption in Appendix D, with some absolute constant $\gamma > 0$ depending only on the eigenvalue decay, the expected TNR of our classifier $TNR_{\widehat{w}}$ is lower bounded by $\mathbb{E}(TNR_{\widehat{w}}) \geq TNR_{w^*} - O\big(d^\gamma/\sqrt{n}\big)$.*

**Corollary 4** (TPR). *Under the condition in Theorem 2, the expected TPR of our classifier is lower bounded by $\mathbb{E}(TPR_{\widehat{w}}) = 1 - \mathbb{E}(FNR_{\widehat{w}}) \geq 1 - \alpha - O\big(L^{-1/2}\log L\big) - O\big(\mathbb{E}|V_L - 1|^{1/3}\big)$.*

**Corollary 5** (FPR). *Under the condition in Theorem 3, the expected FPR of our classifier is upper bounded by $\mathbb{E}(FPR_{\widehat{w}}) = 1 - \mathbb{E}(TNR_{\widehat{w}}) \leq 1 - TNR_{w^*} + O\big(d^\gamma/\sqrt{n}\big)$.*

We make a few remarks:

- Theorem 2 provides an upper bound on the difference between our classifier's expected FNR and the target level $\alpha$. This excess FNR depends on two factors: (i) the number of tokens $L$, and (ii) the variance ratio $V_L$. As $L$ diverges to infinity and $V_L$ stabilizes to 1, the FNR of our classifier approaches the target level. We remark that assumptions similar to the bounded denominator condition in Theorem 2 are commonly imposed (see e.g., Bolthausen, 1982; Hall & Heyde, 2014).

- Theorem 3 establishes an upper bound on the difference between our classifier's TNR and that of the oracle classifier, which has access to the population-level optimal witness function $w^*$. This difference depends on: (i) the training sample size $n$, and (ii) the dimensionality $d$ of the feature mapping $\phi$. Since $\phi$ is defined over the space of one-dimensional log probabilities of tokens, rather than the token space itself, $d$ is independent of the vocabulary size, and can be treated as fixed. Consequently, as the training data size $n$ grows to infinity, our classifier's TNR converges to that of the oracle classifier. According to Theorem 1, with a sufficiently large $\max_{w \in \mathcal{W}} T_w^{(2*)}$, the expected TNR can reach up to $1 - \alpha$.

- Corollaries 4 and 5 follow directly from Theorems 2 and 3, due to the relationships between TPR and FNR, and between FPR and TNR.

Together, these results suggest that our classifier is expected to achieve a high AUC as the significance level $\alpha$ varies, which we will verify empirically in the next section.

## 4 Experiments

We conduct numerical experiments in both white-box and black-box settings to illustrate the usefulness of AdaDetectGPT. To save space, some implementation details are provided in Appendix C.

**Datasets**. We consider five widely-used datasets for comparing different detectors, including *SQuAD* for Wikipedia-style question answering (Rajpurkar et al., 2016), *WritingPrompts* for story generation (Fan et al., 2018), *XSum* for news summarization (Narayan et al., 2018), *Yelp* for crowd-sourced product reviews (Zhang et al., 2015), and *Essay* for high school and university-level essays (Verma et al., 2024). Following Bao et al. (2024), we randomly sample 500 human-written paragraphs from each dataset and generate an equal number of machine-authored paragraphs by prompting an LLM with the first 120 tokens of the human-written text and requiring it to complete the text with up to 200 tokens. This is a challenging setting where LLM-generated text is mixed with human writing. To evaluate AdaDetectGPT, we compute the AUC on each of the five datasets, with its witness function $\widehat{w}$ trained on two randomly selected datasets that differ from the test dataset.

**Benchmark methods.** In white-box settings, we compare the proposed AdaDetectGPT against **eight** state-of-the-art detectors: *Likelihood*, *Entropy*, *LogRank* (Gehrmann et al., 2019), LogRank Ratio (*LRR*, Su et al., 2023), *DetectGPT* (Mitchell et al., 2023) and its variants Normalized Perturbed log Rank (*NPR*, Su et al., 2023), *Fast-DetectGPT* (Bao et al., 2024), *DNAGPT* (Yang et al., 2024b). In black-box settings, we further compare against *RoBERTaBase* and *RoBERTaLarge* (Solaiman et al., 2019), *Binoculars* (Hans et al., 2024), *RADAR* (Hu et al., 2023), and *BiScope* (Guo et al., 2024a), but omit *DetectGPT*, *NPR* and *DNAGPT* due to their high computational cost. This yields **ten** baseline algorithms. We measure the detection power of each detector using AUC.

**White-box results.** We first consider the *white-box setting* where we employ various source models summarized in Table S4 for text generation, with parameter sizes ranging from 1 to 20 billion. Table 1

Table 1: AUC scores of various detectors in the white-box setting. The "Relative" rows report the percentage improvement of AdaDetectGPT over Fast-DetectGPT.

| Dataset | Method | Source Model | | | | | |
|---|---|---|---|---|---|---|---|
| | | GPT-2 | OPT-2.7 | GPT-Neo | GPT-J | GPT-NeoX | Avg. |
| SQuAD | Likelihood | 0.7043 | 0.6872 | 0.6722 | 0.6414 | 0.5948 | 0.6600 |
| | Entropy | 0.5513 | 0.5289 | 0.5392 | 0.5385 | 0.5402 | 0.5396 |
| | LogRank | 0.7328 | 0.7169 | 0.7047 | 0.6738 | 0.6179 | 0.6892 |
| | LRR | 0.7703 | 0.7623 | 0.7553 | 0.7286 | 0.6632 | 0.5849 |
| | NPR | 0.8783 | 0.8342 | 0.8311 | 0.7330 | 0.6454 | 0.6182 |
| | DNAGPT | 0.8640 | 0.8413 | 0.8106 | 0.7671 | 0.6933 | 0.7953 |
| | DetectGPT | 0.8565 | 0.8167 | 0.8047 | 0.7124 | 0.6380 | 0.6047 |
| | Fast-DetectGPT | 0.9042 | 0.8801 | 0.8731 | 0.8417 | 0.7866 | 0.8571 |
| | AdaDetectGPT | **0.9265** | **0.9058** | **0.9088** | **0.8618** | **0.8292** | **0.8864** |
| | Relative (↑) | 23.2696 | 21.4914 | 28.1306 | 12.6659 | 19.9396 | 20.4909 |
| Writing | Likelihood | 0.7297 | 0.7119 | 0.7032 | 0.6937 | 0.6811 | 0.7039 |
| | Entropy | 0.5154 | 0.5132 | 0.5098 | 0.5126 | 0.5275 | 0.5157 |
| | LogRank | 0.7510 | 0.7332 | 0.7253 | 0.7141 | 0.7029 | 0.7253 |
| | LRR | 0.7801 | 0.7599 | 0.7624 | 0.7489 | 0.7362 | 0.6015 |
| | NPR | 0.8760 | 0.8329 | 0.8373 | 0.8008 | 0.7854 | 0.4942 |
| | DNAGPT | 0.8962 | 0.8503 | 0.8590 | 0.8450 | 0.8199 | 0.8541 |
| | DetectGPT | 0.8540 | 0.8060 | 0.8173 | 0.7681 | 0.7514 | 0.6286 |
| | Fast-DetectGPT | 0.8972 | 0.8891 | 0.8904 | 0.8879 | 0.8782 | 0.8886 |
| | AdaDetectGPT | **0.9352** | **0.9175** | **0.9290** | **0.9190** | **0.9158** | **0.9233** |
| | Relative (↑) | 36.9089 | 25.5699 | 35.2136 | 27.7142 | 30.8963 | 31.1543 |
| XSum | Likelihood | 0.6410 | 0.6264 | 0.6197 | 0.6086 | 0.5906 | 0.6173 |
| | Entropy | 0.5637 | 0.5571 | 0.5864 | 0.5606 | 0.5679 | 0.5671 |
| | LogRank | 0.6623 | 0.6493 | 0.6493 | 0.6310 | 0.6109 | 0.6406 |
| | LRR | 0.6952 | 0.6795 | 0.6967 | 0.6612 | 0.6447 | 0.5361 |
| | NPR | 0.8211 | 0.7713 | 0.8273 | 0.7676 | 0.7290 | 0.6178 |
| | DNAGPT | 0.7531 | 0.7283 | 0.7160 | 0.6939 | 0.6634 | 0.7109 |
| | DetectGPT | 0.8073 | 0.7639 | 0.8244 | 0.7599 | 0.7239 | 0.6110 |
| | Fast-DetectGPT | 0.8293 | 0.8067 | 0.8137 | 0.7926 | 0.7622 | 0.8009 |
| | AdaDetectGPT | **0.8534** | **0.8420** | **0.8532** | **0.8347** | **0.8061** | **0.8379** |
| | Relative (↑) | 14.1250 | 18.2659 | 21.1936 | 20.3301 | 18.4492 | 18.5779 |

reports the AUC scores of various detectors across all combinations of datasets and five source models. It can be seen that AdaDetectGPT achieves the highest AUC across all combinations of datasets and source models, outperforming Fast-DetectGPT – the best baseline method – by 12.5%-37%. We also evaluate AdaDetectGPT on three more advanced open-source LLMs: Qwen2.5 (Bai et al., 2025), Mistral (Jiang et al., 2023), and LLaMA3 (Grattafiori et al., 2024). As shown in Table S7, AdaDetectGPT delivers consistent improvements over Fast-DetectGPT and maintains competitive performance across five datasets, achieving the best results in most cases. These findings highlight the advantage of using an adaptively learned witness function for classification.

In Appendix F.3, we analyze the *computational cost* of AdaDetectGPT. Training the witness function typically requires less than one minute across different sample sizes and dimensions, with memory usage below 0.5 GB. In Appendix F.4, we further conduct a *sensitivity analysis* to investigate the sensitivity of AdaDetectGPT's AUC score to various factors that may affect the estimation of the witness function. Our results show that AdaDetectGPT is generally robust to the size of training data, the number of B-spline features, and the distributional shift between training and test data, consistently maintaining superior performance over baselines.

**Black-box results**. We next consider the *black-box setting*, where the task is to detect text generated by three widely used advanced LLMs: GPT-4o (Hurst et al., 2024), Claude-3.5-Haiku (Anthropic, 2024), and Gemini-2.5-Flash (Comanici et al., 2025). In this setting, token-level log-probabilities are not publicly accessible. To implement Fast-DetectGPT and AdaDetectGPT, we use `google/gemma-2-9b` and `google/gemma-2-9b-it` (Team et al., 2024) as the sampling and scoring models, respectively, to construct the classification statistics described in Section 2. The results are reported in Tables 2 and S8. Overall, Fast-DetectGPT remains the strongest baseline, although it occasionally

Table 2: AUC scores of various detectors to detect text generated by GPT-4o and Claude-3.5 across datasets.

| Method | GPT-4o | | | | | Claude-3.5 | | | | |
|---|---|---|---|---|---|---|---|---|---|---|
| | XSum | Writing | Yelp | Essay | Avg. | XSum | Writing | Yelp | Essay | Avg. |
| RoBERTaBase | 0.5141 | 0.5352 | 0.6029 | 0.5739 | 0.5655 | 0.5206 | 0.5386 | 0.5630 | 0.5593 | 0.5566 |
| RoBERTaLarge | 0.5074 | 0.5827 | 0.5027 | 0.6575 | 0.5626 | 0.5462 | 0.6149 | 0.5105 | 0.6063 | 0.5695 |
| Likelihood | 0.5194 | 0.7661 | 0.8425 | 0.7849 | 0.7282 | 0.6780 | 0.7502 | 0.7134 | 0.6160 | 0.6894 |
| Entropy | 0.5397 | 0.7021 | 0.7291 | 0.6951 | 0.6665 | 0.5935 | 0.6594 | 0.6625 | 0.5742 | 0.6142 |
| LogRank | 0.5123 | 0.7478 | 0.8259 | 0.7786 | 0.7161 | 0.5109 | 0.6653 | 0.7244 | 0.7111 | 0.6529 |
| LRR | 0.5116 | 0.6099 | 0.6828 | 0.6930 | 0.6243 | 0.5268 | 0.5560 | 0.5527 | 0.6535 | 0.5723 |
| Binoculars | 0.9022 | 0.9572 | 0.9840 | 0.9777 | 0.9552 | 0.9012 | 0.9393 | 0.9752 | 0.9603 | 0.9440 |
| RADAR | **0.9580** | 0.8046 | 0.8558 | 0.8394 | 0.8644 | **0.9187** | 0.7264 | 0.8424 | 0.9152 | 0.8507 |
| BiScope | 0.8333 | 0.8733 | 0.9700 | 0.9600 | 0.9092 | 0.8533 | 0.8800 | 0.8800 | 0.9567 | 0.8925 |
| Fast-DetectGPT | 0.9048 | 0.9588 | **0.9847** | 0.9800 | 0.9571 | 0.9019 | 0.9361 | **0.9768** | 0.9608 | 0.9439 |
| AdaDetectGPT | 0.9072 | **0.9611** | 0.9832 | **0.9841** | **0.9589** | 0.9176 | **0.9400** | 0.9728 | **0.9610** | **0.9478** |
| Relative ($\uparrow$) | 2.4288 | 5.6095 | — | 20.4444 | 4.2454 | 16.0326 | 6.0543 | — | 0.3405 | 6.9929 |

Table 3: Detection of LLM-generated text under two adversarial attacks in black-box settings.

| DetectGPT | Paraphrasing | | | | Decoherence | | | |
|---|---|---|---|---|---|---|---|---|
| | Xsum | Writing | PubMed | Avg. | Xsum | Writing | PubMed | Avg. |
| Fast (GPT-J/GPT-2) | 0.9178 | **0.9137** | 0.7944 | 0.8753 | 0.7884 | 0.9595 | 0.7870 | 0.8449 |
| Ada (GPT-J/GPT-2) | **0.9225** | 0.9121 | **0.8029** | **0.8792** | **0.8765** | **0.9597** | **0.8284** | **0.8882** |
| Fast (GPT-J/Neo-2.7) | 0.9602 | **0.9185** | 0.7310 | 0.8699 | 0.8579 | 0.9701 | 0.7609 | 0.8630 |
| Ada (GPT-J/Neo-2.7) | **0.9623** | 0.9181 | **0.7587** | **0.8797** | **0.9230** | **0.9704** | **0.8124** | **0.9019** |
| Fast (GPT-J/GPT-J) | 0.9537 | **0.9458** | 0.7041 | 0.8679 | 0.8836 | **0.9869** | 0.7550 | 0.8752 |
| Ada (GPT-J/GPT-J) | **0.9587** | 0.9449 | **0.7308** | **0.8781** | **0.9336** | 0.9864 | **0.8008** | **0.9070** |

underperforms Binoculars or RADAR. Nonetheless, AdaDetectGPT consistently improves upon Fast-DetectGPT across various datasets, with gains on Essay reaching up to 37.8%.

Additionally, we evaluate AdaDetectGPT's robustness to two adversarial attacks, paraphrasing and decoherence, in the black-box setting. As shown in Table 3, AdaDetectGPT demonstrates greater resilience than Fast-DetectGPT to adversarially perturbed texts. The improvement reaches up to 10% for paraphrasing and up to 85% for decoherence. Finally, we employ the same five LLMs from Table 1 to compare Fast-DetectGPT and AdaDetectGPT in black-box settings. Following Bao et al. (2024), we use GPT-J as the source LLM for detecting each of the remaining four target LLMs. Due to space constraints, the results are presented in Table S11 of Appendix F.5, where AdaDetectGPT uniformly outperforms Fast-DetectGPT in all cases, with improvements of up to 29%.

# 5 Conclusion

We propose AdaDetectGPT, an adaptive LLM detector that learns a witness function $w$ to boost the performance of existing logits-based detectors. A natural approach to learning $w$ is to maximize the TNR of the resulting detector for a fixed FNR level $\alpha$. Our proposal has two novelties. First, by connecting Fast-DetectGPT's statistic to martingale theory and applying the MCLT, we obtain a closed-form expression for the classification threshold that achieves FNR control at level $\alpha$. Second, the TNR is a highly complex function of $\alpha$ and $w$, which makes the learned witness function $\alpha$-dependent — that is, it maximizes the TNR at a particular FNR level but does not guarantee optimality at other FNR levels. To address this, we derive a lower bound on the TNR and propose to learn $w$ by maximizing this lower bound. Our lower bound separates the effects of $\alpha$ and $w$: the witness function $w$ affects the lower bound only through $T_w^{(2)*}$, which is independent of $\alpha$. Consequently, the witness function that maximizes this lower bound simultaneously maximizes it across all FNR levels.

In our implementation, we opted to learn the witness function via a B-spline basis due to the straightforward nature of the optimization (finding the optimal witness function boils down to solving a system of linear equations) and its favorable theoretical properties (the estimation error can attain Stone's optimal convergence rate, Stone, 1982, see Appendix E.6 for more details). The number of basis functions can be selected in a data driven way via Lepski's method (Lepski & Spokoiny, 1997).

## Acknowledgments

Chengchun Shi's and Jin Zhu's research was partially supported by the EPSRC grant EP/W014971/1. Hongyi Zhou's and Ying Yang's research was partially supported by NSFC 12271286 & 11931001. Hongyi Zhou's research was also partially supported by the China Scholarship Council.

The authors thank the anonymous referees and the area chair for their insightful and constructive comments, which have led to a significantly improved version of the paper.

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

## A  Additional related works: Watermarking-based detection

Watermarking embeds subtle signals into LLM-generated text to distinguish it from human-written text (see Ji et al., 2025, Section 4.2, for a recent review of LLM watermarking). Aaronson & Kirchner (2023) propose a watermarking technique based on Gumbel sampling. Follow-up works have focused on preserving text quality during watermarking (Christ et al., 2024; Dathathri et al., 2024; Giboulot & Furon, 2024; Liu & Bu, 2024; Wouters, 2024; Wu et al., 2024), enhancing watermark detection (Dathathri et al., 2024; Huo et al., 2024; Cai et al., 2025) and maintaining robustness against adversarial edits (Golowich & Moitra, 2024).

Our work is related to a line of research that frames watermark detection as a statistical hypothesis testing problem (see e.g., Kirchenbauer et al., 2023; Hu et al., 2024; Kuditipudi et al., 2024; Zhao et al., 2024; Li et al., 2025a,b; Chen et al., 2025b). Under this framework, rejection of the null hypothesis (that no watermark is present) provides statistical evidence that the text was likely generated by an LLM.

## B  Details on the analytic example in Section 3

In this section, we provide rigorous discussion about the analytic example presented in Section 3. Noted that

$$
\begin{aligned}
\mathbb{E}_{\widetilde{X}_t \sim q, X_{<t} \sim p}\left\{w(\log q(X_t)\right\} &= q(1)w(\log q(1)) + q(0)w(\log q(0)), \\
\mathbb{E}_{X_{<t+1} \sim p}\left\{w(\log q(X_t))\right\} &= p_t(1)w(\log q(1)) + p_t(0)w(\log q(0)).
\end{aligned}
$$

It follows that

$$
\begin{aligned}
&\mathbb{E}_{\widetilde{X}_t \sim q, X_{<t} \sim p}\left\{w(\log q(X_t)\right\} - \mathbb{E}_{X_{<t+1} \sim p}\left\{w(\log q(X_t))\right\} \\
&= (q(1) - p_t(1))\left[w(\log q(1)) - w(\log q(0))\right].
\end{aligned}
$$

If $w$ is an identity function, i.e., $w(x) = x$, then the statistics (5) becomes

$$
\frac{1}{L}\log\left(\frac{q(1)}{q(0)}\right)\sum_{t=1}^{L}(q(1) - p_t(1)).
$$

In this case, (5) converges to zero as $q \to 1/2$ regardless the distribution of $p_t$. However, if we consider adaptive witness function, the statistics in (5) becomes

$$
\frac{1}{L}\left[w(\log q(1)) - w(\log q(0))\right]\sum_{t=1}^{L}(q(1) - p_t(1)).
$$

When $q(1) \neq 1/2$ (without generality, we assume $q(1) = 1 - q(0) > 1/2$), there always exists a witness function $w(z) = \mathbb{I}\left\{z > \frac{\log q(1) + \log q(0)}{2}\right\}$ such that (5) becomes

$$
\begin{aligned}
&\frac{1}{L}\left[\mathbb{I}\{\log q(1) > \log q(0)\} - \mathbb{I}\{\log q(0) > \log q(1)\}\right]\sum_{t=1}^{L}(q(1) - p_t(1)) \\
&= \frac{1}{L}\sum_{t=1}^{L}(q(1) - p_t(1)) = q(1) - \frac{1}{L}\sum_{t=1}^{L}p_t(1),
\end{aligned}
$$

which is independent of the log ratio.

## C  Experiment details

**Details for witness function estimation**. In this part, we illustrate how we fetch external text datasets for training witness function in our experiments. As mentioned in the main text, when testing the performance of AdaDetectGPT on one dataset (e.g., XSum), we randomly select two other datasets (e.g., SQuAD and WritingPrompt) for training the witness function. This ensures the data for testing would not be included for training.

Recall that the population version of objective function for estimating $w$ is

$$\frac{\sum_t [\mathbb{E}_{X_{<t}\sim p, \widetilde{X}_t \sim q_t} w(\log q_t(\widetilde{X}_t|X_{<t}))] - \sum_t [\mathbb{E}_{X_{<t}\sim p, \widetilde{X}_t \sim p_t} w(\log q_t(\widetilde{X}_t|X_{<t}))]}{\sqrt{\sum_t \mathbb{E}_{X_{<t}\sim p} \mathrm{Var}_{\widetilde{X}_t \sim q_t}(w(\log q_t(\widetilde{X}_t|X_{<t})))}}. \tag{10}$$

In the implementation, we made three modifications to this objective function to facilitate the computation, accommodate black-box settings with unavailable logits and handle prompts that are not explicitly included in the text:

1. We replace the expectation $\mathbb{E}_{X_{<t}\sim p}$ in the numerator of (10) with its empirical average over the human-authored passages $\{\boldsymbol{X}^{(i)}\}_i$. This leads to the following objective function:

$$\frac{\sum_i \sum_t \mathbb{E}_{\widetilde{X}_t \sim q_t}[w(\log q_t(\widetilde{X}_t|X_{<t}^{(i)}))] - \sum_i \sum_t [w(\log q_t(X_t^{(i)}|X_{<t}^{(i)}))]}{n\sqrt{\sum_t \mathbb{E}_{X_{<t}\sim p} \mathrm{Var}_{\widetilde{X}_t \sim q_t}(w(\log q_t(\widetilde{X}_t|X_{<t})))}}. \tag{11}$$

2. Taking the expectation $\mathbb{E}_{\widetilde{X}_t \sim q_t}$ and the variance $\mathrm{Var}_{\widetilde{X}_t \sim q_t}$ is time-consuming, as these operations need to be repeated $L$ times – once at every token position $t$. To address this, we approximate $X_{<t}^{(i)}$ in the first term of the numerator of (11) by $\widetilde{X}_t^{(i)}$, sampled from the LLM distribution $q_t$. This, in turn, allows us to approximate the expectation by

$$\frac{\sum_i \sum_t [w(\log q_t(\widetilde{X}_t^{(i)}|\widetilde{X}_{<t}^{(i)}))] - \sum_i \sum_t [w(\log q_t(X_t^{(i)}|X_{<t}^{(i)}))]}{n\sqrt{\sum_t \mathbb{E}_{X_{<t}\sim p} \mathrm{Var}_{\widetilde{X}_t \sim q_t}(w(\log q_t(\widetilde{X}_t|X_{<t})))}},$$

where $\widetilde{\boldsymbol{X}}^{(i)}$ denotes the $i$th passage generated from $q$ by prompting the LLM to rewrite $\boldsymbol{X}^{(i)}$. As for the variance operator, we similarly approximate $\mathbb{E}_{X_{<t}\sim p}$ in the denominator by $\mathbb{E}_{X_{<t}\sim q}$. This allows us to upper bound the denominator by $n\sqrt{\sum_t \mathrm{Var}(w(\log q_t(\widetilde{X}_t|\widetilde{X}_{<t})))}$, leading to the following lower bound of the objective function,

$$\frac{\sum_i \sum_t [w(\log q_t(\widetilde{X}_t^{(i)}|\widetilde{X}_{<t}^{(i)}))] - \sum_i \sum_t [w(\log q_t(X_t^{(i)}|X_{<t}^{(i)}))]}{n\sqrt{\sum_t \mathrm{Var}(w(\log q_t(\widetilde{X}_t|\widetilde{X}_{<t})))}},$$

where the variance in the denominator can be estimated by the sampling variance estimator

$$\widehat{\mathrm{Var}}(w(\log q_t(\widetilde{X}_t|\widetilde{X}_{<t}))) = \frac{1}{n-1}\sum_{i=1}^n \left[w(\log q_t(\widetilde{X}_t^{(i)}|\widetilde{X}_{<t}^{(i)})) - \frac{1}{n}\sum_{j=1}^n w(\log q_t(\widetilde{X}_t^{(j)}|\widetilde{X}_{<t}^{(j)}))\right]^2.$$

3. To further simplify the objective function, we interchange the order of $\sum_t$ and $\widehat{\mathrm{Var}}$ in the denominator, leading to

$$\frac{\sum_i \sum_t [w(\log q_t(\widetilde{X}_t^{(i)}|\widetilde{X}_{<t}^{(i)}))] - \sum_i \sum_t [w(\log q_t(X_t^{(i)}|X_{<t}^{(i)}))]}{n\sqrt{\widehat{\mathrm{Var}}(\sum_t w(\log q_t(\widetilde{X}_t|\widetilde{X}_{<t})))}}. \tag{12}$$

Additionally, since the numerator incorporates both human- and LLM-authored text, we refine the denominator of Equation (12) by replacing the sampling variance estimator with a simple average of the estimators computed using human- and LLM-written text

$$\frac{1}{2}\widehat{\mathrm{Var}}\left(\sum_t w(\log q_t(\widetilde{X}_t|\widetilde{X}_{<t}))\right) + \frac{1}{2}\widehat{\mathrm{Var}}\left(\sum_t w(\log q_t(X_t|X_{<t}))\right).$$

This yields our final objective function, in the form of a two-sample $t$-test statistic,

$$\frac{\sum_i \sum_t [w(\log q_t(\widetilde{X}_t^{(i)}|\widetilde{X}_{<t}^{(i)}))] - \sum_i \sum_t [w(\log q_t(X_t^{(i)}|X_{<t}^{(i)}))]}{n\sqrt{0.5\widehat{\mathrm{Var}}(\sum_t w(\log q_t(\widetilde{X}_t|\widetilde{X}_{<t}))) + 0.5\widehat{\mathrm{Var}}(\sum_t w(\log q_t(X_t|X_{<t})))}}. \tag{13}$$

Compared to its population-level version (10), (13) is more suitable for black-box settings. In such settings, the distribution $q$ is unknown, making the expectation and variance over $q$ in (10) infeasible to compute. In contrast, (13) relies on text generated by the LLM. Therefore, even without access to $q$, we can still prompt the target LLM to produce $\widetilde{\boldsymbol{X}}$. Likewise, for detecting text generated under specific prompts, we can incorporate these prompts into the rewriting process to produce $\widetilde{\boldsymbol{X}}$.

We next discuss the computation of $\widehat{w}$ that maximizes (13). To ease notation, we denoted $\log q(X_t^{(i)}|X_{<t}^{(i)})$ as $z_{it}^{(h)}$. Similarly, for the log-probabilities computed from machine-generated text, we define them as $z_{it}^{(m)}$s. Using these notations, (13) can be represented by

$$\frac{1}{\sqrt{\mathrm{Var}(\sum_t w(z_t^{(m)})) + \mathrm{Var}(\sum_t w(z_t^{(h)}))}} \left( \sum_{i=1}^n \frac{1}{L} \sum_{t=1}^L w(z_{it}^{(m)}) - \sum_{i=1}^n \frac{1}{L} \sum_{t=1}^L w(z_{it}^{(h)}) \right), \quad (14)$$

up to some proportional constant.

Recall that we restrict the witness function to take a linear form of $w(z) = \phi(z)^\top \beta$, where $\phi(z)$ denotes the B-spline basis function (De Boor, 1978) and $\beta$ denotes the regression coefficients. Then numerator of (14) then becomes

$$\sum_{i=1}^n \frac{1}{L} \sum_{t=1}^L \phi(z_{it}^{(m)})^\top \beta - \sum_{i=1}^n \frac{1}{L} \sum_{t=1}^L \phi(z_{it}^{(h)})^\top \beta,$$

whereas the denominator becomes $\sqrt{\beta^\top (\widehat{\Sigma}^{(m)} + \widehat{\Sigma}^{(h)})\beta}$, where $\widehat{\Sigma}^{(h)} = \sum_{i=1}^n \widehat{\Sigma}_i^{(h)}$,

$$\widehat{\Sigma}_i^{(h)} = \frac{1}{L} (\mathbf{Z}_i^{(h)})^\top \mathbf{Z}_i^{(h)} - \widehat{\mu}_i^{(h)} (\widehat{\mu}_i^{(h)})^\top,$$

$$\mathbf{Z}_i^{(h)} = \left( \phi(z_{i1}^{(h)}), \ldots, \phi(z_{iL}^{(h)}) \right)^\top,$$

$$\widehat{\mu}_i^{(h)} = \frac{1}{L} \sum_{t=1}^L \phi(z_{it}^{(h)})^\top,$$

and $\widehat{\Sigma}^{(m)}$ can be similarly defined. Consequently, the objective function can be rewritten as:

$$\frac{\left[ \sum_{i=1}^n \frac{1}{L} \sum_{t=1}^L \phi(z_{it}^{(m)})^\top - \sum_{i=1}^n \frac{1}{L} \sum_{t=1}^L \phi(z_{it}^{(h)})^\top \right] \beta}{\sqrt{\beta^\top (\widehat{\Sigma}^{(h)} + \widehat{\Sigma}^{(m)})\beta}}$$

$$= \left[ \sum_{i=1}^n \frac{1}{L} \sum_{t=1}^L \phi(z_{it}^{(m)})^\top - \sum_{i=1}^n \frac{1}{L} \sum_{t=1}^L \phi(z_{it}^{(h)})^\top \right] \beta \times \frac{1}{\|(\widehat{\Sigma}^{(h)} + \widehat{\Sigma}^{(m)})^{1/2}\beta\|_2}$$

$$= \left[ \sum_{i=1}^n \frac{1}{L} \sum_{t=1}^L \phi(z_{it}^{(m)})^\top - \sum_{i=1}^n \frac{1}{L} \sum_{t=1}^L \phi(z_{it}^{(h)})^\top \right] \times (\widehat{\Sigma}^{(h)} + \widehat{\Sigma}^{(m)})^{-\frac{1}{2}} \alpha,$$

where $\alpha = (\widehat{\Sigma}^{(h)} + \widehat{\Sigma}^{(m)})^{1/2}\beta / \|(\widehat{\Sigma}^{(h)} + \widehat{\Sigma}^{(m)})^{1/2}\beta\|_2$ whose $\ell_2$ norm equals 1. It is immediate to see that the argmax $\widehat{\alpha}$ has a closed-form expression,

$$\widehat{\alpha} = \frac{\widetilde{\alpha}}{\|\widetilde{\alpha}\|_2}$$

where

$$\widetilde{\alpha} = (\widehat{\Sigma}^{(h)} + \widehat{\Sigma}^{(m)})^{-\frac{1}{2}} \left[ \sum_{i=1}^n \frac{1}{L} \sum_{t=1}^L \phi(z_{it}^{(m)}) - \sum_{i=1}^n \frac{1}{L} \sum_{t=1}^L \phi(z_{it}^{(h)}) \right]. \quad (15)$$

This leads to

$$\widehat{\beta} = (\widehat{\Sigma}^{(h)} + \widehat{\Sigma}^{(m)})^{-1/2} \widehat{\alpha}.$$

and $\widehat{w}(z) = \phi(z)^{\top}\widehat{\beta}$.

**Pre-trained language models.** We assess the performance of our method using text generated from various pre-trained language models outlined in Table S4. Following the setting in Bao et al. (2024), for the models with over 6B parameters, we employ half-precision (`torch.float16`), otherwise, we use full-precision (`torch.float32`).

Table S4: Description of the source models that is used to produce machine-generated text. [†]: we present the address of models in https://huggingface.co/.

| Name | Model[†] | Scale (Billion) |
|---|---|---|
| GPT-2 (Radford et al., 2019) | openai-community/gpt2-xl | 1.5B |
| GPT-Neo (Black et al., 2021) | EleutherAI/gpt-neo-2.7B | 2.7B |
| OPT-2.7 (Zhang et al., 2022) | facebook/opt-2.7b | 2.7B |
| GPT-J (Wang & Komatsuzaki, 2021) | EleutherAI/gpt-j-6B | 6B |
| Qwen2.5 (Yang et al., 2024a) | Qwen/Qwen2.5-7B | 7B |
| Mistral (Jiang et al., 2023) | mistralai/Mistral-7B-v0.3 | 7B |
| Llama (AIMeta, 2024) | meta-llama/Meta-Llama-3-8B | 8B |
| GPT-NeoX (Black et al., 2022) | EleutherAI/gpt-neox-20b | 20B |

**Setup of the closed-source LLMs**. For the gpt-4o, the version is set as `gpt-4o-2024-08-06`. The generation process by sending the following messages to the service. Message for XSum and Writing is the same as that described in Section C.2 in Bao et al. (2024). We describe that for Yelp and Essay are:

```
[
  {'role': 'system', 'content': 'You are a Review writer on Yelp.'},
  {'role': 'user', 'content': 'Please write an article with about 150
     ↪ words starting exactly with: <prefix>'},
]
```

and

```
[
  {'role': 'system', 'content': 'You are a student of high school and
     ↪ university level. And now, you are an Essay writer.'},
  {'role': 'user', 'content': 'Please write an essay with about 200
     ↪ words starting exactly with: <prefix>'},
]
```

respectively.

For Claude-3.5-Haiku, the system instruction was set analogously (e.g., Yelp review or essay writer), while the user role contained the corresponding content prompt.

For Gemini, the instruction was fed into the `system_instruction` parameter, with a value identical to the concatenation of the system content and the user content used for GPT-4o.

For all closed-source models, the temperature parameter is set to 0.8 to encourage the generated text to be creatively diverse and less predictable.

**Setting on experiments with adversarial attacks**. In Table 3, we have conducted experiments to evaluate the robustness of AdaDetectGPT against 2 adversarial attacks: (i) paraphrasing, where an LLM is instructed to rephrase human-written text, and (ii) decoherence, where the coherence LLM-generated text is intentionally reduced to avoid detection. These experiments were carried out across 3 datasets and 3 types of sampling and scoring models setup, resulting in a total of 18 settings. Both adversarial attacks were implemented following Bao et al. (2024).

**Implementations of baselines**. For the baselines considered in our experiments, we use the existing implementation provided in https://github.com/baoguangsheng/fast-detect-gpt, which is distributed in the MIT License. We run DetectGPT and NPR with default 100 perturbations with

the T5 model (Raffel et al., 2020) and run DNA-GPT with a truncate-ratio of 0.5 and 10 prefix completions per passage.

**Evaluation Metric**. We measure the detection accuracy by AUC (short for "area under the curve"). AUC ranges from 0.0 to 1.0, an AUC of 1.0 indicates a perfect classifier and vice versa. The relative improvement of AdaDetectGPT over FastDetectGPT is calculated by $\frac{\text{AdaDetectGPT}-\text{FastDetectGPT}}{1.0-\text{FastDetectGPT}}$, which represents how much improvement has been made relative to the maximum possible improvement for FastDetectGPT.

**Hardware details**. Most of experiments are run on a Tesla A100 GPU (40GB) with 10 vCPU Intel Xeon Processor and 72GB RAM. For the experiments where the source model is GPT-NeoX, we run on a H20-NVLink (96GB) GPU with 20 vCPU Intel(R) Xeon(R) Platinum and 200GB RAM.

# D Technical assumptions

In this section, we list the assumptions required for the theorems presented in Section 3 to hold, and discuss when they are expected to hold and how they may be relaxed.

**Assumption 1** (Margin). *With $T_w(\bullet)$ defined as in (4) and $w^*(\bullet)$ defined as the optimizer of (9), for any $\alpha \in (0,1)$ there are constants $\delta_\alpha, C_\alpha$ depending only on $\alpha$ such that for any $x \leq \delta_\alpha$ it holds that $\mathbb{P}_{\boldsymbol{X} \sim p}(|T_{w^*}(\boldsymbol{X}) - z_\alpha| \leq x) \leq C_\alpha x$.*

We also require the following technical conditions hold in order to obtain TNR lower bound and FNR control (Theorem 1 and Theorem 2).

**Assumption 2** (Minimum eigenvalue). *For each $t = 1, \ldots, L$ introduce the quantities*

$$\mu_t^{(1)} = \mathbb{E}_{X_{<t} \sim p} \mathbb{E}_{\widetilde{X}_t \sim q_t} \phi(\log q_t(\widetilde{X}_t \mid X_{<t})),$$

$$\Sigma_t = \mathbb{E}_{X_{<t} \sim p} \mathbb{E}_{\widetilde{X}_t \sim q_t} \phi(\log q_t(\widetilde{X}_t \mid X_{<t})) \phi(\log q_t(\widetilde{X}_t \mid X_{<t}))^\top - \mu_t^{(1)}(\mu_t^{(1)})^\top.$$

*There are absolute constant $C > 0$ and $\gamma > 0$ such that $\lambda_{min}(\Sigma_t) \geq Cd^{-\gamma}$ for all $t$.*

**Assumption 3** (Equal variance). *For any non-constant witness function $w$, define*

$$\sigma_{q,L}^2 := \frac{1}{L} \sum_{t=1}^{L} \text{Var}_{\widetilde{X}_t \sim q_t} \left( w(\log q_t(\widetilde{X}_t | \widetilde{X}_{<t})) \right),$$

$$\sigma_{p,L}^2 := \frac{1}{L} \sum_{t=1}^{L} \text{Var}_{\widetilde{X}_t \sim p_t} \left( w(\log q_t(\widetilde{X}_t | \widetilde{X}_{<t})) \right).$$

$\sigma_{q,L}^2, \sigma_{p,L}^2$ *are lower bounded by some constant $\sigma_w^2 > 0$ almost surely. Moreover, $\sigma_{q,L} - \sigma_{p,L} \to 0$ in probability as $L \to \infty$.*

**Assumption 4.** *For any witness function $w$, define*

$$\bar{\sigma}_{q,L}^2 = \frac{1}{L} \sum_{t=1}^{L} \text{Var}_{\boldsymbol{X} \sim q} \left( w(\log q_t(\widetilde{X}_t | \widetilde{X}_{<t})) \right),$$

$$\bar{\sigma}_{p,L}^2 = \frac{1}{L} \sum_{t=1}^{L} \text{Var}_{\boldsymbol{X} \sim p} \left( w(\log q_t(\widetilde{X}_t | \widetilde{X}_{<t})) \right).$$

*If $\boldsymbol{X} \sim q$, then $\bar{\sigma}_{q,L}^2 / \sigma_{q,L}^2 \to 1$ in probability. If $\boldsymbol{X} \sim p$, then $\bar{\sigma}_{p,L}^2 / \sigma_{p,L}^2 \to 1$ in probability.*

Conditions similar to Assumption 1 are commonly assumed; see, for instance, Audibert & Tsybakov (2007); Qian & Murphy (2011); Luedtke & Van Der Laan (2016); Shi et al. (2020a,b, 2022). Assumption 2 is mild, since the constant $\gamma$ is allowed to be arbitrarily large. Assumption 3 basically requires the conditional variance of logits be asymptotically equivalent for human-authored and machine-generated passages. This assumption is not overly restrictive, as the variance discrepancy between the two types of passages is relatively small in our dataset (see Table S5, where the ratio f the variances are very closed to 1). Assumption 4 is commonly assumed in martingale central limit theorem literature, see e.g. Bolthausen (1982); Hall & Heyde (2014). Our empirical results

further support the validity of Assumption 4: the sample mean of the ratio remains nearly constant as a function of $L$, while its variance (in parentheses) approaches zero as $L \to \infty$ across all three datasets (see Table S5), suggesting that this condition is practical. Furthermore, two commonly employed hypothesis tests — the Kolmogorov–Smirnov (KS) test and the Shapiro–Wilk (SW) test — are conducted to evaluate whether the proposed statistic follows a normal distribution. As shown in Table S6, almost all $p$-values exceed 0.1 (most by a large margin), indicating that our test statistic passes normality tests in most cases. These test results also provide strong empirical support for the validity of MCLT regularity conditions.

Table S5: Sample mean and variance (in parentheses) of the ratio evaluated on 3 datasets as $L$ increases.

| $L$ | 100 | 150 | 200 | 250 | 300 | 350 |
|---|---|---|---|---|---|---|
| XSum | 1.09(0.12) | 1.07(0.09) | 1.06(0.08) | 1.04(0.07) | 1.01(0.06) | 0.99(0.05) |
| SQuAD | 1.03(0.10) | 1.02(0.07) | 1.02(0.06) | 1.02(0.05) | 1.01(0.05) | 1.03(0.04) |
| Writing | 1.10(0.06) | 1.09(0.05) | 1.09(0.04) | 1.08(0.03) | 1.07(0.03) | 1.05(0.03) |

Table S6: $p$-values of KS and SW tests across 3 datasets and 2 source LLMs.

| LLM | Test | XSum | SQuAD | Writing |
|---|---|---|---|---|
| GPT-Neo | KS | 0.72 | 0.54 | 0.18 |
| GPT-Neo | SW | 0.50 | 0.65 | 0.89 |
| GPT-2 | KS | 0.10 | 0.52 | 0.28 |
| GPT-2 | SW | 0.37 | 0.026 | 0.14 |

# E   Proofs

## E.1   Notations

Throughout the proofs we will make use of the following notation. We will denote absolute constants by $\kappa_1, \kappa_2, \cdots$. For a sequence of random variables $\{X_n \mid n \geq 1\}$ with distribution functions $\{F_{X_n} | n \geq 1\}$ and some (possibly degenerate) random variable $Y$ with distribution function $F_Y$ we write $X_n \overset{p}{\to} Y$ as $n \to \infty$ if $\lim_{n \to \infty} \mathbb{P}(|X_n - Y| > \delta) = 0$ for all $\delta > 0$, and $X_n \overset{d}{\to} Z$ if $\lim_{n \to \infty} F_{X_n}(x) = F_Y(x)$ at every continuity point of $F_Y(\cdot)$. For a vector $x = (x_1, \ldots, x_d)^\top \in \mathbb{R}^d$ we write $\|x\|_p = (\sum_{j=1}^d x_j^p)^{1/p}$ with $0 < p < \infty$ for its $\ell_p$-norm.

## E.2   Preparatory results

We introduce three auxiliary results in this section. Theorem S6 presents a concentration inequality that is critical to establishing the learning guarantees of AdaDetectGPT in Theorem 3. Theorem S7 formally states the MCLT. Finally, Lemma S1 can be viewed as a non-asymptotic version of Theorem S7 which provides an explicit error bound on the accuracy of the normal approximation in the MCLT.

**Theorem S6** (Bounded differences inequality). *Let $\mathcal{X}$ be a measurable space. A function $f : \mathcal{X}^n \to \mathbb{R}$ has the bounded difference property for some constants $c_1, \ldots, c_n$, i.e., for each $i = 1, \ldots, n$,*

$$\sup_{\substack{x_1, \ldots, x_n \\ x_i' \in \mathcal{X}}} |f(x_1, \ldots, x_{i-1}, x_i, x_{i+1}, x_n) - f(x_1, \ldots, x_{i-1}, x_i', x_{i+1}, \ldots x_n)| \leq c_i. \quad (16)$$

*Then, if $X_1, \ldots, X_n$ is a sequence of identically distributed random variables and (16) holds, putting $Z = f(X_1, \ldots, X_n)$ and $\nu = \frac{1}{4} \sum_{i=1}^n c_i^2$ for any $t > 0$, it holds that*

$$\mathbb{P}(Z - \mathbb{E}(Z) > t) \leq e^{-t^2/(2\nu)}.$$

*Proof of Theorem S6.* See Section 2 in Wainwright (2019). □

**Theorem S7** (Martingale central limit theorem). *Let $\{M_{n,i} \mid 1 \le i \le k_n, n \ge 1\}$ be a zero mean square integrable martingale array with respect to the filtrations $\{\mathcal{F}_{n,i} \mid 1 \le i \le k_n, n \ge 1\}$ having increments $X_{n,i} = M_{n,i} - M_{n,i-1}$. If the following conditions hold*

**C1:** $\sum_{i=1}^{k_n} \mathbb{E}\left[X_{n,i}\mathbf{1}_{\{|X_{n,i}|>\delta\}} \mid \mathcal{F}_{n,i-1}\right] \xrightarrow{p} 0$ *as $n \to \infty$ for all $\delta > 0$*

**C2:** $\sum_{i=1}^{k_n} \mathbb{E}\left[X_{n,i}^2 \mid \mathcal{F}_{n,i-1}\right] \xrightarrow{p} \sigma^2$ *as $n \to \infty$*

**C3:** *the $\sigma$-fields are nested: $\mathcal{F}_{n,i} \subseteq \mathcal{F}_{n+1,i}$ for $1 \le i \le k_n$ and $n \ge 1$*

*then $M_{n,k_n} \xrightarrow{d} Z$ as $n \to \infty$, where $Z \sim \mathcal{N}\left(0, \sigma^2\right)$.*

*Proof.* See Corollary 3.1 in Hall & Heyde (2014) and Theorem 2 in Bolthausen (1982). □

**Lemma S1** (Convergence rates in MCLT). *Let $\boldsymbol{X} = (X_1, \dots X_n)$ be sequences of real valued random variables satisfying for all $1 \le t \le n$,*

$$\mathbb{E}(X_t|X_{<t}) = 0 \quad \text{almost surely.}$$

*Let $\sigma_t^2 = \mathbb{E}(X_t^2|X_{<t})$, $\bar{\sigma}_t^2 = \mathbb{E}(X_t^2)$, $s_n^2 = \sum_{t=1}^n \bar{\sigma}_t^2$ and $V_n^2 = \sum_{t=1}^n \sigma_t^2/s_n^2$. Suppose $|X_n|$ is bounded by some constant almost surely for all $n$ and $s_n/\sqrt{n}$ is bounded away from zero. Then*

$$\sup_{z\in\mathbb{R}}\left|\mathbb{P}\left(\frac{\sum_{t=1}^n X_t}{\sqrt{\sum_{t=1}^n \sigma_t^2}} \le z\right) - \Phi(z)\right| = O\left(\frac{\log n}{\sqrt{n}} + (\mathbb{E}|V_n^2 - 1|)^{1/3}\right),$$

*where $\Phi(\bullet)$ is the cumulative distribution function of standard normal distribution.*

*Proof.* It follows from Corollary 1 of Bolthausen (1982) and the condition $s_n/\sqrt{n}$ is bounded away from zero that

$$\sup_{z\in\mathbb{R}}\left|\mathbb{P}\left(\frac{\sum_{t=1}^n X_t}{s_n} \le z\right) - \Phi(z)\right| = O\left(\frac{n\log n}{s_n^3} + (\mathbb{E}|V_n^2 - 1|)^{1/3}\right) = O\left(\frac{\log n}{\sqrt{n}} + (\mathbb{E}|V_n^2 - 1|)^{1/3}\right).$$

It follows that

$$
\begin{aligned}
&\sup_{z\in\mathbb{R}}\left|\mathbb{P}\left(\frac{\sum_{t=1}^n X_t}{\sqrt{\sum_{t=1}^n \sigma_t^2}} \le z\right) - \Phi(z)\right| \\
=\ & \sup_{z\in\mathbb{R}}\left|\mathbb{P}\left(\frac{\sum_{t=1}^n X_t}{s_n} \le z + \left(\frac{\sum_{t=1}^n X_t}{\sqrt{\sum_{t=1}^n \sigma_t^2}}\right)(V_n - 1)\right) - \Phi(z)\right| \\
\le\ & \sup_{z\in\mathbb{R}}\mathbb{E}\left|\mathbb{P}\left(\frac{\sum_{t=1}^n X_t}{s_n} \le z + \left(\frac{\sum_{t=1}^n X_t}{\sqrt{\sum_{t=1}^n \sigma_t^2}}\right)(V_n - 1)\right) - \Phi\left(z + \left(\frac{\sum_{t=1}^n X_t}{\sqrt{\sum_{t=1}^n \sigma_t^2}}\right)(V_n - 1)\right)\right| \\
& + \sup_{z\in\mathbb{R}}\mathbb{E}\left|\Phi\left(z + \left(\frac{\sum_{t=1}^n X_t}{\sqrt{\sum_{t=1}^n \sigma_t^2}}\right)(V_n - 1)\right) - \Phi(z)\right| \\
\le\ & O\left(\frac{\log n}{\sqrt{n}} + (\mathbb{E}|V_n^2 - 1|)^{1/3}\right) + \sup_{z\in\mathbb{R}}|\Phi'(z)| \times \mathbb{E}\left|\left(\frac{\sum_{t=1}^n X_t}{\sqrt{\sum_{t=1}^n \sigma_t^2}}\right)(V_n - 1)\right|. \quad (17)
\end{aligned}
$$

By the definition $\Phi(z)$, we have that

$$\sup_{z\in\mathbb{R}}|\Phi'(z)| = \sup_{z\in\mathbb{R}}\frac{1}{\sqrt{2\pi}}\exp(-z^2/2) \le \frac{1}{\sqrt{2\pi}}. \quad (18)$$

Leveraging the facts that $\sup_t |X_t|$ are upper bounded and $s_n/\sqrt{n}$ is lower bounded, we obtain that

$$
\begin{aligned}
\mathbb{E}\left|\left(\frac{\sum_{t=1}^n X_t}{\sqrt{\sum_{t=1}^n \sigma_t^2}}\right)(V_n - 1)\right| &= \mathbb{E}\left|\left(\frac{\sum_{t=1}^n X_t}{s_n}\right)V_n(V_n - 1)\right| \\
&\leq \left(\mathbb{E}\left\{\frac{(\sum_{t=1}^n X_t)^{3/2}}{s_n^{3/2}}V_n^{3/2}\right\}\right)^{2/3}\left(\mathbb{E}\left\{(V_n - 1)^3\right\}\right)^{1/3} \\
&= O\left(\left\{\mathbb{E}|V_n - 1|^2\right\}^{1/3}\right) \\
&= O\left(\left\{\mathbb{E}|V_n^2 - 1|\right\}^{1/3}\right),
\end{aligned}
\tag{19}
$$

where the third-to-last line is derived from Hölder inequality, and the second-to-last equality holds due to the boundedness of $V_n$. Combining equations (17), (18) and (19), we obtain

$$
\sup_{z\in\mathbb{R}}\left|\mathbb{P}\left(\frac{\sum_{t=1}^n X_t}{\sqrt{\sum_{t=1}^n \sigma_t^2}} \leq z\right) - \Phi(z)\right| \leq O\left(\frac{\log n}{\sqrt{n}} + (\mathbb{E}|V_n^2 - 1|)^{1/3}\right),
\tag{20}
$$

which finishes the proof. $\qquad\square$

**Lemma S2.** *Let $\Phi(\bullet)$ be the cumulative distribution function of standard normal distribution and $\Phi'(\bullet)$ be its derivative. Then for any random variable $X$,*

$$
\mathbb{E}\Phi(z_\alpha + X) \geq \min\{1 - \alpha, \alpha + \Phi'(z_\alpha)\mathbb{E}X\},
$$

*where $0 < \alpha < 1/2$, $z_\alpha$ is the $\alpha$-th quantile of standard normal distribution.*

*Proof of Lemma S2.* Since $\Phi'(x) = (\sqrt{2\pi})^{-1}\exp\left(-x^2/2\right)$, we noted that $\Phi'(x) \geq \Phi'(z_\alpha)$ holds if and only if $z_\alpha \leq x \leq z_{1-\alpha}$. Therefore, if $0 \leq X < z_{1-\alpha} - z_\alpha$, then by the mean value theorem,

$$
\Phi(z_\alpha + X) = \Phi(z_\alpha) + \Phi'(\xi)X \geq \alpha + \Phi'(z_\alpha)X,
$$

where $\xi$ lies between $z_\alpha$ and $z_{1-\alpha}$. If $X \leq 0$, then

$$
\Phi(z_\alpha + X) = \Phi(z_\alpha) + \Phi'(\eta)X \geq \alpha + \Phi'(z_\alpha)X,
$$

where $\eta$ lies between $X$ and $z_\alpha$. Moreover, if $X \geq z_{1-\alpha} - z_\alpha$, then $z_\alpha + X \geq z_{1-\alpha}$, It follows that $\Phi(z_\alpha + X) \geq \Phi(z_{1-\alpha}) = 1 - \alpha$. Therefore,

$$
\begin{aligned}
\mathbb{E}\Phi(z_\alpha + X) &\geq \mathbb{E}\min\{\alpha + \Phi'(z_\alpha)X, 1 - \alpha\} \\
&\geq \min\{\alpha + \Phi'(z_\alpha)\mathbb{E}X, 1 - \alpha\},
\end{aligned}
$$

where the last inequality follows from Jensen's inequality. This finishes the proof. $\qquad\square$

In Lemma S3 below, we provide an upper bound for the parameter estimation error. Before doing so, we define

$$
Q_*(\beta) = \left\{\beta^\top\Sigma\beta\right\}^{-\frac{1}{2}}\beta^\top\mu
\tag{21}
$$

$$
\widehat{Q}_n(\beta) = \left\{\beta^\top\widehat{\Sigma}_n\beta\right\}^{-\frac{1}{2}}\beta^\top\widehat{\mu}_n, \quad n \in \mathbb{N}
\tag{22}
$$

where $\widehat{\mu}_n = L^{-1}\sum_{t=1}^L \widehat{\mu}_t^{(1)} - \widehat{\mu}_t^{(2)}$, $\mu = L^{-1}\sum_{t=1}^L \mu_t^{(1)} - \mu_t^{(2)}$ and

$$
\widehat{\mu}_t^{(1)} = \frac{1}{n}\sum_{i=1}^n \mathbb{E}_{\widetilde{X}_t \sim q_t}\phi\left(\log q_t\left(\widetilde{X}_t \mid X_{<t}^{(i)}\right)\right)
$$

$$
\widehat{\mu}_t^{(2)} = \frac{1}{n}\sum_{i=1}^n \phi\left(\log q_t\left(X_t^{(i)} \mid X_{<t}^{(i)}\right)\right)
$$

$$
\mu_t^{(1)} = \mathbb{E}_{X_{<t}\sim p}\mathbb{E}_{X_t\sim q_t}\phi\left(\log q_t\left(X_t \mid X_{<t}\right)\right)
$$

$$
\mu_t^{(2)} = \mathbb{E}_{X_{<t}\sim p}\mathbb{E}_{X_t\sim p_t}\phi\left(\log q_t\left(X_t \mid X_{<t}\right)\right)
$$

for each $t = 1, \ldots, L$. Similarly, define $\widehat{\Sigma}_n = L^{-1} \sum_{t=1}^{L} \widehat{\Sigma}_t$ and $\Sigma = L^{-1} \sum_{t=1}^{L} \Sigma_t$ where

$$\widehat{\Sigma}_t = \frac{1}{n} \sum_{t=1}^{n} \mathbb{E}_{\widetilde{X}_t \sim q_t} \left[ \phi\left(\log q_t\left(\widetilde{X}_t \mid X_{<t}^{(i)}\right)\right) \phi\left(\log q_t\left(\widetilde{X}_t \mid X_{<t}^{(i)}\right)\right)^{\top} \right] - \widehat{\mu}_t^{(1)} \left(\widehat{\mu}_t^{(1)}\right)^{\top}$$

for each $t = 1, \ldots, L$.

**Lemma S3.** *Grant the assumptions in Section D hold. Let $\beta^*$ be the maximizer of the function (21) over all $\beta$'s with $\ell_2$ norm equal to 1 and let $\widehat{\beta}$ be the maximizer of the empirical counterpart (22). There are absolute constants $\kappa_1$ and $\kappa_2$ depending only on* the constants stated in the assumptions *such that for any $z > 0$ it holds that $\|\widehat{\beta} - \beta^*\|_2 \leq z$ with probability at least* $1 - \kappa_1 \exp\left(-\kappa_2 d^{-5\gamma} n (\min\{z, 1\})^2\right)$.

*Proof.* Observing the $\widehat{\beta} \in \arg\max_{\beta} \widehat{Q}_n(\beta)$ and $\beta^* \in \arg\max_{\beta} Q^*(\beta)$ for any $z > 0$

$$\mathbb{P}\left(\left\|\widehat{\beta} - \beta^*\right\|_2 > z\right) = \mathbb{P}\left(\sup_{\beta : \|\beta - \beta^*\|_2 > z} Q_n(\beta) - Q_n(\beta^*) > 0\right) \tag{23}$$

$$\leq \mathbb{P}\left(\sup_{\beta : \|\beta - \beta^*\|_2 > z} |Q_n(\beta) - Q_*(\beta)| > \frac{1}{2} \inf_{\beta : \|\beta - \beta^*\|_2 > z} [Q_*(\beta^*) - Q_*(\beta)]\right)$$

$$+ \mathbb{P}\left(|Q_n(\beta^*) - Q_*(\beta^*)| > \frac{1}{2} \inf_{\beta : \|\beta - \beta^*\|_2 > z} [Q_*(\beta^*) - Q_*(\beta)]\right).$$

For $\beta$ such that $\|\beta - \beta^*\|_2 > z$, due to Assumption 2 with some absolute $\kappa_1$ we must have that

$$Q_*(\beta^*) - Q_*(\beta) \geq \kappa_1 d^{-\gamma} \|\beta - \beta^*\|_2 \geq \kappa_1 z d^{-\gamma}. \tag{24}$$

For any $\beta \in \mathbb{R}^d$ with $\|\beta\|_2 = 1$ it holds that

$$|Q_n(\beta) - Q_*(\beta)|$$

$$\leq \left(\beta^{\top}\Sigma\beta\right)^{-\frac{1}{2}} \left|\beta^{\top}(\widehat{\mu}_n - \mu)\right| + \left|\beta^{\top}\widehat{\mu}_n\right| \left|\left(\beta^{\top}\widehat{\Sigma}_n\beta\right)^{-\frac{1}{2}} - \left(\beta^{\top}\Sigma\beta\right)^{-\frac{1}{2}}\right|$$

$$\leq \left(\beta^{\top}\Sigma\beta\right)^{-\frac{1}{2}} \left|\beta^{\top}(\widehat{\mu}_n - \mu)\right|$$

$$+ \frac{1}{2}\left\{\min\left(\beta^{\top}\widehat{\Sigma}_n\beta, \beta^{\top}\Sigma\beta\right)\right\}^{-\frac{3}{2}} \left|\beta^{\top}\widehat{\mu}_n\right| \left|\beta^{\top}\widehat{\Sigma}_n\beta - \beta^{\top}\Sigma\beta\right| \tag{25a}$$

$$\leq \{\lambda_{\min}(\Sigma)\}^{-\frac{1}{2}} \left|\beta^{\top}(\widehat{\mu}_n - \mu)\right| \tag{25b}$$

$$+ \frac{1}{2}\left\{\min\left(\lambda_{\min}(\widehat{\Sigma}_n), \lambda_{\min}(\Sigma)\right)\right\}^{-\frac{3}{2}} \left|\beta^{\top}\widehat{\mu}_n\right| \left|\beta^{\top}\widehat{\Sigma}_n\beta - \beta^{\top}\Sigma\beta\right|, \tag{25c}$$

where in particular (25a) holds due to the inequality

$$\left|\frac{1}{\sqrt{x}} - \frac{1}{\sqrt{y}}\right| = \left|\frac{x - y}{\sqrt{xy}(\sqrt{x} + \sqrt{y})}\right| \leq \frac{1}{2\left(\min(x, y)\right)^{\frac{3}{2}}} |x - y| \tag{26}$$

for $x, y > 0$. For (25b) note that due to the boundedness of $\phi(\bullet)$ for any $\beta$ with $\|\beta\|_2 = 1$ it holds that $\beta^{\top}(\widehat{\mu}_n - \mu)$ is the average of $n$ i.i.d. random variables each bounded in absolute value by a constant which does not depend on $\beta$. Therefore lower bounding $\lambda_{\min}(\Sigma) \geq \kappa_2 d^{-\gamma}$ for some absolute $\kappa_2$ using again the boundedness of $\phi(\bullet)$ and applying Hoeffding's inequality, for any $z > 0$

$$\mathbb{P}\left((25b) > \frac{1}{2}\kappa_1 z d^{-\gamma}\right) \leq \mathbb{P}\left(\left|\beta^{\top}(\widehat{\mu}_n - \mu)\right| > \kappa_3 z d^{-\frac{3\gamma}{2}}\right) \leq 2\exp\left(-\frac{\kappa_4 n z^2}{d^{3\gamma}}\right) \tag{27}$$

for certain absolute $\kappa_3, \kappa_4$. The following argument will be valid on the event

$$\lambda_{\min}(\widehat{\Sigma}_n) \geq \frac{1}{2}\lambda_{\min}(\Sigma). \tag{28}$$

For (25c) notice first that with $\|\beta\|_2 = 1$ the quantity $|\beta^{\top}\widehat{\mu}_n|$ is almost surely bounded from above by some absolute constant independent of $\beta$ and $d$. Moreover due to the boundedness of $\phi(\bullet)$ it is easy

to see that the statistic $|\beta^\top \widehat{\Sigma}_n \beta - \beta^\top \Sigma \beta|$ is a self-bounding function of $n$ random variables with constants (see equation (16)) $c_i \propto \frac{1}{n}$ for $i = 1, \ldots, n$ which again do not depend on $\beta$. Therefore, on the event (28) applying Theorem S6 we obtain that

$$\mathbb{P}\left((25\text{c}) > \frac{1}{2}\kappa_1 d^{-\gamma}\right) \leq \mathbb{P}\left(\left|\beta^\top \widehat{\Sigma}_n \beta - \beta^\top \Sigma \beta\right| > \kappa_5 z d^{-\frac{5\gamma}{2}}\right) \leq \exp\left(-\frac{\kappa_6 n z^2}{d^{5\gamma}}\right) \quad (29)$$

for certain absolute $\kappa_5, \kappa_6$. Finally, note that

$$\lambda_{\min}(\widehat{\Sigma}) = \min_{\beta:\|\beta\|_2=1} \beta^\top \widehat{\Sigma}_n \beta \geq \lambda_{\min}(\Sigma) - \max_{\beta:\|\beta\|_2=1} \beta^\top (\widehat{\Sigma}_n - \Sigma)\beta \quad (30)$$

and arguing as in (29), the final term (30) is no larger than $\frac{1}{2}\kappa_2 d^{-\gamma}$ with probability at least

$$1 - 2\exp\left(-\frac{\kappa_7 n}{d^{2\gamma}}\right).$$

Since by Assumption 2 we must have that $\kappa_2 d^{-\gamma} \leq \lambda_{\min}(\Sigma)$ the event (28) must hold with the above probability. Since the above arguments hold for any $\beta$ with $\|\beta\|_2 = 1$, plugging (27) and (29) back into (23) and accounting for the event (30) the stated result follows. $\qquad\square$

### E.3 Proof of Theorem 1

According to the decomposition $T_w(\boldsymbol{X}) = T_w^{(1)}(\boldsymbol{X}) - T_w^{(2)}(\boldsymbol{X})$ with $T_w^{(1)}(\boldsymbol{X}), T_w^{(2)}(\boldsymbol{X})$ defined by

$$
\begin{aligned}
T_w^{(1)}(\boldsymbol{X}) &= \frac{\sum_t [w(\log q_t(X_t|X_{<t})) - \mathbb{E}_{\widetilde{X}_t \sim p_t} w(\log q_t(\widetilde{X}_t|X_{<t}))]}{\sqrt{\sum_t \mathrm{Var}_{\widetilde{X}_t \sim q_t}(w(\log q_t(\widetilde{X}_t|X_{<t})))}} \\
T_w^{(2)}(\boldsymbol{X}) &= \frac{\sum_t [\mathbb{E}_{\widetilde{X}_t \sim q_t} w(\log q_t(\widetilde{X}_t|X_{<t})) - \mathbb{E}_{\widetilde{X}_t \sim p_t} w(\log q_t(\widetilde{X}_t|X_{<t})))]}{\sqrt{\sum_t \mathrm{Var}_{\widetilde{X}_t \sim q_t}(w(\log q_t(\widetilde{X}_t|X_{<t})))}},
\end{aligned}
\quad (31)
$$

we obtain that the TNR can be represented as

$$\mathbb{P}_{\boldsymbol{X}\sim p}(T_w(\boldsymbol{X}) \leq z_\alpha) = \mathbb{P}_{\boldsymbol{X}\sim p}\left(T_w^{(1)}(\boldsymbol{X}) \leq z_\alpha + T_w^{(2)}(\boldsymbol{X})\right) \quad (32)$$

It is easy to verify that when $\boldsymbol{X} \sim p$, $T_w^{(1)}(\boldsymbol{X})\sigma_{q,L}/\sigma_{p,L}$ converges to standard normal distribution. Specifically, using Lemma S1, we obtain that

$$
\begin{aligned}
\mathbb{P}_{\boldsymbol{X}\sim p}(T_w(\boldsymbol{X}) \leq z_\alpha) &= \mathbb{P}_{\boldsymbol{X}\sim p}\left(T_w^{(1)}(\boldsymbol{X})\frac{\sigma_{q,L}}{\sigma_{p,L}} \leq (z_\alpha + T_w^{(2)}(\boldsymbol{X}))\frac{\sigma_{q,L}}{\sigma_{p,L}}\right) \\
&\geq \Phi(z_\alpha + T_w^{(2)}(\boldsymbol{X})) + \left(\Phi\left((z_\alpha + T_w^{(2)}(\boldsymbol{X}))\frac{\sigma_{q,L}}{\sigma_{p,L}}\right) - \Phi(z_\alpha + T_w^{(2)}(\boldsymbol{X}))\right) \\
&\quad + O\left(\log L/\sqrt{L}\right) + o_p(1) \\
&\geq \Phi(z_\alpha + T_w^{(2)}(\boldsymbol{X})) - \sup_{z\in\mathbb{R}}\Phi'(z) \times \left|z_\alpha + T_w^{(2)}(\boldsymbol{X})\right| \times \left|\frac{\sigma_{q,L}}{\sigma_{p,L}} - 1\right| \\
&\quad + O\left(\log L/\sqrt{L}\right) + o_p(1),
\end{aligned}
$$

where the little-$o_p$ term arises due to the asymptotic equivalence between $\sigma_{p,L}$ and $\bar{\sigma}_{p,L}$ in Assumption 4.

Take expectation on both sides, we have by Assumption 3 that

$$\mathbb{P}_{\boldsymbol{X}\sim p}(T_w(\boldsymbol{X}) \leq z_\alpha) \geq \mathbb{E}\Phi(z_\alpha + T_w^{(2)}(\boldsymbol{X})) + o(1) + O(\log L/\sqrt{L}).$$

Next, define $\widetilde{\sigma}_{q,L}^2 = \mathbb{E}_{\boldsymbol{X}\sim p}\sigma_{q,L}^2$. It follows that $T_w^{(2*)}(\boldsymbol{X}) = \mathbb{E}\left\{T_w^{(2)}(\boldsymbol{X})\frac{\sigma_{q,L}}{\widetilde{\sigma}_{q,L}}\right\}$. Under the equal variance assumption in Assumption 3, we also have $\sigma_{q,L} - \widetilde{\sigma}_{q,L} \to 0$ in probability. It follows that

for any $\epsilon > 0$,

$$
\begin{aligned}
\mathbb{E}\Phi(z_\alpha + T_w^{(2)}(\boldsymbol{X})) &= \mathbb{E}\Phi(z_\alpha + T_w^{(2)}(\boldsymbol{X}))\mathbb{I}\{|\sigma_{q,L} - \widetilde{\sigma}_{q,L}| \leq \epsilon\} \\
&\quad + \mathbb{E}\Phi(z_\alpha + T_w^{(2)}(\boldsymbol{X}))\mathbb{I}\{|\sigma_{q,L} - \widetilde{\sigma}_{q,L}| > \epsilon\} \\
&\geq \mathbb{E}\Phi(z_\alpha + T_w^{(2)}(\boldsymbol{X}))\mathbb{I}\{|\sigma_{q,L} - \widetilde{\sigma}_{q,L}| \leq \epsilon\} \\
&\geq \mathbb{E}\Phi\left(z_\alpha + T_w^{(2)}(\boldsymbol{X})\frac{\sigma_{q,L}}{\widetilde{\sigma}_{q,L} + \mathrm{sgn}(T_w^{(2)})\epsilon}\right)\mathbb{I}\{|\sigma_{q,L} - \widetilde{\sigma}_{q,L}| \leq \epsilon\} \\
&\geq \mathbb{E}\Phi\left(z_\alpha + T_w^{(2)}(\boldsymbol{X})\frac{\sigma_{q,L}}{\widetilde{\sigma}_{q,L} + \mathrm{sgn}(T_w^{(2)})\epsilon}\right) \\
&\quad - \mathbb{E}\Phi\left((z_\alpha + T_w^{(2)}(\boldsymbol{X}))\frac{\sigma_{q,L}}{\widetilde{\sigma}_{q,L} + \mathrm{sgn}(T_w^{(2)})\epsilon}\right)\mathbb{I}\{|\sigma_{q,L} - \widetilde{\sigma}_{q,L}| > \epsilon\} \\
&\geq \mathbb{E}\Phi\left((z_\alpha + T_w^{(2)}(\boldsymbol{X}))\frac{\sigma_{q,L}}{\widetilde{\sigma}_{q,L} + \mathrm{sgn}(T_w^{(2)})\epsilon}\right) - \mathbb{P}(|\sigma_{q,L} - \widetilde{\sigma}_{q,L}| > \epsilon),
\end{aligned}
$$

where the first inequality is obtained due to $\Phi$ is non-negative and the second inequality holds due to the monotonicity and boundedness of $\Phi$. Together with Lemma S2 and Assumption 3, we obtain

$$
\begin{aligned}
&\mathbb{P}_{\boldsymbol{X}\sim p}\left(T_w(\boldsymbol{X}) \leq z_\alpha\right) \\
&\geq \min\left\{1 - \alpha, \alpha + \phi(z_\alpha)\mathbb{E}\left\{T_w^{(2)}(\boldsymbol{X})\frac{\sigma_{q,L}}{\widetilde{\sigma}_{q,L}}\right\}\right\}\frac{\widetilde{\sigma}_{q,L}}{\widetilde{\sigma}_{q,L} + \mathrm{sgn}(T_w^{(2)})\epsilon} \\
&\quad - \mathbb{P}\{|\sigma_{q,L} - \widetilde{\sigma}_{q,L}| \geq \epsilon\} + O\left(\log L/\sqrt{L}\right) + o(1).
\end{aligned}
\tag{33}
$$

Let $L \to \infty$ and using the fact that $\mathbb{E}\left\{T_w^{(2)}(\boldsymbol{X})\frac{\sigma_{q,L}}{\widetilde{\sigma}_{q,L}}\right\} = T_w^{(2*)}(\boldsymbol{X})$, we obtain that TNR is asymptotically lower bounded by $\min\{1 - \alpha, \alpha + \phi(z_\alpha)T_w^{(2*)}(\boldsymbol{X})\}\frac{\widetilde{\sigma}_{q,L}}{\widetilde{\sigma}_{q,L} + \mathrm{sgn}(T_w^{(2)})\epsilon}$. By taking $\epsilon \to 0$, then the conclusion of Theorem 1 follows.

**Remark 1.** *In fact, the equal variance condition (Assumption 3) can be relaxed. Specifically, it is not necessary for the two variance $\sigma_{q,L}^2$ and $\sigma_{p,L}^2$ to be asymptotically equivalent in probability. Rather, it suffices to require their ratio to converge to some positive constant in probability, i.e.,*

$$
\frac{\sigma_{q,L}}{\sigma_{p,L}} \xrightarrow{P} K_0.
$$

*Since $K_0$ need not be 1, this proportionality condition is considerably weaker than the equal variance assumption and is more likely to hold in practice. Under this relaxed condition, following nearly identical arguments to those in Theorem 3, we can show that TNR is asymptotically lower bounded by:*

$$
\min\left\{1 - \Phi(K_0 z_\alpha), \Phi(K_0 z_\alpha) + \phi(K_0 z_\alpha)K_0 T_w^{(2*)}\right\}.
$$

*This lower bound differs from the one in Theorem 3 due to the change in assumptions. However, since $\phi(K_0 z_\alpha)$ depends solely on $\alpha$ (not on the witness function $w$), our core conclusion – maximizing the lower bound is equivalent to maximizing $T_w^{(2*)}$ – remains valid. Thus, our proposed methodology remains theoretically sound.*

### E.4 Proof of Theorem 2

*Proof.* Denote $Z_t = \widehat{w}\left(\log q_t(X_t|X_{<t})\right) - \mathbb{E}_{\widetilde{X}_t \sim q_t(\bullet|X_{<t})}\widehat{w}\left(\log q_t(X_t|X_{<t})\right)$. Then if $\boldsymbol{X} \sim q$, we have $\mathbb{E}\{Z_t|X_{<t}\} = 0$ almost surely. Without loss of generality, assume $\widehat{w}$ is bounded. Otherwise, we can define $\widehat{w} = \phi^\top\widehat{\beta}/\|\widehat{\beta}\|_2$ to make it bounded. Under the lower bound assumption in Assumption 3, it is easy to verify that $Z_t$ satisfies all conditions of Lemma S1. Therefore, by invoking Lemma S1,

we obtain for any $\alpha \in (0,1)$,

$$
\begin{aligned}
\mathrm{FNR}_{\widehat{w}} - \alpha &= \mathbb{P}_{\boldsymbol{X} \sim q}(T_{\widehat{w}}(\boldsymbol{X}) \le z_\alpha) - \Phi(z_\alpha) \\
&= \mathbb{P}_{\boldsymbol{X} \sim q}\left( \frac{\sum_{t=1}^{L} Z_t}{\sum_{t=1}^{L} \mathbb{E}\{Z_t^2 | X_{<t}\}} \le z_\alpha \right) - \Phi(z_\alpha) \\
&= O\left( \frac{\log L}{\sqrt{L}} \right) + O\big((\mathbb{E}|V_L - 1|)^{1/3}\big).
\end{aligned}
$$

Taking expectation on both sides, we obtain

$$
\mathbb{E}(\mathrm{FNR}_{\widehat{w}}) = \alpha + O\left( \frac{\log L}{\sqrt{L}} \right) + O\big((\mathbb{E}|V_L - 1|)^{1/3}\big).
$$

This completes the proof. $\qquad\square$

### E.5  Proof of Theorem 3

*Proof.* Since $\mathbb{E}(\mathrm{TNR}_{\widehat{w}}) \ge \mathrm{TNR}_{w^*} - \mathbb{E}(|\mathrm{TNR}_{\widehat{w}} - \mathrm{TNR}_{w^*}|)$, it is enough to upper bound the second term in the last expression. Denote by $\widehat{T}_n(\bullet)$ and $\widehat{T}^*(\bullet)$ respectively the classifier (4) using witness functions $\widehat{w}(\bullet) = \phi(\bullet)^\top \widehat{\beta}$ and $w^*(\bullet) = \phi(\bullet)^\top \beta^*$ (see the definition of $\widehat{\beta}$ and $\beta^*$ in Lemma S3). Write $\widehat{\Delta}_n(\boldsymbol{x}) = |\widehat{T}_n(\boldsymbol{x}) - T^*(\boldsymbol{x})|$ and $\widehat{\Delta}_n = \sup_{\boldsymbol{x}} \Delta_n(\boldsymbol{x})$. For any $z_\alpha > 0$ we have that

$$
\begin{aligned}
|\mathrm{TNR}_{\widehat{w}} - \mathrm{TNR}_{w^*}| &= \left| \mathbb{P}_{\boldsymbol{X} \sim p}\left( \widehat{T}_n(\boldsymbol{X}) \le z_\alpha \right) - \mathbb{P}_{\boldsymbol{X} \sim p}\left( T^*(\boldsymbol{X}) \le z_\alpha \right) \right| \\
&= \left| \int \mathbf{1}_{\{\widehat{T}_n(\boldsymbol{x}) \le z_\alpha\}} - \mathbf{1}_{\{T^*(x) > z_\alpha\}} \mathrm{d}p(\boldsymbol{x}) \right| \\
&\le \int \left| \mathbf{1}_{\{\widehat{T}_n(\boldsymbol{x}) \le z_\alpha\}} - \mathbf{1}_{\{T^*(x) > z_\alpha\}} \right| \mathrm{d}p(\boldsymbol{x}) \\
&= \int \mathbf{1}_{\{\widehat{T}_n(\boldsymbol{x}) \le z_\alpha, T^*(\boldsymbol{x}) > z_\alpha\}} + \mathbf{1}_{\{\widehat{T}_n(\boldsymbol{x}) > z_\alpha, T^*(\boldsymbol{x}) \le z_\alpha\}} \mathrm{d}p(\boldsymbol{x}) \\
&\le 2 \int \mathbf{1}_{\{|T^*(\boldsymbol{x}) - z_\alpha| \le \widehat{\Delta}_n\}} \mathrm{d}p(\boldsymbol{x}) \\
&= 2 \mathbb{P}_{\boldsymbol{X} \sim p}\left( |T^*(\boldsymbol{X}) - z_\alpha| \le \widehat{\Delta}_n \right).
\end{aligned} \tag{34}
$$

Due to Assumption 1 on the event

$$
\left\{ \widehat{\Delta}_n \le \delta_0 \right\} \tag{35}
$$

we will have that $|\mathrm{TNR}_{\widehat{w}} - \mathrm{TNR}_{w^*}| \le \kappa_3 \widehat{\Delta}_n$ for some absolute $\kappa_3$. We therefore focus on bounding the quantity $\widehat{\Delta}_n$. For each $w \in \Omega$ and each $j = 1, \dots, L$, we introduce the quantities:

$$
\begin{aligned}
Y_j^{(w)} &= w\left( \log q_j\left( X_j \mid X_{<j} \right) \right), \\
\mu_j^{(w)} &= \mathbb{E}_{\widetilde{X}_j \sim q_j} w\left( \log q_j\left( \widetilde{X}_j \mid X_{<j} \right) \right), \\
(\sigma_j^{(w)})^2 &= \mathrm{Var}_{\widetilde{X}_j \sim q_j} w\left( \log q_j\left( \widetilde{X}_j \mid X_{<j} \right) \right).
\end{aligned}
$$

With this notation in place we have that for any $\boldsymbol{x}$

$$
\widehat{\Delta}_n(\boldsymbol{x}) \le \frac{1}{\sqrt{L}} \left| \sum_{j=1}^{L} y_j^{(\widehat{w})} - \mu_j^{(\widehat{w})} \right| \left| \left[ L^{-1} \sum_{j=1}^{L} \left( \sigma_j^{(\widehat{w})} \right)^2 \right]^{-1/2} - \left[ L^{-1} \sum_{j=1}^{L} \left( \sigma_j^{(w^*)} \right)^2 \right]^{-1/2} \right| \tag{36a}
$$

$$
+ \left\{ \frac{1}{L} \sum_{j=1}^{L} \left( \sigma_j^{(w^*)} \right)^2 \right\}^{-\frac{1}{2}} \frac{1}{\sqrt{L}} \left| \sum_{j=1}^{L} \left( Y_j^{(\widehat{w})} - Y_j^{(w^*)} \right) - \left( \mu_j^{(\widehat{w})} - \mu_j^{(w^*)} \right) \right|, \tag{36b}
$$

where for clarity we have suppressed dependence on $x$ above. For ease of notation put $Z_t = \log q_t(X_t \mid X_{<t})$ and $\widetilde{Z}_t = \log q_t(\widetilde{X}_t \mid X_{<t})$ where $\widetilde{X}_t \sim q_t$. Write also $\phi(\bullet) = (B_1(\bullet), \dots, B_d(\bullet))^\top$. Recalling that $w(\bullet) = \phi(\bullet)^\top \beta$ for arbitrary $j = 1, \dots, L$ we have

$$
\begin{aligned}
\left| \left(\sigma_j^{(\widehat{w})}\right)^2 - \left(\sigma_j^{(w^*)}\right)^2 \right| &\leq \mathbb{E}\left[ \sum_{l_1=1}^{d} \sum_{l_2=1}^{d} \left| \widehat{\beta}_{l_1} \widehat{\beta}_{l_2} - \beta_{l_1}^* \beta_{l_2}^* \right| \left( |B_{l_1}(Z_j) B_{l_2}(z_j)| \right. \right. \\
&\qquad\qquad \left. \left. + \left| \mathbb{E}\left[ B_{l_1}(\widetilde{Z}_j) \right] \mathbb{E}\left[ B_{l_2}(\widetilde{Z}_j) \right] \right| + 2 \left| B_{l_2}(z_j) \mathbb{E}\left[ B_{l_1}(\widetilde{Z}_j) \right] \right| \right) \right] \\
&\leq \frac{\kappa_4}{2} \sum_{l_1=1}^{d} \sum_{l_2=1}^{d} \left| \widehat{\beta}_{l_1} \widehat{\beta}_{l_2} - \beta_{l_1}^* \beta_{l_2}^* \right| \\
&= \frac{\kappa_4}{2} \sum_{l_1=1}^{d} \sum_{l_2=1}^{d} \left| \widehat{\beta}_{l_1} \left( \widehat{\beta}_{l_2} - \beta_{l_2}^* \right) - \beta_{l_2}^* \left( \beta_{l_1}^* - \widehat{\beta}_{l_1} \right) \right| \\
&\leq \frac{\kappa_4}{2} \sum_{l_1=1}^{d} \sum_{l_2=1}^{d} \left| \widehat{\beta}_{l_1} \right| \left| \widehat{\beta}_{l_2} - \beta_{l_2}^* \right| + \frac{\kappa_4}{2} \sum_{l_1=1}^{d} \sum_{l_2=1}^{d} \left| \beta_{l_2}^* \right| \left| \widehat{\beta}_{l_1} - \beta_{l_1}^* \right| \\
&= \kappa_5 \sqrt{d} \left\| \widehat{\beta} - \beta^* \right\|_1 \kappa_5 \\
&\leq d \left\| \widehat{\beta} - \beta^* \right\|_2 ,
\end{aligned}
$$

for absolute $\kappa_4, \kappa_5$. Consequently, using inequality (26) and Assumption 2, on the event (30) we obtain that with absolute $\kappa_6$:

$$
(36a) \leq \kappa_6 d^3 \frac{1}{\sqrt{L}} \left| \sum_{j=1}^{L} y_j^{(\widehat{w})} - \mu_j^{(\widehat{w})} \right| \left\| \widehat{\beta} - \beta^* \right\|_2. \tag{37}
$$

Observe that conditional on $\widehat{\beta}$ the term $\sum_{j=1}^{L}(y_j^{(\widehat{w})} - \mu_j^{(\widehat{w})})$ is a martingale with increments bounded from above almost surely by a constant independent on $\widehat{\beta}$; by the Azuma–Hoeffding inequality the normalized sum in (37) has sub-Gaussian tails. By Lemma S3 the term $\|\widehat{\beta} - \beta^*\|_2$ likewise has sub-Gaussian tails. Therefore on the relevant events we obtain that (37) has sub-exponential tails, and consequently for any $z > 0$

$$
\mathbb{P}\left( 37 > z \right) \leq \kappa_7 \exp\left( -\kappa_8 \min\left\{ z^2 \frac{n}{d^{5\gamma+6}}, z \sqrt{\frac{n}{d^{5\gamma+6}}} \right\} \right) \tag{38}
$$

for certain absolute $\kappa_7, \kappa_8$. Similar arguments show that the normalized sum in (36b) has the same tail behavior as (38). Since the above augments do not depend on $x$ we obtain that (38) likewise described the tail behavior of $\widehat{\Delta}_n$. Consequently, on the relevant events we obtain that

$$
\begin{aligned}
\mathbb{E}\left| \mathrm{TNR}_{\widehat{w}} - \mathrm{TNR}_{w^*} \right| &= \int_0^\infty \mathbb{P}\left( \left| \mathrm{TNR}_{\widehat{w}} - \mathrm{TNR}_{w^*} \right| > z \right) \mathrm{d}z \\
&\leq \int_0^\infty \mathbb{P}\left( \kappa_3 \widehat{\Delta}_n > z \right) \mathrm{d}z \\
&\leq \kappa_9 \int_0^\infty \exp\left( -\kappa_{10} \min\left\{ z^2 \frac{n}{d^{5\gamma+6}}, z \sqrt{\frac{n}{d^{5\gamma+6}}} \right\} \right) \mathrm{d}z \\
&\leq \kappa_{11} \sqrt{\frac{d^{5\gamma+6}}{n}} \tag{39}
\end{aligned}
$$

for certain absolute $\kappa_9, \kappa_{10}, \kappa_{11}$. When the events (35) and (26) do not hold from (34) we have the conservative bound $\mathbb{E}\left| \mathrm{TNR}_{\widehat{w}} - \mathrm{TNR}_{w^*} \right| \leq 1$. However, the probability of these events not holding is smaller than (39) up to constants. Therefore, the stated result follows by the law of total expectation. $\qquad \square$

### E.6 Smooth witness functions

An important quality of spline estimators, which in part motivated our choice of estimator for the witness function, is their ability to learn smooth regression functions at optimal rates. Following the proof of Theorem 2.1 in Chen & Christensen (2015) one can show that when the optimal witness (that is, the function minimizing the functional (21) among all functions with bounded $\ell_2$ norm) is $\beta$-Hölder smooth, the estimated witness function attains the rate

$$\sup_z |\widehat{w}(z) - w^*(z)| = O_P\left(\sqrt{\frac{d\log(n)}{n}}\right) + O(d^{-\beta}).$$

Consequently, choosing the number of spline bases as $d = \Theta((n/\log(n))^{\frac{1}{2\beta+1}})$ the sup norm loss will be of the order $O_P\left((\log(n)/n)^{\frac{\beta}{2\beta+1}}\right)$. This rate was shown to be optimal by Stone (1982). One can therefore show that choosing the number of spline bases in this way the expected true negative rate will be lower bounded as

$$\mathbb{E}(\mathrm{TNR}_{\widehat{w}}) \geq \mathrm{TNR}_{w^*} - O\left((\log(n)/n)^{\frac{\beta}{2\beta+1}}\right).$$

In this section we provide a sketch of this result.

We argue along the same lines as the proof of Theorem 3. Since $\mathbb{E}(\mathrm{TNR}_{\widehat{w}}) \geq \mathrm{TNR}_{w^*} - \mathbb{E}(|\mathrm{TNR}_{\widehat{w}} - \mathrm{TNR}_{w^*}|)$ is enough to upper bound $\mathbb{E}(|\mathrm{TNR}_{\widehat{w}} - \mathrm{TNR}_{w^*}|)$. Define the quantity $\widehat{\Delta}_n = \sup_{\boldsymbol{x}} |\widehat{T}_n(\boldsymbol{x}) - T^*(\boldsymbol{x})|$, and introduce the event

$$A(\kappa_1) = \left\{\widehat{\Delta}_n \leq \kappa_1 \times (d^{-\beta} + (\log(n)/n)^{\frac{\beta}{2\beta+1}})\right\} \tag{40}$$

Therefore, with arbitrary $\kappa_1$ for some absolute $\kappa_2$ we have that

$$\mathbb{E}(|\mathrm{TNR}_{\widehat{w}} - \mathrm{TNR}_{w^*}|) = \mathbb{E}(|\mathrm{TNR}_{\widehat{w}} - \mathrm{TNR}_{w^*}| \times \mathbf{1}_{\{A(\kappa_1)\}}) + \mathbb{E}(|\mathrm{TNR}_{\widehat{w}} - \mathrm{TNR}_{w^*}| \times \mathbf{1}_{\{\neg A(\kappa_1)\}})$$
$$\leq 2\mathbb{E}(\mathbb{P}_{\boldsymbol{X}\sim p}(|T^*(\boldsymbol{X}) - z_\alpha| \leq \widehat{\Delta}_n) \times \mathbf{1}_{\{A(\kappa_1)\}}) + \mathbb{P}(\neg A(\kappa_1)) \tag{41a}$$
$$\leq \kappa_1 \mathbb{E}(\hat{\Delta}_n \mid A(\kappa_1)) + \mathbb{P}(\neg A(\kappa_1)) \tag{41b}$$

where (41a) follows from the arguments leading up to (34) and (41b) holds on the event (35). Using the mean value theorem and Assumption 3 one can show that up to constants $\hat{\Delta}_n$ is smaller than $\sup_z |\widehat{w}(z) - w^*(z)|$, therefore by (40) the first term in (41b) will be of the order $O(d^{-\beta} + (\log(n)/n)^{\frac{\beta}{2\beta+1}})$. Examining the proof of Theorem 2.1 in Chen & Christensen (2015) it can be seen that for any $\kappa_3 > 0$ on may choose $\kappa_1$ so that $\mathbb{P}(\neg A(\kappa_1)) < n^{-\kappa_3}$. Therefore, we can choose $\kappa_1$ appropriately so that the second term in (41b) will be of the order $o((\log(n)/n)^{\frac{\beta}{2\beta+1}})$. Finally letting $d = \Theta((n/\log(n))^{\frac{1}{2\beta+1}})$ yields the desired result.

As mentioned in the main text, when the optimal witness function is believed to be smooth but the order of the smoothness is not known the number of spline bases can be chosen in a data driven way via Lepski's method (Lepski & Spokoiny, 1997).

# F   Additional numerical results

## F.1   Results on additional open-source models

Table S7: Performance on three open-source LLMs (Qwen2.5, Mistral, LLaMA3) across five datasets.

| Model | Method | XSum | Writing | Essay | SQuAD | Yelp |
|---|---|---|---|---|---|---|
| Qwen2.5 | Likelihood | 0.6175 | 0.7041 | 0.6755 | 0.5183 | 0.6793 |
| | Entropy | 0.5403 | 0.5043 | 0.5073 | 0.5232 | 0.5236 |
| | LogRank | 0.6325 | 0.7150 | 0.6958 | 0.5166 | 0.6943 |
| | Binoculars | 0.6297 | 0.7578 | 0.8018 | 0.6164 | 0.7199 |
| | TextFluoroscopy | 0.5778 | 0.5110 | 0.5638 | 0.5383 | 0.5060 |
| | RADAR | 0.6469 | 0.6190 | 0.6061 | **0.6262** | 0.6276 |
| | ImBD | 0.6653 | 0.6584 | 0.7874 | 0.5168 | 0.7392 |
| | BiScope | 0.6320 | 0.6610 | 0.6625 | 0.6250 | 0.7050 |
| | Fast-DetectGPT | 0.7523 | 0.8513 | 0.8347 | 0.5016 | 0.8465 |
| | AdaDetectGPT | **0.7963** | **0.8965** | **0.8799** | 0.6044 | **0.8915** |
| | Relative | 17.7682 | 30.3912 | 27.3167 | 0.6431 | 29.3165 |
| Mistral | Likelihood | 0.7409 | 0.8643 | 0.8667 | 0.7068 | 0.7598 |
| | Entropy | 0.5290 | 0.5420 | 0.6052 | 0.5070 | 0.5103 |
| | LogRank | 0.7270 | 0.8446 | 0.8467 | 0.7041 | 0.7499 |
| | Binoculars | 0.7218 | 0.8440 | 0.8314 | 0.7258 | 0.7502 |
| | TextFluoroscopy | 0.6210 | 0.5555 | 0.5127 | 0.5772 | 0.5109 |
| | RADAR | 0.6518 | 0.6537 | 0.6292 | 0.6055 | 0.6018 |
| | ImBD | 0.7683 | 0.8391 | 0.8631 | 0.8073 | 0.7440 |
| | BiScope | 0.7320 | **0.8740** | 0.9000 | 0.7283 | 0.7840 |
| | Fast-DetectGPT | 0.8922 | 0.8151 | 0.9052 | 0.8812 | 0.8902 |
| | AdaDetectGPT | **0.8944** | 0.8275 | **0.9069** | **0.8851** | **0.9026** |
| | Relative | 2.0423 | 6.6718 | 1.8332 | 3.3051 | 11.2763 |
| LLaMA3 | Likelihood | 0.8236 | 0.8929 | 0.9115 | 0.7071 | 0.8915 |
| | Entropy | 0.5545 | 0.5732 | 0.5626 | 0.5047 | 0.5010 |
| | LogRank | 0.8634 | 0.9122 | 0.9351 | 0.7422 | 0.9146 |
| | Binoculars | 0.9546 | 0.9845 | **0.9949** | 0.9469 | 0.9854 |
| | TextFluoroscopy | 0.5479 | 0.5274 | 0.5478 | 0.5535 | 0.5362 |
| | RADAR | 0.7154 | 0.7285 | 0.7835 | 0.6619 | 0.7875 |
| | ImBD | 0.8643 | 0.8837 | 0.8928 | 0.7596 | 0.8677 |
| | BiScope | 0.9450 | 0.9830 | 0.9900 | 0.8783 | 0.9860 |
| | Fast-DetectGPT | 0.9734 | 0.9879 | 0.9901 | 0.9488 | 0.9882 |
| | AdaDetectGPT | **0.9782** | **0.9893** | 0.9924 | **0.9553** | **0.9900** |
| | Relative | 18.0119 | 11.6202 | 22.7215 | 12.6288 | 15.7610 |

## F.2   Additional results on black-box setting

Table S8: AUC scores of various detectors to detect text generated by Gemini-2.5 across datasets.

| Method | XSum | Writing | Yelp | Essay | Avg. |
|---|---|---|---|---|---|
| RoBERTaBase | 0.5311 | 0.5202 | 0.5624 | 0.7279 | 0.5854 |
| RoBERTaLarge | 0.6583 | 0.5888 | 0.6029 | 0.8180 | 0.6845 |
| Likelihood | 0.7127 | 0.7547 | 0.6566 | 0.7565 | 0.7201 |
| Entropy | 0.5754 | 0.5088 | 0.6023 | 0.6038 | 0.5726 |
| LogRank | 0.6084 | 0.5743 | 0.6896 | 0.7504 | 0.6557 |
| LRR | 0.5960 | 0.5382 | 0.5580 | 0.6703 | 0.5906 |
| Binoculars | **0.8500** | 0.9453 | **0.9698** | 0.9908 | **0.9390** |
| RADAR | 0.8184 | 0.5152 | 0.6300 | 0.5891 | 0.6382 |
| BiScope | 0.7633 | 0.6800 | 0.7097 | 0.8167 | 0.7642 |
| Fast-DetectGPT | 0.8404 | 0.9443 | 0.9695 | 0.9914 | 0.9364 |
| AdaDetectGPT | 0.8432 | **0.9484** | 0.9644 | **0.9947** | 0.9377 |
| Relative (↑) | 1.7544 | 7.4163 | — | 37.8238 | 1.9916 |

### F.3 Computational analysis

In this part, we study the runtime for learning the witness function. From Table S9, the runtime of AdaDetectGPT is around 44 seconds and changes marginally with respect to $d$. This is because we can use a closed-form expression to learn a witness function. This time is nearly negligible compared with the time required to compute logits, which involves feeding tokens from multiple passages into LLMs. Furthermore, the training time when $n$ increases is shown in Table S10, and we can see that the runtime for training is typically no more than one minute.

Table S9: Runtime scale with $d$.

| $d$ | 4 | 8 | 12 | 16 | 20 |
|---|---|---|---|---|---|
| Runtime | 44.48 | 44.62 | 44.73 | 44.92 | 45.00 |

Table S10: Runtime (memory in parenthesis) scale with $n$. The runtime is measured in seconds and the memory is measured in GB.

| $n$ | 100 | 150 | 200 | 250 | 300 | 350 |
|---|---|---|---|---|---|---|
| Runtime (Memory) | 9.28(0.36) | 23.45(0.37) | 40.19(0.37) | 44.25(0.37) | 59.57(0.60) | 69.56(0.37) |

### F.4 Sensitivity analysis

Since AdaDetectGPT requires training a witness function, we examine three factors influencing its performance: (1) the size of the training set; (ii) tuning parameters for generating B-spline basis and (iii) distribution shift between training and test data.

**Robust to training data sizes**. We evaluate AdaDetectGPT across varying dataset sizes by setting $n_1 = n_2 \in \{100, 200, 300, 400, 500, 600\}$ for both human- and machine-generated texts. Figure S5 demonstrates that AdaDetectGPT clearly outperforms FastDetectGPT when sample size is large. This is expected because a larger sample size leads to a more accurate estimation of $w$. Notably, even with limited data $n_1 = n_2 = 100$, AdaDetectGPT maintains superior accuracy compared to baseline methods, though the performance gap decreases. These results highlight our method's effectiveness on learning the witness function.

**Insensitivity to tuning parameters**. B-spline relies on two critical tuning parameters: (i) the number of basis functions (n_base) and (ii) the maximum polynomial order. Our experiments fix one parameter while varying the other (with n_base=16 or order=2 as defaults). As shown in Figure S6 in Appendix, AdaDetectGPT achieves the highest AUC scores so long as n_base $\geq 4$. Besides, enlarging n_base improves the AUC of AdaDetectGPT although the improvement becomes marginal when n_base $\geq 16$. Figure S6 also shows that increasing the polynomial order from linear to quadratic visibly improve the performance; while increasing order from quadratic to cubic/quartic has a limited gain. Finally, even when the B-spline basis is set to a piecewise linear function, our method still outperform all baselines.

**Robust against distribution shift**. We create training datasets with different distributions from the test data by varying the number of human prompt tokens in machine-generated text. In contrast, for the test data, the number of human prompt tokens are fixed. As shown in Figure S7, AdaDetectGPT demonstrates high robustness to the distributional discrepancy between training and test data. It achieves the highest AUC across all experimental setup.

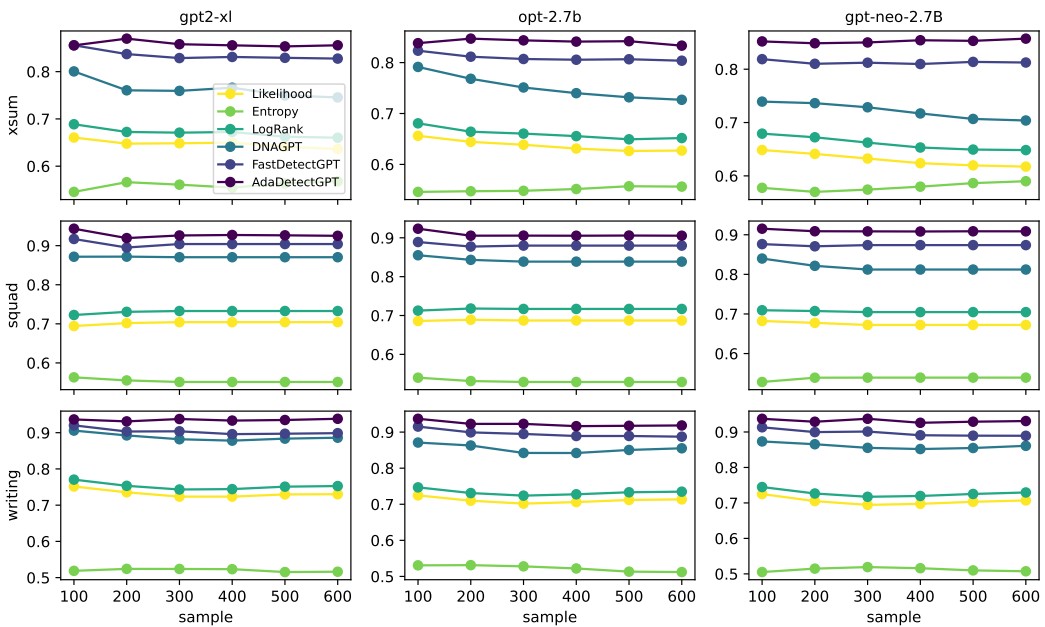

Figure S5: Classification accuracy versus the sample size for training $w$. We omit DetectGPT, NPR, and DNA in this experiments as they are time-consuming.

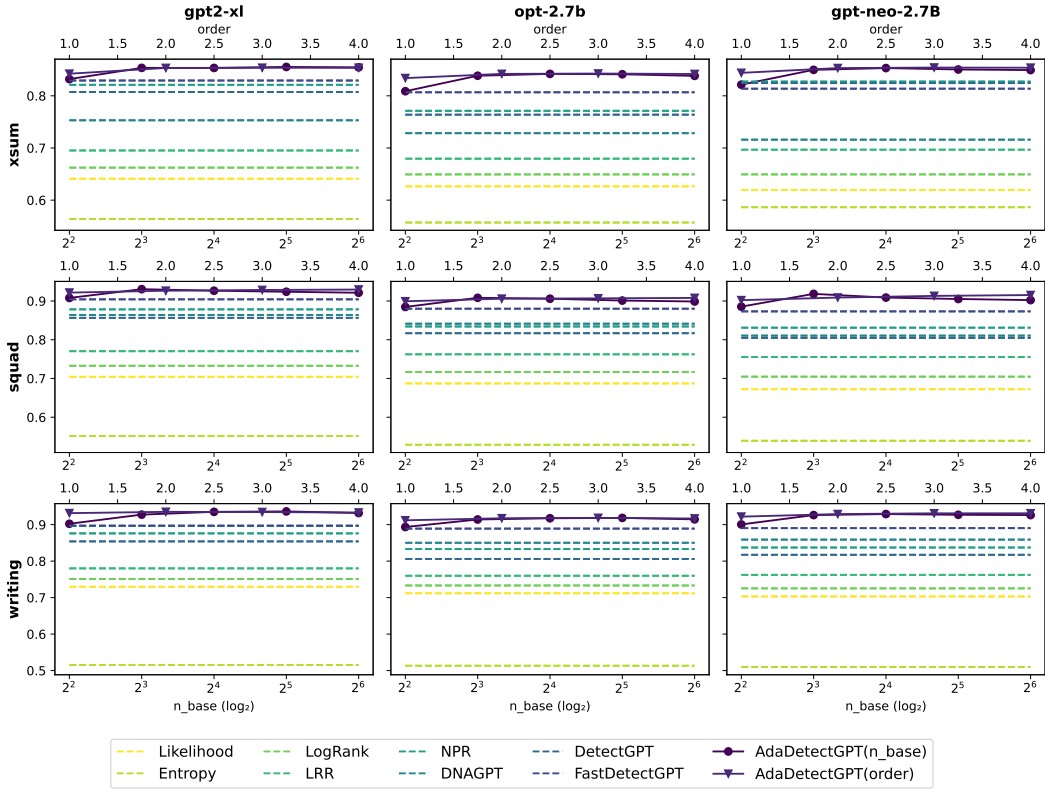

Figure S6: The classification accuracy of AdaDetectGPT and baseline methods. AdaDetect-GPT(`n_base`) present the AUC when the number of basis in B-spline increases as 4, 8, 16, 32, 64 (bottom $x$-axis); while AdaDetectGPT(`order`) shows the AUC when the maximum order of basis in B-spline increases from 1 to 4 (top $x$-axis). The AUC of baseline methods are presented by dash lines.

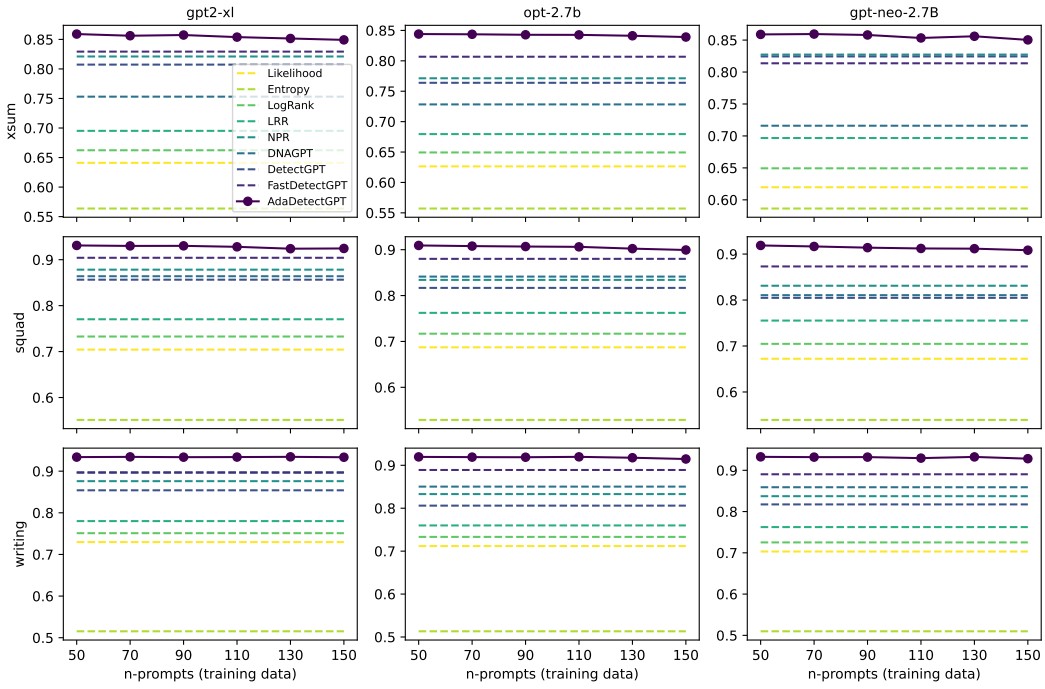

Figure S7: The classification accuracy of AdaDetectGPT when the number of human prompts changes. The AUC of baseline methods are presented by dash lines.

## F.5 Detecting open-source models in black-box setting

Table S11: Zero-shot detection accuracy on five source models under the black-box setting. †: use two surrogate models for sampling and scoring, where the sampling model is GPT-J while the scoring model is GPT-Neo.

| Dataset | Method | Source Model | | | | |
| --- | --- | --- | --- | --- | --- | --- |
| | | GPT-2 | OPT-2.7 | GPT-Neo | GPT-NeoX | Avg. |
| SQuAD | FastDetectGPT | 0.6181 | 0.6495 | 0.6230 | 0.6910 | 0.6813 |
| | AdaDetectGPT | 0.6920 | 0.7195 | 0.7382 | 0.7338 | 0.7460 |
| | Relative | 19.3570 | 19.9651 | 30.5609 | 13.8495 | 20.2957 |
| | FastDetectGPT† | 0.8145 | 0.8166 | 0.9220 | 0.7519 | 0.8188 |
| | AdaDetectGPT† | **0.8249** | **0.8308** | **0.9273** | **0.7609** | **0.8300** |
| | Relative | 5.6301 | 7.7245 | 6.7968 | 3.6121 | 6.2106 |
| Writing | FastDetectGPT | 0.7662 | 0.7918 | 0.7685 | 0.8022 | 0.8028 |
| | AdaDetectGPT | 0.8306 | 0.8529 | 0.8555 | **0.8587** | 0.8636 |
| | Relative | 27.5699 | 29.3365 | 37.6112 | 28.5350 | 30.8124 |
| | FastDetectGPT† | 0.8565 | 0.8497 | 0.9215 | 0.8182 | 0.8582 |
| | AdaDetectGPT† | **0.8780** | **0.8737** | **0.9386** | 0.8567 | **0.8849** |
| | Relative | 14.9666 | 15.9741 | 21.7742 | 21.1856 | 18.8023 |
| XSum | FastDetectGPT | 0.5919 | 0.6445 | 0.5718 | 0.6389 | 0.6468 |
| | AdaDetectGPT | 0.6795 | 0.7238 | 0.6879 | 0.7045 | 0.7261 |
| | Relative | 21.4569 | 22.2991 | 27.1129 | 18.1580 | 22.4439 |
| | FastDetectGPT† | 0.8145 | 0.8166 | 0.9220 | 0.7519 | 0.8188 |
| | AdaDetectGPT† | **0.8249** | **0.8308** | **0.9273** | **0.7609** | **0.8300** |
| | Relative | 9.8060 | 10.5637 | 10.2543 | 8.1057 | 11.1574 |

# G   Broader impact and limitation

AdaDetectGPT is a computationally and statistically efficient detector for machine-generated text, thus safeguarding AI systems against fake news, disinformation, and academic plagiarism.

Despite AdaDetectGPT's strong empirical performance in the black-box setting, its theoretical guarantees are mainly established in the white-box setting. Even when restricting to the white-box setting, LLM text generation often involves sampling parameters (e.g., `temperature` and `top_k`). Using different parameter values can cause the sampling distribution to deviate from that of the target model we aim to detect. This mismatch invalidates MCLT in practice. Fortunately, we observe that the shape of our statistic remains similar, but shifts toward a positive mean (see Figure S8), implying that FNR control under MCLT remains valid, although being more conservative.

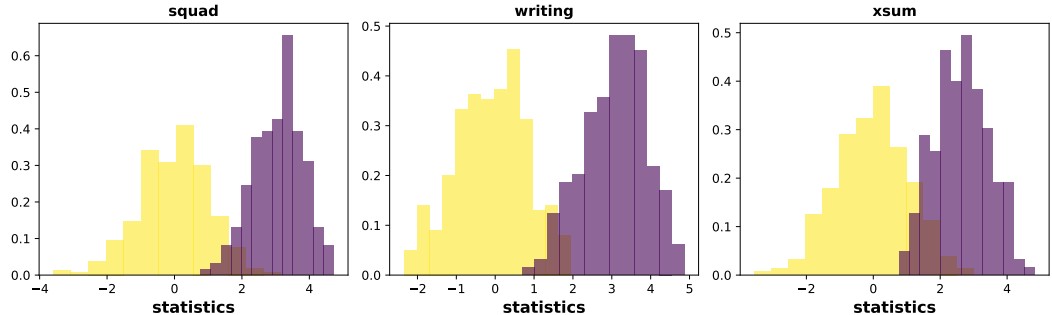

Figure S8: Histogram of our statistics in three datasets. Each panel visualizes the histogram in one dataset. The yellow histogram corresponds to the case when the sampled texts exactly follow the conditional probability of the source model, while purple histogram corresponds to text drawn with from a distribution with different sampling temperatures.

