# OpenReview forum: "AdaDetectGPT: Adaptive Detection of LLM-Generated Text with Statistical Guarantees"
_NeurIPS.cc/2025/Conference — NeurIPS 2025 poster_

### Official Review · Reviewer_6vsq · 2025-06-29

**Clarity:** 3
**Significance:** 3
**Originality:** 3
**Rating:** 5
**Confidence:** 3

**Summary:**

Most current logits-based detectors use statistics based on the log-probabilities of the observed text, computed using the distribution of a specific source LLM. However, relying only on log-probability signals can be sub-optimal. To address this, the authors propose AdaDetectGPT, a new classifier that adaptively learns a witness function from training data to improve detection performance. The method also provides statistical guarantees on performance metrics such as true positive rate, false positive rate, and others.

**Questions:**

Essentially, the proposed method aims to learn a witness function so that the scoring model can better capture the differences between AI- and human-written text.

However, this raises an important question: what if we directly fine-tune the scoring model using the same training dataset, without learning a separate witness function? This approach is exactly what ImBD (Imitate Before Detect) does—fine-tuning the detection model directly on machine- and human-written samples.

A comparison between AdaDetectGPT and this more straightforward fine-tuning baseline would be helpful, as it could clarify the actual benefits of introducing a learned witness function.

Ref: [AAAI 2025 oral] Imitate Before Detect: Aligning Machine Stylistic Preference for Machine-Revised Text Detection

**Ethical Concerns:**

["NO or VERY MINOR ethics concerns only"]

**Final Justification:**

I am accepting the paper mainly because of its promising results, the additional fair comparisons, and the experiments with models such as GPT‑4o, as well as the authors’ effort in training a variant of AdaDetectGPT. However, I did not carefully review the formulas and proofs in the paper, so I cannot guarantee their correctness, and therefore my confidence is only 3.

**Limitations:**

Yes

**Paper Formatting Concerns:**

There are no major formatting problems observed in the current version of the paper. However, I recommend that the authors add a conclusion section, which is helpful for summarizing the contributions and findings.

**Quality:**

3

**Strengths And Weaknesses:**

**Strengths**

The paper is well written and easy to follow.

The problem addressed is important. Learning a witness function from training data and adaptively selecting thresholds in zero-shot settings is highly relevant for practical applications.

The introduction and related work sections are clearly organized and provide a good overview of prior work in this area. However, I suggest the authors reduce the discussion on watermarking, since this paper focuses on passive detection, and watermarking belongs to a different category (active detection).

**Weaknesses**

1. According to the Fast-DetectGPT homepage, the best scoring models currently are falcon-7b and falcon-7b-instruct. Therefore, I suggest the authors re-run the comparisons between AdaDetectGPT and Fast-DetectGPT using these updated scoring models.
2. The main limitation of the paper is that the experimental evaluation is too limited. The models used in the experiments (GPT-2, OPT-2.7B, GPT-Neo) are quite outdated. To show practical usefulness, the authors should include more recent models.
- For open-source models, consider adding LLaMA-3, Qwen, etc.
- For black-box settings, the paper only evaluates GPT-3.5-turbo and GPT-4. However, these models are being phased out. The authors should consider adding newer systems like Claude, GPT-4o, or Gemini.
3. The paper does not evaluate under attack settings, which is important for robustness. This was also emphasized in the original Fast-DetectGPT paper. Including evaluation under adversarial or paraphrased inputs would strengthen the results.
4. Since AdaDetectGPT requires training, it should be compared primarily with other training-based methods. However, many of the baselines in the paper are zero-shot methods.
I recommend the authors include stronger and more recent baselines, such as:
- RADAR (NeurIPS 2023),
- ImBD (AAAI 2025),
- Text Fluoroscopy (EMNLP 2024),
- Biscope (NeurIPS 2024),
- Optional) GPTZero could also be included if resources allow, but the above four are more important.

5. The paper is missing a conclusion section, which would help summarize the contributions and findings clearly.

---

If the authors can address the points above—especially by improving the experimental setup—I would be happy to increase my score. Currently, the paper focuses mainly on theory and method design, but the limited evaluation makes it hard to fully assess the effectiveness of the proposed approach in practical scenarios.

---

> ### Author Rebuttal · Authors · 2025-07-30
>
> We wholeheartedly appreciate your insightful comments, which we will carefully incorporate should the paper be accepted. During the rebuttal, we conducted extensive empirical studies to further evaluate AdaDetectGPT under various settings, including:
>
> 1. the use of falcon-7b and falcon-7b-instruct as scoring models;
> 2. the detection of more recent LLMs;
> 3. adversarial attacks;
> 4. comparison against additional baseline methods.
>
> Results detailed below demonstrate that AdaDetectGPT consistently delivers the **most competitive performance** across all settings:
>
> > **Scoring models** (Weakness 1)
>
> Following your suggestion, we have employed falcon-7b as scoring models to compare AdaDetectGPT against FastDetectGPT. The corresponding results have been reported in **Table D2**. It can be seen that AdaDetectGPT continues to achieve the largest AUC in most cases.
>
> > **Adversarial attacks** (Weakness 3)
>
> We have conducted additional experiments to evaluate the robustness of our proposed AdaDetectGPT against **2 adversarial attacks**:
> - *paraphrasing*, where an LLM is instructed to rephrase human-written text, and
> - *decoherence*, where the coherence of LLM-generated text is intentionally reduced to avoid detection.
>
> These experiments were carried out across **3 datasets** and **3 target LLMs**, resulting in a total of **18** settings. Refer to **Table B2** in our Response to potential concern 6 to Reviewer aTfn for the detailed results.
>
> > **Additional baselines and more recent LLMs** (Weaknesses 2, 4 & Question)
>
> We have compared against **5** newly added baseline methods, including **RADAR**, **ImBD**, **Text Fluoroscopy**, **Biscope** (as you pointed out), as well as another recent detector **Binoculars**. These methods were evaluated using **3 recent open-sourced LLMs**, including  Qwen2.5, LLaMA3 and Mistral, across **5 benchmark datasets** (including our newly added Yelp, Essay and XSum, SQuAD, WritingPrompts), yielding a total of **15 scenarios**. We remark that Yelp and Essay are two common benchmark datasets employed in prior works as well (Zhang et al, 2015; Verma et al., NAACL 2024). Additionally, we also used another **3 closed-source models** GPT-4o, Claude and Gemini for comparison.
>
> The results are reported in **Tables D1-D2**. For open-source models (i.e., Qwen2.5, LLaMA3 and Mistral), AdaDetectGPT consistently achieves the highest AUC across nearly all scenarios, outperforming **5** newly added baselines such as **Binoculars**, **RADAR**, **ImBD**, **Text Fluoroscopy**, **Biscope**, as well as Fast-DetectGPT, Likelihood, Entropy, and LogRank considered in our main paper. It can be seen that the improvement over Fast-DetectGPT and Binoculars can reach up to 10% and 30%, respectively.
>
> For proprietary models (i.e., Claude, Gemini and GPT-4o), AdaDetectGPT again remains the most competitive across almost all scenarios, and the improvement over Fast-DetectGPT can reach up to 60% - 80%.
>
> **Table D1: Performance on three recent open-source models, i.e., Qwen2.5, LLaMA3, and Mistral. The “Relative” rows report the percentage improvement of AdaDetectGPT over Fast-DetectGPT.**
> |Data||XSum||||Writing||||Essay|||
> |-|-|-|-|-|-|-|-|-|-|-|-|-|
> |Method|Qwen2.5|Mistral|LLaMA3|Avg.|Qwen2.5|Mistral|LLaMA3|Avg.|Qwen2.5|Mistral|LLaMA3|Avg.|
> |Likelihood|0.6175|0.7409|0.8236|0.7273|0.7041|0.8643|0.8929|0.8204|0.6755|0.8667|0.9115|0.8179|
> |Entropy|0.5403|0.5290|0.5545|0.5413|0.5043|0.5420|0.5732|0.5398|0.5073|0.6052|0.5626|0.5584|
> |LogRank|0.6325|0.7270|0.8634|0.7410|0.7150|0.8446|0.9122|0.8239|0.6958|0.8467|0.9351|0.8258|
> |LRR|0.6368|0.6023|0.8946|0.7113|0.7180|0.6654|0.9161|0.7665|0.7248|0.6938|0.0000|0.4729|
> |NPR|0.7515|0.7823|0.7872|0.7737|0.8247|**0.8948**|0.7897|0.8364|0.7165|0.8741|0.0000|0.5302|
> |DetectGPT|0.7524|0.7821|0.6876|0.7407|0.8161|0.8724|0.6530|0.7805|0.7099|0.8672|0.0000|0.5257|
> |Binoculars|0.6297|0.7218|0.9546|0.7687|0.7578|0.8440|0.9845|0.8621|0.8018|0.8314|**0.9949**|0.8760|
> |TextFluoroscopy|0.5778|0.6210|0.5479|0.5822|0.5110|0.5555|0.5274|0.5313|0.5638|0.5127|0.5478|0.5414|
> |RADAR|0.6469|0.6518|0.7154|0.6714|0.6190|0.6537|0.7285|0.6671|0.6061|0.6292|0.7835|0.6729|
> |ImBD|0.6653|0.7683|0.8643|0.7659|0.6584|0.8391|0.8837|0.7938|0.7874|0.8631|0.8928|0.8477|
> |BiScope|0.6320|0.7320|0.9450|0.7697|0.6610|0.8740|0.9830|0.8393|0.6625|0.9000|0.9900|0.8508|
> |FastDetectGPT|0.7523|0.8922|0.9734|0.8726|0.8513|0.8151|0.9879|0.8848|0.8347|0.9052|0.9901|0.9100|
> |AdaDetectGPT|**0.7963**|**0.8944**|**0.9782**|**0.8896**|**0.8965**|0.8275|**0.9893**|**0.9044**|**0.8799**|**0.9069**|0.9924|**0.9264**|
> |Relative|17.7682|2.0423|18.0119|13.3479|30.3912|6.6718|11.6202|17.0494|27.3167|1.8332|22.7215|18.1975|
>
> |Data||SQuAD||||Yelp|||
> |-|-|-|-|-|-|-|-|-|
> |Method|Qwen2.5|Mistral|LLaMA3|Avg.|Qwen2.5|Mistral|LLaMA3|Avg.|
> |Likelihood|0.5183|0.7068|0.7071|0.6440|0.6793|0.7598|0.8915|0.7769|
> |Entropy|0.5232|0.5070|0.5047|0.5116|0.5236|0.5103|0.5010|0.5117|
> |LogRank|0.5166|0.7041|0.7422|0.6543|0.6943|0.7499|0.9146|0.7863|
> |FastDetectGPT|0.5016|0.8812|0.9488|0.7772|0.8465|0.8902|0.9882|0.9083|
> |Binoculars|0.6164|0.7258|0.9469|0.7631|0.7199|0.7502|0.9854|0.8185|
> |TextFluoroscopy|0.5383|0.5772|0.5535|0.5564|0.5060|0.5109|0.5362|0.5177|
> |RADAR|**0.6262**|0.6055|0.6619|0.6312|0.6276|0.6018|0.7875|0.6723|
> |ImBD|0.5168|0.8073|0.7596|0.6946|0.7392|0.7440|0.8677|0.7836|
> |BiScope|0.6250|0.7283|0.8783|0.7439|0.7050|0.7840|0.9860|0.8250|
> |AdaDetectGPT|0.6044|**0.8851**|**0.9553**|**0.8149**|**0.8915**|**0.9026**|**0.9900**|**0.9281**|
> |Relative|20.6431|3.3051|12.6288|16.9470|29.3165|11.2763|15.7610|21.5350|
>
> **Table D2: Performance on three closed-source LLMs, i.e., GPT-4o, Gemini-2.5; Claude-3.5**
> ||||GPT-4o|||||Gemini-2.5|||||Claude-3.5|||
> |-|-|-|-|-|-|-|-|-|-|-|-|-|-|-|-|
> ||XSum|Writing|Yelp|Essay|Avg.|XSum|Writing|Yelp|Essay|Avg.|XSum|Writing|Yelp|Essay|Avg.|
> RoBERTaBase|0.5141|0.5352|0.6029|0.5739|0.5565|0.5311|0.5202|0.5624|0.7279|0.5854|0.5206|0.5836|0.5630|0.5593|0.5566|
> RoBERTaLarge|0.5074|0.5827|0.5027|0.6575|0.5626|0.6583|0.6588|0.6029|0.8180|0.6845|0.5462|0.6149|0.5105|0.6063|0.5695|
> Likelihood|0.5194|0.7661|0.8425|0.7849|0.7282|0.6067|0.5885|0.7127|0.7547|0.6656|0.5023|0.6780|0.7502|0.7134|0.6610|
> Entropy|0.5397|0.7021|0.7291|0.6951|0.6665|0.5754|0.5028|0.6038|0.6818|0.5910|0.5965|0.6035|0.5944|0.6625|0.6142|
> LogRank|0.5123|0.7478|0.8259|0.7786|0.7161|0.6084|0.5743|0.6896|0.7504|0.6557|0.5109|0.6653|0.7244|0.7111|0.6529|=
> RADAR|**0.9580**|0.8046|0.8558|0.8394|0.8644|0.8184|0.5152|0.6300|0.5891|0.6382|**0.9187**|0.7264|0.8424|0.9152|0.8507|
> BiScope|0.8333|0.8733|0.9700|0.9600|0.9092|0.7633|0.6800|0.7967|0.8167|0.7642|0.8533|0.8800|0.8800|0.9567|0.8925|
> FastDetectGPT|0.6796|0.9165|0.9162|0.9199|0.8580|0.5768|0.7373|0.7419|0.7878|0.7109|0.7207|0.8466|0.9270|0.9579|0.8631|
> AdaDetectGPT|0.9072|**0.9611**|**0.9832**|**0.9841**|**0.9589**|**0.8432**|**0.9484**|**0.9644**|**0.9947**|**0.9377**|0.9176|**0.9400**|**0.9728**|**0.9610**|**0.9478**|
> Relative|71.0264|53.4451|80.0000|80.1332|71.0484|62.9470|80.3570|86.2087|97.4869|78.4355|70.4942|60.8752|62.6561|7.3840|61.9021|
>
> > **Reduce the discussion on watermarking** (Strengths 3)
>
> Thank you for pointing this out, we will follow your suggestion to reduce our discussion on watermarking.
>
> > **Missing conclusion** (Weakness 5)
>
> Due to the page limit, we did not include a conclusion section in the main paper. During the rebuttal, we have drafted a conclusion that we plan to include using the extra page should our paper be accepted. Refer to **Missing conclusion (Minor)** in our **Response to Reviewer UhoV**  for details.
>
> ---
>
> We hope our clarifications and newly-added experiments address your concerns. Let us know if you have any follow-up questions.

---

> > ### Comment · Reviewer_6vsq · 2025-08-03
> > **Experimental setup in your rebuttal**
> >
> > I am still somewhat confused about the experimental setup in your rebuttal. My understanding is that you used Falcon-7B/Falcon-7B-Instruct as the scoring model for Fast-DetectGPT, Binoculars, and AdaDetectGPT to ensure a fair comparison.
> >
> > In Table D2, is this indeed how you implemented Fast-DetectGPT and Binoculars?
> > Specifically, for GPT-4o on XSum, did Fast-DetectGPT (Falcon-7B/Falcon-7B-Instruct) achieve only 0.6796 AUROC, and for Gemini, only 0.5768?
> > And for Binoculars (Falcon-7B/Falcon-7B-Instruct) on GPT-4o XSum, was the AUROC only 0.8333?
> >
> > Could you clarify if my understanding is correct?

---

> ### Author Response · Authors · 2025-08-03
>
> Apologize for the confusion. We make some clarifications.
>
> First, the results for **Fast-DetectGPT** in Table D2 did use **Falcon-7B/Falcon-7B-Instruct** as the scoring model. To confirm this, **Table D3** summarizes the AUCs of Fast-DetectGPT when using both **GPT-J/Neo-2.7** and **Falcon-7B/Falcon-7B-Instruct** for scoring models to detect text generated by ChatGPT, GPT-4, GPT-4o, Gemini-2.5 and Claude-3.5. It can be seen that
>
> * The AUCs using GPT-J/Neo-2.7 for ChatGPT and GPT-4 match those reported in the original Fast-DetectGPT paper;
> * Switching to Falcon-7B/Falcon-7B-Instruct yields significantly improved AUCs, consistent with the findings reported on the Fast-DetectGPT website;
> * GPT-4o, Gemini 2.5, and Claude 3.5 appear to be more challenging to detect, as reflected by their lower AUCs. This likely explains the reduced performance of Fast-DetectGPT you  mentioned.
>
> **Table D3. AUCs of Fast-Detect on GPT-3.5-turbo, GPT-4, GPT-4o, Gemini-2.5, and Claude-3.5.**
> ||||ChatGPT||||GPT-4||
> |-|-|-|-|-|-|-|-|-|
> ||XSum|Writing|Pubmed|Avg.|XSum|Writing|Pubmed|Avg.|
> GPT-J/Neo-2.7|0.9907|0.9916|0.9021|0.9615|0.9067|0.9612|0.8503|0.9061|
> Falcon-7b/Falcon-7b-instruct|**0.9994**|**0.9969**|**0.9800**|**0.9921**|**0.9789**|**0.9968**|**0.9632**|**0.9796**|
>
> ||||GPT-4o|||||Gemini-2.5|||||Claude-3.5|||
> |-|-|-|-|-|-|-|-|-|-|-|-|-|-|-|-|
> ||XSum|Writing|Yelp|Essay|Avg.|XSum|Writing|Yelp|Essay|Avg.|XSum|Writing|Yelp|Essay|Avg.|
> GPT-J/Neo-2.7|0.5801|0.7862|0.8214|0.8039|0.7479|**0.6687**|0.6227|0.6012|0.5927|0.6213|0.6402|0.7231|0.8198|0.7619|0.7363|
> Falcon-7b/Falcon-7b-instruct|**0.6796**|**0.9165**|**0.9162**|**0.9199**|**0.8580**|0.5768|**0.7373**|**0.7419**|**0.7878**|**0.7109**|**0.7207**|**0.8466**|**0.9270**|**0.9579**|**0.8631**|
>
> Second, are you referring to **BiScope** as opposed to **Binoculars**? We did not implement Binoculars in Table D2 and the numbers you referred to correspond to the AUCs of BiScope. For **BiScope**, we used their default scoring model, **Llama-2-7B**. As shown in **Table D4** below, Llama-2-7B outperforms Falcon-7B in nearly all cases.
>
> **Table D4. AUCs of BiScope on recent closed-source LLMs when using Falcon-7b and Llama-2-7b.**
> ||||GPT-4o|||||Gemini-2.5|||||Claude-3.5|||
> |-|-|-|-|-|-|-|-|-|-|-|-|-|-|-|-|
> ||XSum|Writing|Yelp|Essay|Avg.|XSum|Writing|Yelp|Essay|Avg.|XSum|Writing|Yelp|Essay|Avg.|
> BiScope(Falcon-7b)|0.5867|0.7700|0.8133|0.7167|0.7217|0.5067|**0.6800**|0.6433|0.6000|0.6075|0.7300|0.8300|0.7533|0.9300|0.8108|
> BiScope(Llama-2-7b)|**0.8333**|**0.8733**|**0.9700**|**0.9600**|**0.9092**|**0.7633**|**0.6800**|**0.7967**|**0.8167**|**0.7642**|**0.8533**|**0.8800**|**0.8800**|**0.9567**|**0.8925**|
>
> Third, the AUCs of **AdaDetectGPT** in Table D2 were computed using **Gemma-9B/Gemma-9B-Instruct** as the scoring model. We acknowledge that this may result in an unfair comparison, as **Fast-DetectGPT** (based on your recommended Falcon-7b/Falcon-7b-Instruct), **BiScope** (based on its default Llama-2-7b) and other methods (based on Gemma-9B/Gemma-9B-Instruct) used different scoring models. To ensure fairness, we also report the AUCs of Fast-DetectGPT and BiScope using Gemma-9B/Gemma-9B-Instruct in **Table D5**. We also include **Binoculars** with Gemma-9B/Gemma-9B-Instruct as the scoring model in Table D5, as you mentioned this algorithm. We observe that:
>
> * Using Gemma-9B/Gemma-9B-Instruct improves the performance of Fast-DetectGPT, but downgrades the performance of BiScope (compared to its default Llama-2-7b);
> * AdaDetectGPT still generally outperforms Fast-DetectGPT, BiScope and Binoculars, with improvements reaching up to 37.8% over Fast-DetectGPT, though the margins over Fast-DetectGPT are smaller in some cases.
>
> **Table D5. AUCs of BiScope, Binoculars, FastDetect, and AdaDetectGPT on recent closed-source LLMs when all of them using Gemma-9b/Gemma-9b-instruct.**
> ||||GPT-4o|||||Gemini-2.5|||||Claude-3.5|||
> |-|-|-|-|-|-|-|-|-|-|-|-|-|-|-|-|
> ||XSum|Writing|Yelp|Essay|Avg.|XSum|Writing|Yelp|Essay|Avg.|XSum|Writing|Yelp|Essay|Avg.|
> BiScope(Gemma-9b)|0.8133|0.8433|0.8467|0.7633|0.8167|0.6533|0.7233|0.7533|0.6033|0.6833|0.6933|0.7567|0.6800|0.7167|0.7117|
> Binoculars|0.9022|0.9572|0.9840|0.9777|0.9552|**0.8500**|0.9453|0.9698|0.9908|**0.9390**|0.9012|0.9393|0.9752|0.9603|0.9440|
> Fast-DetectGP|0.9048|0.9588|**0.9847**|0.9800|0.9571|0.8404|0.9443|**0.9695**|0.9914|0.9364|0.9019|0.9361|**0.9768**|0.9608|0.9439|
> AdaDetectGPT|**0.9072**|**0.9611**|0.9832|**0.9841**|**0.9589**|0.8432|**0.9484**|0.9644|**0.9947**|0.9377|**0.9176**|**0.9400**|0.9728|**0.9610**|**0.9478**|
> Relative (Ada over Fast)|2.4288|5.6095|-|20.4444|4.2454|1.7544|7.4163|-|37.8238|1.9916|16.0326|6.0543|-|0.3405|6.9929|

---

> ### Author Response · Authors · 2025-08-03
>
> Finally, motivated by your earlier comment, we trained a variant of AdaDetectGPT that fine-tunes its scoring model by directly maximizing the proposed TNR lower bound -- rather than relying on linear function approximation as in our original approach. The results, also shown in **Table D6**, demonstrate much improved performance over **Fast-DetectGPT** and **Binoculars** using **Gemma-9B/Gemma-9B-Instruct** as the scoring model. Moreover, as reported in **Table D6**, this fine-tuned version outperforms **ImBD** by a large margin, highlighting the benefit of optimizing the proposed objective function.
>
> **Table D6. The results of ImBD, FastDetect and AdaDetectGPT (fine-tuned) on recent closed-source LLMs when all of them using Gemma-9b/Gemma-9b-instruct.**
> ||||GPT-4o|||||Gemini-2.5|||||Claude-3.5|||
> |-|-|-|-|-|-|-|-|-|-|-|-|-|-|-|-|
> ||XSum|Writing|Yelp|Essay|Avg.|XSum|Writing|Yelp|Essay|Avg.|XSum|Writing|Yelp|Essay|Avg.|
> |Fast-DetectGPT|0.9048|0.9588|0.9847|0.9800|0.9571|0.8404|0.9443|0.9695|0.9914|0.9364|0.9019|0.9361|0.9768|0.9608|0.9439|
> |ImBD|0.9402|0.9892|0.9928|0.9887|0.9777|0.7636|0.9626|0.9751|0.9970|0.9246|0.9096|**0.9667**|0.9752|0.9378|0.9473|
> |AdaDetectGPT(fine-tuned)|**1.0000**|**0.9999**|**1.0000**|**1.0000**|**1.0000**|**0.9991**|**0.9853**|**1.0000**|**1.0000**|**0.9961**|**0.9880**|0.8656|**0.9988**|**0.9993**|**0.9630**|
> |Relative(overFast-DetectGPT)|100.0000|99.8936|100.0000|100.0000|99.9843|99.7899|94.3998|100.0000|100.0000|98.6508|95.7189|12.3605|98.4161|98.4177|72.9460|
> |Relative(overImBD)|100.0000|99.1803|100.0000|100.0000|99.9004|99.6240|60.6187|100.0000|100.0000|94.8283|86.7748|-|95.3363|98.9286|29.6847|
>
> ---
>
> We hope our responses clarifies your confusions. Let us know if you have any follow-up comment.

---

> > ### Comment · Reviewer_6vsq · 2025-08-04
> >
> > I appreciate the additional experiments provided by the authors, especially the effort in training a variant of AdaDetectGPT.
> >
> > I would like to ask for clarification on the exact API parameters used when generating text with GPT‑4o. For example, what temperature was set, and which version of GPT‑4o was used (e.g., gpt‑4o‑latest or gpt‑4o‑2024‑05‑13)?
> >
> > Finally, I strongly suggest ensuring that the comparisons in the paper use the same scoring model across methods, as this would present a fairer and more convincing evaluation.

---

> > > ### Author Response · Authors · 2025-08-04
> > >
> > > Many thanks for the follow up and for your appreciation of our experiments!
> > >
> > > > Exact API parameters of GPT‑4o
> > >
> > > We use the current default version of GPT-4o provided by OpenAI (i.e., gpt-4o-2024-08-06). Following the the Fast-DetectGPT paper, the temperature parameter is set to 0.8 to encourage the generated text to be creatively diverse and less predictable.
> > >
> > > > Comparisons with the same scoring model
> > >
> > > We fully agree and plan to include the black-box results from **Table D5** -- where all methods use Gemma-9B/Gemma-9B-Instruct as the scoring model -- should the paper be accepted. When incorporating additional results into the paper, we will also ensure that the same scoring model is used across methods whenever feasible. For the white-box results presented in the main paper, all methods (including Fast-DetectGPT and AdaDetectGPT) already use the same scoring model.
> > >
> > > ---
> > >
> > > Once again, we sincerely appreciate your thoughtful feedback and active engagement during the discussion. Please let us know if you have any further comments. If we have adequately addressed your concerns, we would also deeply appreciate if you would consider adjusting your score.

---

> ### Comment · Reviewer_6vsq · 2025-08-04
>
> I appreciate the authors’ additional experiments, especially the effort in training a variant of AdaDetectGPT and ensuring a fairer comparison. In the original submission, my main concerns were that the experiments were not sufficiently comprehensive or fair (note: I did not carefully check the formulas and proofs), for instance, the generation models used were not sufficiently advanced, and the paper lacked a proper conclusion. However, these issues have been addressed in the rebuttal, which leads me to raise my score to 5.
>
> I encourage the authors to incorporate these additional experiments and provide more detailed descriptions of the experimental setup in the revised manuscript or the appendix. This should include all necessary details for reproducibility, such as the parameters of the generation model APIs.
>
> ---
>
> Supplement:
>
> I have currently resolved my doubts, so I have also raised the Quality score from 2 to 3.

---

> > ### Author Response · Authors · 2025-08-04
> >
> > Thank you for raising your score! We are very happy to hear that our experiments address your comments.
> >
> > Shall our paper be accepted, we plan to add results on the new datasets, models and baseline algorithms while ensuring fair comparisons across the baselines. We will also elaborate on the detailed settings of the generation LLMs, such as their version and temperature.
> >
> > Once again, many thanks for reviewing our paper and providing constructive comments!

---

> > > ### Comment · Reviewer_6vsq · 2025-08-06
> > >
> > > Thanks for the previous clarification, I am still somewhat confused about the experiments.
> > >
> > > I have noticed an inconsistency in the reported results for Fast‑DetectGPT.
> > >
> > > In the table provided to me, the AUCs of Fast‑DetectGPT are 0.6796 / 0.9165 / 0.9162 / 0.9199 / 0.8580, while your method reports 0.9072 / 0.9611 / 0.9832 / 0.9841 / 0.9589.
> > >
> > > However, in the version included in your response to Reviewer aTfn (Table B3. AUCs of various methods on recent closed‑source LLMs), the AUCs of Fast‑DetectGPT are 0.9048 / 0.9588 / 0.9847 / 0.9800 / 0.9571, whereas your method remains the same at 0.9072 / 0.9611 / 0.9832 / 0.9841 / 0.9589.
> > >
> > > Could you explain why the reported results of Fast‑DetectGPT differ so significantly between versions, while your method remains unchanged?

---

> > > > ### Author Response · Authors · 2025-08-06
> > > >
> > > > Thanks for raising this. The inconsistency arises from the use of different scoring models:
> > > >
> > > > * In **Table D2**, which we previously shared with you, Fast‑DetectGPT uses Falcon-7b/Falcon-7b-instruct as the scoring model (following your earlier suggestion);
> > > > * In **Table B3**, which we provided to Reviewer aTfn, Fast‑DetectGPT uses Gemma-9b/Gemma-9b-instruct instead.
> > > >
> > > > This difference explains the change in AUC scores for Fast-DetectGPT. To the contrary, our method uses Gemma-9b/Gemma-9b-instruct in both tables, which is why its AUC remains unchanged.
> > > >
> > > > To ensure fairness -- as per your most recent suggestion -- we have updated Fast‑DetectGPT to use Gemma-9B/Gemma-9B-Instruct, and the results are provided in **Table D5** of our response to your comment. As you can see, the results in **Table B3** and **Table D5** are consistent.
> > > >
> > > > ---
> > > >
> > > > We hope this clears up the confusion. Let us know if you have follow-up questions.

---

> > > > > ### Comment · Reviewer_6vsq · 2025-08-06
> > > > >
> > > > > Thank you for the clarification. In the final version of the paper, please explicitly indicate the specific scoring model used.

---

> > > > > > ### Author Response · Authors · 2025-08-06
> > > > > >
> > > > > > Will do. Thank you!!

---

### Official Review · Reviewer_UhoV · 2025-07-01

**Clarity:** 3
**Significance:** 2
**Originality:** 3
**Rating:** 4
**Confidence:** 3

**Summary:**

The paper targets the detection of machine-generated texts. It proposes AdaDetectGPT, where a witness function is adaptively learned from training data to enhance the performance of logits-based detectors. In the theoretical part, statistical guarantees on its true positive rate, false positive rate, true negative rate, and false negative rate are provided. In the empirical part, various source models and datasets are considered, and decent performance improvement is observed.

**Questions:**

See above

**Ethical Concerns:**

["NO or VERY MINOR ethics concerns only"]

**Final Justification:**

More intuitive discussion and more results are added during the rebuttal, which address most of my concerns.

**Limitations:**

yes

**Quality:**

3

**Strengths And Weaknesses:**

# Strengths
- The paper is well written, and the theoretical analyses are thorough.
- The idea of learning the witness function is reasonable and empirically effective.
- The empirical studies are satisfactory.

# Weaknesses
- The theoretical part is a little dense. It may be better to provide a more intuitive discussion to make the paper easier to read. For example, before introducing the witness function, provide an explanation about its motivation and basic concept.
- Can the authors provide more results on more advanced models like DeepSeek-R1 and GPT-4.1, using more realistic data samples beyond XSum, Writing, and PubMed?
- Considering that the relative performance improvement is not too significant (e.g, 10-20%), I wonder how the computational efficiency of AdaDetectGPT compares to Fast-DetectGPT.
- There is a lack of enough ablation studies of the behaviour of the proposed method.
- Minor: The paper does not contain a conclusion section in the main text.

---

> ### Author Rebuttal · Authors · 2025-07-30
>
> We sincerely thank you for your thoughtful feedback, which we will thoroughly integrate once the paper is accepted. Below, we provide comprehensive discussions and substantial empirical analyses to address all your comments.
>
> > Intuitive discussion about the witness function (Weakness 1)
>
> We provide an analytical example, beginning on line 174, which illustrates how classifiers based on token-wise sums of logits may suffer from a loss of power and how this loss of power can be overcome by transforming the logits using a function depending on the distribution of the machine and human generated text. In the paper we have borrowed the term "witness function" from the literature on integral probability metrics, where one constructs a distance on the space of distributions defined on some set $\mathcal{X}$ by choosing a class of functions $\mathcal{F}$ on $\mathcal{X}$ and defining the distance as
>
> $$D _{\mathcal{F}} (\mathbb{P}, \mathbb{Q}) = \sup _{f \in \mathcal{F}} |\mathbb{E} _{X \sim \mathbb{P}} f(X) - \mathbb{E} _{Y \sim \mathbb{Q}} f(Y)|. \tag{1}$$
>
> For any two distributions $\mathbb{P},\mathbb{Q}$ on $\mathcal{X}$, the function at which the right hand side of equation (1) attains its maximum is known as the "witness function" as it "bears witness to" the difference between the two distributions. Returning to AdaDetectGPT, we propose to learn our witness function $w(\bullet)$ by optimizing a lower bound on the true negative rate of the classifier (see Theorem 1) defined in equation (4) over a particular function class. The numerator of this lower bound has the form
>
> $$\sum _t [\mathbb{E} _{X _{<t}\sim p, \widetilde{X} _t \sim q_t} w(\log q _t(\widetilde{X} _t|X _{<t}))-\mathbb{E} _{X _{<t}\sim p, \widetilde{X} _t \sim p _t} w(\log q _t(\widetilde{X} _t|X _{<t}))],$$
>
> where each summand can be associated with the term on the right hand side of equation (1) evaluated at a particular $f(\bullet)$, if the expectations were replaced with conditional expectations. For this reason, we refer to the function appearing in the AdaDetectGPT classifier as a witness function. In the manuscript we briefly mention the connection to integral probability metrics on line 222, however we would be happy to provide a fuller discussion if needed.
>
> > Advanced models and more datasets (Weakness 2)
>
> Following your suggestions, we have considered more advanced models and datasets. Specifically, we added **six recent LLMs**, including three open-sourced LLMs (Qwen2.5, LLaMA3, and Mistral) and three closed-source models (GPT-4o, Claude and Gemini), across **five** benchmark datasets (including our newly added Yelp, Essay, XSum, SQuAD, and Writing). This yields a total of **30 new scenarios**. As can be seen from **Table D1** and **Table D2** in our **Response to Reviewer 6vsq**, the proposed AdaDetectGPT generally returns the highest AUC score, confirming the effectiveness of our proposal.
>
> > Improvement and Runtime (Weakness 3)
>
> In terms of improvement, we would like to clarify that AdaDetectGPT can improve upon FastDetectGPT by 25%-36% in the Writing dataset, as can be from Table 1 in the submitted paper. Additionally, during the rebuttal period, we found that AdaDetectGPT improves upon FastDetectGPT by over 60% -- 80% for detecting text generated by GPT-4o, Gemini-2.5; Claude-3.5 (see **Table D2** in our **Response to Reviewer 6vsq**), which is quite significant.
>
> Additionally, AdaDetectGPT is computationally efficient, as the average time for AdaDetectGPT to analyze each passage is around half a second—efficient enough to handle most GPT detection cases. Finally, we agree that AdaDetectGPT involves a witness function learning procedure that is not needed for FastDetectGPT. However, since we derived a closed-form for the witness function, the training procedure can be completed in one minute, as can be seen from **Table A3** and **Table A4** in our **Response to Reviewer cReh**.
>
> > Ablation study (Weakness 4)
>
> Our approach consists of two main components: (i) deriving a lower bound on the TNR to serve as the objective for adaptively learning the detector, and (ii) parameterizing the witness function using a linear function approximation, which reduces learning to solving a system of linear equations.
>
> In response to your concern, we conducted an ablation study to assess the impact of the second component. Specifically, instead of using a linear model, we fine-tuned the scoring model as suggested by Reviewer 6vsq. The resulting variant, referred to as AdaDetectGPT*, is compared with the original AdaDetectGPT in **Table C1** below.
>
> Overall, AdaDetectGPT* demonstrates improved performance over AdaDetectGPT. However, this comes at a significant computational cost: while AdaDetectGPT completes training in under one minute with a 0.6Gb GPU memory requirement (see **Table A3** and **Table A4** in our **Response to Reviewer cReh**), AdaDetectGPT* requires substantially more time and resources. Typically, the training time of AdaDetectGPT* ranges from five to eight minutes, and the required GPU memory ranges from 3–5Gb. Therefore, AdaDetectGPT is much more computationally attractive. Notice that AdaDetectGPT already outperforms all baseline methods in most scenarios. Thus, considering the trade-off between training cost and detection performance, the original AdaDetectGPT might offer a more practical solution.
>
> **Table C1. Ablation studies on the effect of the witness function (FastDetectGPT vs. AdaDetectGPT) and the learning method (AdaDetectGPT vs. AdaDetectGPT\* )**
> | Data | Method | Qwen2.5 | Mistral | LLaMA3 | Avg. |
> | - | - | - | - | - | - |
> | XSum | FastDetectGPT | 0.7523 | 0.8922 | 0.9734 | 0.8726 |
> | XSum | AdaDetectGPT* | 0.7710 | 0.9722 | 0.9975 | 0.9135 |
> | XSum | AdaDetectGPT | 0.7963 | 0.8944 | 0.9782 | 0.8896 |
> | Writing | FastDetectGPT | 0.8513 | 0.8151 | 0.9879 | 0.8848 |
> | Writing | AdaDetectGPT* | 0.8804 | 0.9786 | 0.9996 | 0.9529 |
> | Writing | AdaDetectGPT | 0.8965 | 0.8275 | 0.9893 | 0.9044 |
> | Essay | FastDetectGPT | 0.8347 | 0.9052 | 0.9901 | 0.9100 |
> | Essay | AdaDetectGPT* | 0.8484 | 0.9964 | 0.9989 | 0.9479 |
> | Essay | AdaDetectGPT | 0.8799 | 0.9069 | 0.9924 | 0.9264 |
> | SQuAD |FastDetectGPT | 0.5016 | 0.8812 | 0.9488 | 0.7772 |
> | SQuAD |AdaDetectGPT* | 0.5191 | 0.9791 | 0.9844 | 0.8275 |
> | SQuAD |AdaDetectGPT | 0.6044 | 0.8851 | 0.9553 | 0.8149 |
> | Yelp | FastDetectGPT | 0.8465 | 0.8902 | 0.9882 | 0.9083 |
> | Yelp | AdaDetectGPT* | 0.8604 | 0.9666 | 0.9999 | 0.9423 |
> | Yelp | AdaDetectGPT | 0.8915 | 0.9026 | 0.9900 | 0.9281 |
>
> > Missing conclusion (Minor)
>
> Due to the page limit, we did not include a conclusion section in the main paper. During the rebuttal, we have drafted a conclusion (see below) that we plan to include using the extra page should our paper be accepted:
>
> We propose AdaDetectGPT, an adaptive LLM detector that learns a witness function $w$ to boost the performance of existing logits-based detectors. A natural approach to learning $w$ is to maximize the TNR of the resulting detector for a fixed FNR level $\alpha$. However, the TNR is a highly complicated function of $\alpha$ and $w$, which makes the learned witness function inherently $\alpha$-dependent -- that is, it maximizes the TNR at a particular FNR level but does not guarantee optimality at other FNR levels. To address this, we derive a lower bound on the TNR and propose to learn $w$ by maximizing this lower bound. Notice that the lower bound separates the effects of $\alpha$ and $w$: the witness function $w$ affects the lower bound only through $T_w^{(2)*}$, which is independent of $\alpha$. Consequently, the witness function that maximizes this lower bound simultaneously maximizes it across all FNR levels.
>
> In our implementation, we opted to learn the witness function via a B-spline basis  due to the straightforward nature of the optimization (finding the optimal witness function boils down to solving a system of linear equations) and its  favorable theoretical properties (the estimation error can be shown to attain Stone's optimal convergence rate, Stone, 1982). The number of $B$-spline basis functions can be selected in a data driven way for instance via Lepski's method; see Lepski \& Spokoiny (1997).
>
> ----
>
> We hope our clarifications and newly-added experiments address your concerns. Let us know if you have any follow-up questions.

---

> ### Comment · Reviewer_UhoV · 2025-08-05
> **Thanks**
>
> I appreciate the authors' detailed response to my questions, which effectively address most of my concerns. As such, I will maintain my rating of 4.

---

> > ### Author Response · Authors · 2025-08-05
> >
> > Thank you for your reply. We are glad to hear that our rebuttal has addressed your concerns. We also greatly appreciate your instructive comments and feedback.

---

### Official Review · Reviewer_aTfn · 2025-07-02

**Clarity:** 3
**Significance:** 2
**Originality:** 2
**Rating:** 5
**Confidence:** 4

**Summary:**

This paper introduces a classifier based on a witness function from training data to enhance the performance of logits-based machine-text detectors. Experiments are conducted both in white-box (access to LLM logit) and black-box settings (different models at train and test time), across several different benchmarks. The approach consistently improves over the strongest baseline included (Fast-DetectGPT).

**Questions:**

* How robust is the proposed approach expected to be to paraphrasing attacks?
* Since the approach must be trained, how sensitive it is to potential distribution shifts from training to test time?
* Please provide a discussion of the limitations of the proposed approach.

**Ethical Concerns:**

["NO or VERY MINOR ethics concerns only"]

**Final Justification:**

The authors provide a strong rebuttal which addresses several of my main concerns (e.g., the need to know which LLM is being detected). As a result, I have increased my score.

**Limitations:**

There does appear to be any substantive discussion of limitations.

**Quality:**

3

**Strengths And Weaknesses:**

Strengths

* This paper tackles an important problem (machine-text detection).
* The proposed approach appears to be quite principled and theoretical results are presented to bolster the empirical claims.
* The paper considers a challenging setting where human writing is mixed with machine writing (“prompting an LLM with the first 120 tokens of the human-written text and requiring it to complete the text with up to 200 tokens.”).

Potential concerns
* Learning the witness function requires training data. How is performance impacted by distribution shifts from the training to test conditions (e.g., text from a different genre or language at test time)?
* While a number of baselines are considered, some notable methods are not, including Binoculars (https://arxiv.org/abs/2401.12070).
* There is no conclusion and limited discussion of the results. In particular, there is no discussion of limitations of the proposed approach.
* The paper focuses first on a “white-box setting where the source model we have is the same as the target model we wish to detect” -- is this a realistic setting? Given the number of LLMs and the potential for adversaries to fine-tune their own LLMs, what realistic applications of machine-text detection does this work address?
* The experiments in Table 2 address black-box setting, which partially addresses the above concern. However, is it possible that the choice of evaluation data (standard benchmark tasks) impacts the results? Specifically, could data contamination influence the performance of the proposed approach?
* The paper does not address robustness to adversarial attacks, such as paraphrasing attacks.
* While three different datasets are considered in the experiments, I would have liked to see more diversity to help bolster the empirical claims. For example, although it is good to focus on challenging mixed human-machine text settings, it would be good to confirm the results are consistent in settings of purely machine/human text as well.

---

> ### Author Rebuttal · Authors · 2025-07-30
>
> We wholeheartedly appreciate your insightful comments, which we will carefully incorporate should the paper be accepted. **In summary**, during the rebuttal, we have conducted extensive empirical studies to evaluate the proposed AdaDetectGPT under a wide range of settings to address your concerns, including:
>
> - (i) distributional shifts;
> - (ii) adversarial attacks;
> - (iii) comparisons with additional baseline methods; and
> - (iv) evaluations on more diverse datasets.
>
> Results detailed below demonstrate that AdaDetectGPT consistently delivers the most competitive performance across all settings.
>
> > **Distributional shifts** (Potential concern 1)
>
> AdaDetectGPT is robust to distributional shifts; see Figure S4 in the Appendix. This figure reports the AUC of AdaDetectGPT when the distribution of the test data differs from that of the training data used to learn the witness function. Specifically, during training, we vary the number of human prompt tokens in the LLM-generated text, whereas in the test data, the number of human prompt tokens is fixed. It can be seen that the AUC hardly changes as we vary the number of human prompt tokens in the training data. Additionally, our method achieves the highest AUC in all cases.
>
> > **Additional baselines** (Potential concern 2).
>
> We have compared against **5** newly added baseline methods, including **Binoculars** you pointed out, as well as four other recent detectors **RADAR**, **ImBD**, **Text Fluoroscopy** and **Biscope**. These methods were evaluated using **3 recent LLMs** (Qwen2.5, LLama3, Mistral) across **5 benchmark datasets** (including our newly added Yelp, Essay and Xsum, SQuAD, WritingPrompts). Refer to **Table D1** in our **Response to Reviewer 6vsq** for the detailed results. In particular, our AdaDetectGPT consistently outperforms **Binoculars** in most scenarios and the improvement can reach up to 60% (see the table below where we average the AUC across the LLMs).
>
> **Table B1. AUC scores of Binoculars and AdaDetectGPT and the relative improvement. "Relative" reports the percentage improvement of AdaDetectGPT over Binoculars.**
> | Data | XSum | SQuAD | Writing | Yelp | Essay |
> | - | - | - | - | - |  - |
> | Binoculars | 0.7687 | 0.7631 | 0.8621 | 0.8185 | 0.8760 |
> | AdaDetectGPT | 0.8896 | 0.8149 | 0.9044 | 0.9281 | 0.9264 |
> | Relative | 52.28% | 21.90% | 30.67% | 60.37% | 40.61% |
>
> > **Adversarial attacks** (Potential concern 6)
>
> We have conducted additional experiments to evaluate the robustness of our proposed AdaDetectGPT against **2** adversarial attacks: (i) paraphrasing, where an LLM is instructed to rephrase human-written text, and (ii) decoherence, where the coherence LLM-generated text is intentionally reduced to avoid detection. These experiments were carried out across **3** datasets and **3** target LLMs, resulting in a total of **18** settings. Both adversarial attacks were implemented following Bao, et al., ICLR2024.
>
> **Table B2. Detection of LLM texts under attack.**
> | | | paraphrasing | | | |decoherence | | |
> | - | - | - | - | - | - | - | - | - |
> | | Xsum | Writing | PubMed | Avg. | Xsum | Writing | PubMed | Avg. |
> | FastDetectGPT(GPT-J/GPT-2) | 0.9178 | **0.9137** | 0.7944 | 0.8753 | 0.7884 | 0.9595 | 0.7870 | 0.8449 |
> | AdaDetectGPT(GPT-J/GPT-2) | **0.9225** | 0.9121 | **0.8029** | **0.8792** | **0.8765** | **0.9597** | **0.8284** | **0.8882** |
> | FastDetectGPT(GPT-J/Neo-2.7) | 0.9602 | **0.9185** | 0.7310 | 0.8699 | 0.8579 | 0.9701 | 0.7609 | 0.8630 |
> | AdaDetectGPT(GPT-J/Neo-2.7) | **0.9623** | 0.9181 | **0.7587** | **0.8797** | **0.9230** | **0.9704** | **0.8124** | **0.9019** |
> | FastDetectGPT(GPT-J/GPT-J) | 0.9537 | **0.9458** | 0.7041 | 0.8679 | 0.8836 | **0.9869** | 0.7550 | 0.8752 |
> | AdaDetectGPT(GPT-J/GPT-J) | **0.9587** | 0.9449 | **0.7308** | **0.8781** | **0.9336** | 0.9864 | **0.8008** | **0.9070** |
>
> Results are reported in the table above. It can be seen that As shown, AdaDetectGPT consistently achieves higher AUCs than Fast-DetectGPT in most of the 18 scenarios. And the improvement reaches up to 10\% for paraphrasing and up to 85\% for decoherence. These results suggest that AdaDetectGPT attains more robust performance against adversarial attacks.
>
> > **More diverse datasets** (Potential concern 7)
>
> We have conducted two analyses to address your comment. First, we have included two additional datasets -- **Yelp** and **Essay** -- in addition to XSum, SQuAD, WritingPrompts and found AdaDetectGPT remains the most competitive detector on both datasets. Refer to **Table D1 and Table D2** in our **Response to Reviewer 6vsq** for the detailed results. These two datasets have been employed in prior works as well (Mao, et al., ICLR2024; Verma, et al., NAACL2024).
>
> Second, our black-box settings use purely human/machine text, where the machine-generated text consists solely of outputs produced by LLMs. As can be seen from **Table D2**  in our **Response to Reviewer 6vsq**, AdaDetectGPT achieves the most competitive performance among state-of-the-art detectors.
>
> > **Missing conclusion** (Potential concern 3).
>
> Due to the page limit, we did not include a conclusion section in the main paper. During the rebuttal, we have drafted a conclusion that we plan to include using the extra page should our paper be accepted. The conclusion summarizes our proposed procedure and discusses the data contamination issue you highlighted (see our next response). Refer to **Missing conclusion (Minor)** in our **Response to Reviewer UhoV** for details.
>
> > **Realistic applications & data contamination** (Potential concerns 4 & 5).
>
> Your comment raises three important questions: (i) the realism of white-box settings; (ii) the potential issue of data contamination; (iii) the target application considered in this work. We address each of these points in detail below.
>
> - *Realism of white-box settings*.
>
> First, white-box settings are particularly relevant in scenarios where the goal is to detect outputs from open-source LLMs such as Qwen, LLaMA3, and Mistral. Note that we have conducted experiments on these models and found AdaDetectGPT continues to perform well; see **Table D1** in our **Response to Reviewer 6vsq** for detailed results.
>
> Second, we clarify that while the white-box setting is used to motivate and develop our methodology, AdaDetectGPT is also applicable in black-box settings. As you pointed out, our main paper already includes some black-box experiments, which we are glad to hear have partially addressed your concern. To further demonstrate AdaDetectGPT's usefulness, we have conducted additional black-box experiments during the rebuttal on other widely-used closed-source LLMs Claude and Gemini. AdaDetectGPT again attains the most competitive performance -- see our **Table D2** in **Response to Reviewer 6vsq**.
>
> - *Data contamination*.
>
> We fully agree that data contamination is a critical issue in evaluating LLM-related tasks. In particular, when evaluation data is contained in the training dataset, it can lead to overly optimistic estimates of the proposed algorithm's performance. We have carefully reviewed our evaluation protocol and did not find any such contamination.
>
> In our setup, training data is used for learning the witness function. As mentioned in Section B.1 of the Appendix, we employ cross-validation to separate training and testing datasets. For instance, in our main paper, we use three benchmark datasets: XSum, SQuAD, and WritingPrompts. When testing on one dataset (e.g., XSum), the witness function is trained using the other two datasets (e.g., SQuAD and WritingPrompts). Notice that these datasets are semantically and topically different -- XSum provides news summaries, SQuAD involves Wikipedia-based question answering, and WritingPrompts consists of creative human-written stories. This further reduces any risk of data leakage between training and testing.
>
> - *Target application*.
>
> This work focuses on determining whether a piece of text was generated by a user-specified LLM -- either open-source (white-box) or closed-source (black-box). Your comment highlights two practical challenges.
>
> First, given the vast and growing number of LLMs, it is impractical to detect outputs from all possible models. Therefore, we focus on a set of widely-used and popular LLMs. In fact, following the submission of our paper, we have developed a private website hosting our detector. Due to NeurIPS policy, we are unable to release its link at this stage. However, on the website, users can select from a list of supported target LLMs to check whether a given text was generated by any of them (including a “union” option to check if it came from any of the listed models). We plan to regularly update the set of supported models to reflect the evolving popularity of LLMs -- for instance, by removing outdated ones and incorporating new, widely adopted models.
>
> Second, you raise the important point that adversaries might fine-tune LLMs specifically to fool detectors. We acknowledge this is a challenging and largely open problem. Nonetheless, as we have discussed in our Response 3, our method can still be competitive in certain adversarial scenarios (e.g., paraphrasing and decoherence).
>
>
> ---
>
> We hope our clarifications and newly-added experiments address your concerns. Let us know if you have any follow-up questions.

---

> > ### Comment · Reviewer_aTfn · 2025-08-05
> >
> > Thanks for the response! This addresses some of my concerns. However, I'm still somewhat concerned about the need to know which LLM you're trying to detect, which seems like an important limitation of the approach. Perhaps the authors could offer a more substantial rebuttal to this concern (while I appreciate the release of a demo, this doesn't actually respond to the technical concern).

---

> ### Author Response · Authors · 2025-08-06
>
> Many thanks for the follow-up. Following prior works (e.g., Gehrmann, et al., ACL2019 and Bao, et al., ICLR2024), we focus on the setting with a specified target LLM. However, our approach can be easily extended to the case without specifying the target LLM, and we appreciate the opportunity to clarify this point.
>
> As mentioned in our rebuttal, we provide a "union" option that enables users to determine whether a text was generated by _any_ of several popular models (e.g., GPT, Gemini, or Claude), without the need to specify which particular model it comes from. We have also conducted preliminary theoretical and empirical studies, which show that even without specifying a target model, the resulting detector achieves performance that is comparable to that of our original detector tailored to a specific model. We refer to such a desirable property as the **oracle property**, meaning that the method works as well as if **the target model were known in advance**.
>
> **Methodology**. To elaborate our methodology, let $q^{(i)}$ denote the $i$-th target model in the list. Our goal is to detect whether a given text was generated by any model in the set $\{ q^{(i)} \}_i$. Toward that end, for each $q^{(i)}$, we apply our procedure to learn a witness function $w^{(i)}$, which is then used to construct our detection statistic denoted by $T^{(i)}$. Notice that each $T^{(i)}$ is applied to detect text generated by its target $q^{(i)}$. To detect any model in the set, we propose **a simple extension: take the maximum over all statistics**, i.e., $\max_i T^{(i)}$. We refer to this extension as **AdaDetectGPT (Max)**.
>
> **Experiment**. To evaluate performance, we conduct an additional experiment under the black-box setting described in our rebuttal. The goal is to detect whether a given text was generated by _any_ of the following three popular models: GPT-4o, Gemini-2.5, or Claude-3.5. We compare the performance of **AdaDetectGPT (Max)** — our extension for detecting across multiple models - against the original **AdaDetectGPT** and other baselines (e.g., **Fast-DetectGPT**) that assume prior knowledge of the specific target model used to generate the text. Results are presented in **Table B3** below.
>
> It can be seen that **AdaDetectGPT (Max)** achieves comparable performance to **AdaDetectGPT**, which knows the target model in advance. In some cases, it even outperforms AdaDetectGPT. This empirically verifies the oracle property of AdaDetectGPT (Max). Moreover, it generally outperforms other baseline methods that require prior knowledge of the specific target model used to generate the text.
>
> **Table B3. AUCs of various methods on recent closed-source LLMs. The best one is bold and the second one is italic.**
> ||||GPT-4o|||||Gemini-2.5|||||Claude-3.5|||
> |-|-|-|-|-|-|-|-|-|-|-|-|-|-|-|-|
> ||XSum|Writing|Yelp|Essay|Avg.|XSum|Writing|Yelp|Essay|Avg.|XSum|Writing|Yelp|Essay|Avg.|
> RoBERTaBase|0.5141|0.5352|0.6029|0.5739|0.5565|0.5311|0.5202|0.5624|0.7279|0.5854|0.5206|0.5836|0.5630|0.5593|0.5566|
> RoBERTaLarge|0.5074|0.5827|0.5027|0.6575|0.5626|0.6583|0.6588|0.6029|0.8180|0.6845|0.5462|0.6149|0.5105|0.6063|0.5695|
> Likelihood|0.5194|0.7661|0.8425|0.7849|0.7282|0.6067|0.5885|0.7127|0.7547|0.6656|0.5023|0.6780|0.7502|0.7134|0.6610|
> Entropy|0.5397|0.7021|0.7291|0.6951|0.6665|0.5754|0.5028|0.6038|0.6818|0.5910|0.5965|0.6035|0.5944|0.6625|0.6142|
> LogRank|0.5123|0.7478|0.8259|0.7786|0.7161|0.6084|0.5743|0.6896|0.7504|0.6557|0.5109|0.6653|0.7244|0.7111|0.6529|
> RADAR|**0.9580**|0.8046|0.8558|0.8394|0.8644|0.8184|0.5152|0.6300|0.5891|0.6382|**0.9187**|0.7264|0.8424|0.9152|0.8507|
> BiScope|0.8333|0.8733|0.9700|0.9600|0.9092|0.7633|0.6800|0.7967|0.8167|0.7642|0.8533|0.8800|0.8800|0.9567|0.8925|
> Fast-DetectGPT|0.9048|0.9588|**0.9847**|0.9800|0.9571|0.8404|0.9443|0.9695|**0.9914**|0.9364|0.9019|0.9361|**0.9768**|_0.9608_|0.9439|
> AdaDetectGPT|0.9072|**0.9611**|0.9832|**0.9841**|**0.9589**|**0.8432**|_0.9484_|_0.9644_|_0.9947_|**0.9377**|*0.9176*|**0.9400**|*0.9728*|**0.9610**|**0.9478**|
> AdaDetectGPT(Max)|_0.9075_|_0.9610_|_0.9833_|**0.9839**|**0.9589**|_0.8412_|**0.9489**|0.9632|**0.9953**|_0.9371_|0.9163|_0.9391_|0.9721|0.9604|_0.9470_|
> Relative(AdaDetectGPT over Fast-DetectGPT)|2.7557|5.2859|-|19.5556|4.2713|0.4734|8.3732|-|45.5959|1.2055|14.6739|4.5929|-|-|5.4279|

---

> ### Author Response · Authors · 2025-08-06
>
> *Theory*. To provide intuition for why the oracle property may hold, we consider two illustrative cases, leaving a rigorous proof to future work.
>
> - **Case 1: All $ q^{(i)}$ are similar**. In this case, the learned witness functions $w^{(i)}$ tend to be similar as well, resulting in detection statistics $T^{(i)}$ being similar across models. Consequently, taking the maximum over $T^{(i)}$ yields a statistic that behaves similarly to each individual $T^{(i)}$. As such, the FNR and TNR of $\max_i T^{(i)}$ become very similar to each $T^{(i)}$, justifying the oracle property in this case.
>
> - **Case 2: All $q^{(i)}$ differ substantially**. Suppose the text is generated by the $i$-th model. In this case, the statistic $T^{(i)}$, computed using the correctly specified model, is expected to dominate the others. Therefore, taking the maximum over all $T^{(i)}$ effectively selects the oracle statistic $T^{(i)}$, leading to similar FNR/TNR as the model-specific detector. This again supports the oracle property.
>
> In more general cases where some $q^{(i)}$ are similar while others differ substantially, similar arguments can be applied to show the taking the maximum over $T^{(i)}$ tends to approximate the oracle statistic generated under the true model $T^{(i)}$.
>
> ---
>
> We hope our responses address your comment. Let us know if you have any follow-up question.

---

### Official Review · Reviewer_cReh · 2025-07-03

**Clarity:** 2
**Significance:** 3
**Originality:** 3
**Rating:** 4
**Confidence:** 3

**Summary:**

This paper presents AdaDetectGPT, a novel approach for detecting LLM-generated text that enhances existing logits-based detection methods by adaptively learning a witness function from training data. The core idea is to transform raw log probabilities through an optimized witness function rather than using them directly, as done in previous methods like DetectGPT and Fast-DetectGPT. The authors derive a lower bound on the true negative rate (TNR) and learn the witness function by maximizing this bound, which involves solving a simple linear system when using B-spline basis functions. The method provides statistical guarantees for all four key classification metrics (TPR, FPR, TNR, FNR) based on the martingale central limit theorem, and includes a principled threshold selection mechanism for controlling the false negative rate. Extensive experiments across multiple datasets (SQuAD, WritingPrompts, XSum) and language models demonstrate substantial improvements over state-of-the-art baselines, with AUC gains of 12.5%-37% in white-box settings and up to 58% in black-box settings, while the approach is extended to handle black-box scenarios by using multiple source LLMs and taking the maximum statistic across them.

**Questions:**

- How does AdaDetectGPT's performance scale with increasingly sophisticated LLMs and potential adversarial attacks? Could you evaluate on more recent and diverse models (e.g., Qwen2.5, LLaMA3, Claude, Gemini) and discuss robustness against paraphrasing or other evasion techniques commonly used to fool detection systems?

- While the paper claims computational efficiency, could you provide detailed runtime comparisons and memory usage analysis, especially for the witness function learning phase? How does the training time scale with dataset size n and dimension d?

- Why specifically choose B-spline basis functions for the witness function parametrization? Could you provide theoretical or empirical comparison with other basis functions (e.g., polynomial, Fourier, or neural network-based)? Additionally, what principled guidance can you offer for selecting the dimension d of the feature mapping φ, especially given its impact on the TNR bound in Theorem 3?

 - Could you provide more rigorous verification or empirical validation of the equal variance condition and stochastic dominance condition mentioned in Theorem 1? These conditions are crucial for the theoretical guarantees, but their practical satisfaction seems under-explored. Please include experimental analysis showing when these conditions hold in real scenarios and how violations might affect performance.

**Ethical Concerns:**

["NO or VERY MINOR ethics concerns only"]

**Final Justification:**

This paper is interesting, and the authors have addressed the vast majority of my concerns during the rebuttal phase. The relaxed proportionality condition (Equation 1) with empirical validation (Table A1) and normality tests (Table A2)  strengthen the theoretical foundation by satisfying MCLT regularity conditions, while experiments across 6 LLMs and robustness against evasion techniques substantially enhance empirical validation. The B-spline theoretical justification with error bounds (Equation 2) and principled dimension selection guidelines provide valuable theoretical rigor, and runtime analysis demonstrates the practical efficiency advantages of this computationally feasible approach. However, since the authors did not include the conclusion in the main text, this represents a fairly significant issue, so I am inclined toward a weak accept.

**Limitations:**

Yes

**Paper Formatting Concerns:**

This paper lack of Conclusion Section

**Quality:**

2

**Strengths And Weaknesses:**

**Strengths:**

- AdaDetectGPT introduces an innovative approach by adaptively learning a witness function from training data to enhance logits-based detectors, rather than relying solely on raw log probabilities. This represents a significant methodological advancement that bridges statistics-based and ML-based detection methods.

- The method achieves substantial improvements over state-of-the-art baselines, with AUC improvements ranging from 12.5% to 37% in white-box settings and up to 58% in black-box settings. These consistent improvements across multiple datasets and language models demonstrate the effectiveness of the approach.

- Unlike existing logits-based detectors that generally lack systematic statistical analysis, AdaDetectGPT provides finite-sample error bounds for all four key classification metrics (TPR, FPR, TNR, FNR), offering theoretical rigor and reliability guarantees for practical deployment.


**Weaknesses:**

- A conclusion section would be a valuable addition to the paper.

- The paper relies on the equal variance condition and stochastic dominance condition, while the application of the Markov central limit theorem requires relatively strong regularity conditions. The verification of these conditions in the paper appears to be somewhat insufficient.

- The TNR lower bound in Theorem 1 depends on the choice of α. While the authors claim to have eliminated this dependency through bound optimization, it seems that the issue has essentially been shifted to the selection of the witness function.

- The choice of B-spline basis functions may benefit from additional theoretical justification, and there appears to be limited guidance regarding the selection of dimension d.

- The evaluation could potentially be strengthened by including a more diverse range of target LLMs, such as recent open-source models like Qwen2.5, LLaMA3, and Mistral, as well as proprietary models like Claude and Gemini. The dataset coverage is also somewhat limited, with only 3 datasets included.

---

> ### Author Rebuttal · Authors · 2025-07-30
>
> We wholeheartedly appreciate your insightful comments and suggestions, which we will carefully incorporate should the paper be accepted. During the rebuttal, we have conducted extensive theoretical and empirical investigations to address all your concerns.
>
> > **Validation of technical assumptions** (Weakness 2 & Question 4)
>
> Excellent comment! Your comment is concerned with the validity of three conditions: (i) the equal variance condition; (ii) the stochastic dominance condition; (iii) the regularity condition for the MCLT to hold.
>
> We did not elaborate on these assumptions in the main paper, and we appreciate the opportunity to clarify and discuss them in response to your concerns. In summary:
> 1. During the rebuttal, we have conducted both theoretical and empirical investigations and found that the equal variance condition can be relaxed to a much weaker proportionality condition (see Equation (1) below), which is likely to hold empirically (see **Table A1** below).
> 2. The stochastic dominance condition was introduced to derive an alternative, tighter lower bound than the one presented in Theorem 1. It is not required for the validity of Theorem 1 itself.
> 3. We have conducted several normality tests on our proposed statistic. Results suggest that in almost all cases, our statistic passed these normality tests (see **Table A2**), indicating the regularity conditions required for the MCLT are likely met in practical scenarios.
>
> We next elaborate on these points.
>
> First, the equal variance condition imposed in Theorem 1 can be relaxed. Specifically, our theoretical investigations during the rebuttal reveal that it is not necessary for the two variances $\sigma_{q,L}^2$ and $\sigma_{p,L}^2$ to be asymptotically equivalent in probability. Rather, it suffices to require their ratio to converge to some positive constant in probability, i.e.,
> $$
> \frac{\sigma_{q,L}}{\sigma_{p,L}} \stackrel{P}{\to} K_0. \quad (1)
> $$
> Since $K_0$ need not be 1, this proportionality condition is considerably weaker than the equal variance assumption and is more likely to hold in practice. Under this relaxed condition, following nearly identical arguments to those in Theorem 1, we can show that TNR is asymptotically lower bounded by:
> $$
> \min \\{1-\Phi(K_0 z_\alpha),\ \Phi(K_0 z_\alpha)+\phi(K_0 z_\alpha) K_0 T_w^{(2*)}\\}.
> $$
>
> This lower bound differs from the one in Theorem 1 due to the change in assumptions. However, since $\phi(K_0 z_\alpha)$ depends solely on $\alpha$ (not on the witness function $w$), our main conclusion — maximizing the lower bound is equivalent to maximizing $T_w^{(2*)}$ — remains valid. Thus, the proposed methodology remains theoretically sound.
>
> Empirical results further support the validity of this assumption: the sample mean of the ratio remains nearly constant as a function of $L$, while its variance (in parentheses) approaches zero as $L\to \infty$ across all three datasets (see **Table A1**), suggesting that this condition is practical.
>
> Second, for the stochastic dominance condition, we clarify that its primary purpose was to apply Lemma S3 in deriving an alternative tighter lower bound (detailed in Remark 1 after the proof of Theorem 1 in the supplementary material). This condition is therefore not essential for the validity of Theorem 1 and can be safely omitted without affecting our main conclusion.
>
> Third, we applied **2 standard hypothesis tests** — the Kolmogorov–Smirnov (KS) test and the Shapiro–Wilk (SW) test — to evaluate whether our proposed statistic follows a normal distribution. These tests were conducted across **3 benchmark datasets** and **2 LLM source distributions**, yielding a total of **12 p-values**. As shown in **Table A2**, almost all p-values exceed 0.1 (most by a large margin), indicating that our test statistic passes normality tests in most cases, providing strong empirical support for the validity of MCLT regularity conditions.
>
> **Table A1: Sample mean and variance (in parentheses) of the ratio evaluated on 3 datasets as $L$ increases**
> L|100|150|200|250|300|350
> -|-|-|-|-|-|-
> XSum|1.09(0.12)|1.07(0.09)|1.06(0.08)|1.04(0.07)|1.01(0.06)|0.99(0.05)
> SQuAD|1.03(0.10)|1.02(0.07)|1.02(0.06)|1.02(0.05)|1.01(0.05)|1.03(0.04)
> Writing|1.10(0.06)|1.09(0.05)|1.09(0.04)|1.08(0.03)|1.07(0.03)|1.05(0.03)
>
> **Table A2: $p$-values of KS and SW tests across 3 datasets and 2 source LLMs**
> | |Test|Xsum|SQuAD|Writing
> |-|-|-|-|-
> |GPT-Neo|KS|0.72|0.54|0.18
> |GPT-Neo|SW|0.50|0.65|0.89
> |GPT-2|KS|0.10|0.52|0.28
> |GPT-2|SW|0.37|0.026|0.14
>
> > **Experiments on more datasets and LLM models** (Weakness 5 & Question 1)
>
> Following your suggestion, we have conducted experiments during the rebuttal to compare AdaDetectGPT against existing detectors across **6 recent LLMs** and **5 datasets**. Due to space limit, refer to our **Table D1 & Table D2** in **Response to Reviewer 6vsq** for details.
>
> > **Experiments against evasion techniques** (Question 1)
>
> Following your suggestion, we have conducted additional experiments to evaluate the robustness of AdaDetectGPT against **two** evasion techniques:  (i) paraphrasing and  (ii) decoherence across 3 datasets and 3 target LLMs, resulting in a total of **18** settings. Refer to **Table B2** in our **Response to potential concern 6 to Reviewer aTfn** for detailed results.
>
> > **B-spline basis functions** (Weakness 4 & Question 3)
>
> We opted to learn the witness function via a B-spline basis due to the straightforward nature of the optimization: finding the optimal witness function reduces to solving a system of linear equations.
>
> Theoretically, when the optimal witness function is $\beta$-smooth (with larger $\beta$ indicating greater smoothness), we can derive the following supremum error bound for our estimated witness function $\hat{w}$, following arguments similar to those used in the analysis of B-spline estimators (e.g., Theorem 2.1 of Chen & Christensen, 2015):
> $$\sup_{z}|\hat{w}(z)-w^*(z)|=O_p\left(\sqrt{\frac{d\log n}{n}}\right)+O(d^{-\beta}).\tag{2}
> $$
>
> Using similar arguments to the proof of our Theorem 3,  we can show that the difference in TNR between the oracle $w^*$ and the estimated $\hat{w}$ is also of order (2).
> By setting $d=\Theta\Big(\big(n/\log n\big)^{1/(2\beta+1)}\Big),$ this yields the optimal rate of convergence: $O_p\Big(\big(n/ \log n\big)^{-\beta/(2\beta+1)}\Big).$
>
> In comparison:
> - *Local polynomial partition series* estimators can achieve a similar uniform convergence rate (see Cattaneo & Farrell, 2013), and therefore yield a comparable rate in TNR difference.
> - *Neural networks* can also attain a similar convergence rate, but in terms of the $\ell_2$ norm rather than the uniform $\ell_\infty$ norm. As a result, the proof technique used in Theorem 3 does not directly apply. Instead, under the margin condition (Assumption 1), the TNR difference converges more slowly at a rate of $O_p\Big(\big(n/\log n\big)^{-\beta / (3\beta+1.5)}\Big)$. In addition, neural networks are known to learn certain non-smooth functions at near-optimal rates (see Schmidt-Hieber, 2020).
>
> Regarding the choice of dimension $d$, in line 292, we note that numerical experiments suggest: "AdaDetectGPT is generally robust to … the number of B-spline feature mappings." However, we agree that $d$ can be chosen in a more principled way:
> - If the smoothness parameter $\beta$ is known, then one can set  $d=\Theta\big(\left(n/\log n\right)^{1/(2\beta+1)}\big)$ to achieve the optimal rate in Equation (2), which in turn maximizes the expected TNR of AdaDetectGPT.
> - If $\beta$ is unknown, $d$ can be chosen in a data-driven way, for instance via Lepski's method (see Lepski & Spokoiny, 1997).
>
> > **Computational efficiency** (Question 2)
>
> From **Table A3**, the runtime of AdaDetectGPT is around 44 seconds and changes marginally with respect to $d$. This is because we can use a closed-form expression to learn a witness function. This time is nearly negligible compared with the time required to compute logits, which involves feeding tokens from multiple passages into LLMs. Furthermore, the training time when $n$ increases is shown in **Table A4**. It shows AdaDetectGPT saves more runtime and memory than ImBD.
>
> **Table A3: Runtime scale with $d$**
> d|4|8|12|16|20
> -|-|-|-|-|-
> Runtime|44.48|44.62|44.73|44.92|45.00
>
> **Table A4: Runtime (memory in parenthesis) scale with $n$ comparing with ImBD (SOTA)**
> N|100|150|200|250|300|350|
> -|-|-|-|-|-|-
> ImBD|45.65(9.56)|68.20(9.57)|93.00(9.57)|116.08(9.56)|136.53(9.56)|158.15(9.58)|
> AdaDetectGPT|9.28(0.36)|23.45(0.37)|40.19(0.37)|44.25(0.37)|59.57(0.60)|69.56(0.37)|
>
> > **Missing conclusion** (Weakness 1)
>
> During the rebuttal, we have drafted a conclusion that we plan to include using the extra page should our paper be accepted. Refer to **Missing conclusion (Minor)** in **Response to Reviewer UhoV** for details.
>
> > **TNR lower bound** (Weakness 3)
>
> We clarify the rationale behind the derivation of our lower bound below. Recall that our goal is to learn a witness function $w$ that maximizes the TNR of the resulting detector for a fixed FNR level $\alpha$. However, the TNR is a very complicated function of $\alpha$ and $w$, which makes the learned witness function that maximizes the TNR $\alpha$-dependent. Specifically, it maximizes the TNR at a particular FNR level but does not guarantee optimality at other FNR levels.
>
> To address this, we derive a lower bound on the TNR and propose learning $w$ by maximizing this lower bound. As you pointed out, this lower bound also involves $\alpha$. The key advantage is that the lower bound separates the effects of $\alpha$ and $w$: the witness function $w$ affects the lower bound only through $T_w^{(2)*}$, which is independent of $\alpha$. Consequently, the witness function that maximizes this lower bound simultaneously maximizes the bound across all FNR levels.
>
> Thus,  the problem reduces to selecting a single witness function that maximizes $\alpha$-independent $T_w^{(2)*}$.
>
> We hope this clarifies your confusion.

---

> ### Author Response · Authors · 2025-08-05
>
> Dear Reviewer cReh, once again, we greatly appreciate your constructive comments and feedback. Would you please let us know if our responses have adequately addressed your comments, or if you have any follow-up questions?

---

> ### Comment · Reviewer_cReh · 2025-08-08
> **Thanks for your response**
>
> Thank you for the comprehensive response addressing my concerns. Most of my concerns have been solved.
>
> The relaxed proportionality condition (Equation 1) with empirical validation (Table A1) and normality tests (Table A2)  strengthen the theoretical foundation by satisfying MCLT regularity conditions, while experiments across 6 LLMs and robustness against evasion techniques substantially enhance empirical validation. The B-spline theoretical justification with error bounds (Equation 2) and principled dimension selection guidelines provide valuable theoretical rigor, and runtime analysis demonstrates the practical efficiency advantages of this computationally feasible approach.
>
> By the way, I think the relaxed proportionality condition should be prominently featured in the main paper, with consideration given to empirically demonstrating Lepski's method for dimension selection, while the expanded experimental results need to be integrated into the main evaluation section. Overall, the rebuttal addresses the paper's limitations while strengthening both theoretical rigor and practical applicability of the proposed approach. I will increase my score from 3 to 4.

---

> > ### Author Response · Authors · 2025-08-08
> >
> > Thank you for raising your score! We are very delighted to hear that our rebuttal has addressed your concerns.
> >
> > Should our paper be accepted, we will endeavor to incorporate the changes you mentioned into our paper.
> >
> > Once again, many thanks for reviewing our paper and providing constructive comments!

---

### Decision · Program_Chairs · 2025-09-17

**Decision:**

Accept (poster)

**Comment:**

This paper provides a new solution on the line of traing-based and logits-based LLM-generated text detection. Basically, it is a mixture of current training based approach and zero-shot logits-based method.

This paper is interesting and the solution is with depth and good theoretical analysis as well. Extensive experiements are conducted with SOTA approaches such as DNA-GPT, ast-DetectGPT, etc. Moreover, during the rebuttal, the authors provide more results of baselines, including training based ones.

In general, the rebuttal is focusing on more baselines to compare. The authors have provided more results on it.

I think the major misunderstanding in the initial feedback was due to the lack of precision in some of the expression. The authors can refer the survey paper such as A Survey on Detection of LLMs-Generated Content. EMNLP'24 to make the background and the contribution/position more clear.

Overall, it is a good paper.